# HEARTSVG: a fast and accurate method for identifying spatially variable genes in large-scale spatial transcriptomics

Xin Yuan [1,2], Yanran Ma[1], Ruitian Gao[1], Shuya Cui [1,2], Yifan Wang[1], Botao Fa[3], Shiyang Ma[4], Ting Wei[1], Shuangge Ma [2,5] ✉ & Zhangsheng Yu [1,2,4,6] ✉

Identifying spatially variable genes (SVGs) is crucial for understanding the spatiotemporal characteristics of diseases and tissue structures, posing a distinctive challenge in spatial transcriptomics research. We propose HEARTSVG, a distribution-free, test-based method for fast and accurately identifying spatially variable genes in large-scale spatial transcriptomic data. Extensive simulations demonstrate that HEARTSVG outperforms state-of-the-art methods with higher $F_1$ scores (average $F_1$ Score=0.948), improved computational efficiency, scalability, and reduced false positives (FPs). Through analysis of twelve real datasets from various spatial transcriptomic technologies, HEARTSVG identifies a greater number of biologically significant SVGs (average AUC = 0.792) than other comparative methods without prespecifying spatial patterns. Furthermore, by clustering SVGs, we uncover two distinct tumor spatial domains characterized by unique spatial expression patterns, spatial-temporal locations, and biological functions in human colorectal cancer data, unraveling the complexity of tumors.

Spatial transcriptomics enables the measurement of gene expression and positional information in tissues[1–6]. The evolution of spatial transcriptomics technologies advanced the reconstruction of tissue structure and provided profound insights into developmental biology, physiology, cancer, and other fields[2,7,8]. However, the complexity and high dimensionality of spatial transcriptomics (ST) data pose new challenges and requirements for analytical approaches[8,9]. One crucial analytical challenge in spatial transcriptomics studies is the identification of spatially variable genes (SVGs) whose expressions correlate with spatial location[7,10,11], also known as SE genes (genes with spatial expression patterns)[12]. Identifying SVGs promotes characterizing spatial patterns within tissues and predicting spatial domains[7,10,13,14]. Several methods have been developed for detecting SVGs. Trendsceek[15] models the data as marked point processes and tests the significant

dependency between spatial distributions and expression levels of pairwise points. SpatialDE[11] decomposes gene expression variability into a spatial component and an independent noise term based on Gaussian process regression and tests statistical significance by comparing the SpatialDE model to a null model without the spatial variance component. SPARK[16], an extension of SpatialDE, uses the Gaussian process regression as the underlying data model and ten different spatial kernels to represent common spatial patterns in biological data, thereby improving statistical power. SPARK-X[12] tests the dependence of gene expressions and spatial locations based on the covariance test framework. scGCO[17] applies graph cuts in computer vision to address SVG identification. It utilizes the hidden Markov random field to identify candidate regions with spatial dependence for individual genes and tests their dependence under the complete spatial

[1]Department of Bioinformatics and Biostatistics, School of Life Sciences and Biotechnology, Shanghai Jiao Tong University, Shanghai, China. [2]SJTU-Yale Joint Center for Biostatistics and Data Science Organization, Shanghai Jiao Tong University, Shanghai, China. [3]Department of Biochemistry and Molecular Biology, School of Basic Medical Sciences, Xi'an Jiaotong University, Xi'an, Shanxi, China. [4]Clinical Research Institute, Shanghai Jiao Tong University School of Medicine, Shanghai, China. [5]Department of Biostatistics, Yale University, New Haven, USA. [6]Center for Biomedical Data Science, Translational Science Institute, Shanghai Jiao Tong University School of Medicine, Shanghai, China. ✉e-mail: shuangge.ma@yale.edu; yuzhangsheng@sjtu.edu.cn

randomness framework. Squidpy[18] uses Moran's I to determine SVG and calculates the *p*-value based on standard normal approximation from 100 random permutations.

Trendsceek, SpatialDE, and SPARK have limited applicability for large-scale datasets due to their high computational complexity. Trendsceek employs the permutation strategy to compute multiple statistics of different paired points, which requires extensive computational work and is only scalable to small-scale datasets. The Gaussian process framework hinders the detection of SVGs and model parameter convergence in SpatialDE and SPARK when analyzing high-dimensional and sparse ST data. SPARK-X offers significantly faster computational speed than the aforementioned methods, but its effectiveness depends heavily on how well the constructed spatial covariance matrix matches the true underlying spatial patterns. The above four methods identify SVGs by searching for predefined relationships between expressions and locations. They have limited generalizability to a wide range of spatial patterns due to the arbitrary nature of the true spatial pattern of SVGs and the resulting uncertainty in the relationship between expression and coordinates. scGCO has the capability to identify SVGs with unknown exact locations and shapes, however, it suffers from false negatives due to the limited accuracy of the graph cuts algorithm in identifying candidate regions for SVGs, especially in sparse ST datasets. The accuracy of Squidpy depends on the number of random permutations. Increasing the number of random permutations enhances the reliability of the results; however, it comes at the cost of increased time consumption, making the process more time-intensive.

Hence, we propose HEARTSVG to overcome the limitations without prior knowledge or specification of information about SVGs. We take an opposite approach by identifying non-SVGs and using this information to infer the presence of SVGs. Although the relationship between gene expression and spatial position of an SVG is uncertain, it is unequivocal that a non-SVG has "no relationship" between gene expression and spatial position. HEARTSVG identifies non-SVGs by testing the serial autocorrelations in the marginal expressions across global space. By excluding non-SVGs, the remaining genes are considered as SVGs. As a test-based method without assuming underlying spatial patterns, HEARTSVG detects SVGs with arbitrary spatial expression shapes and is suitable for diverse types of large-scale ST data. We conduct extensive simulations and apply HEARTSVG method to twelve real ST datasets generated from different technologies (including 10X Visium, Slide-seqV2, MERFISH, and HDST) to demonstrate its accuracy, robustness, and computational efficiency. HEARTSVG outperforms existing methods in simulations with higher accuracy metrics, computational efficiency, and lower false positives (FPs). When analyzing real ST data, HEARTSVG identifies biologically meaningful SVGs with distinct spatial expression patterns across diverse datasets obtained from different spatial transcriptomic technologies. HEARTSVG has the potential to scale to datasets comprising millions of data points and offers a comprehensive range of meticulously designed analytical tools for studying SVGs, enabling the unraveling of complex biological phenomena.

## Results
### Overview of HEARTSVG
HEARTSVG aims to identify SVGs that display spatial expression patterns in spatial transcriptomics data. Each gene in the ST data is presented as a vector containing three elements: gene $\mathbf{g} = (x, y, e)^T$, where $x$ and $y$ are defined as the row and column positions of the spot, respectively, and $e$ is the gene expression count of the gene at the spatial coordinates $(x, y)$. HEARTSVG is based on the intuitive concept that the non-SVG does not display a spatial expression pattern, its expression distribution is expected to be independent and random, with marginal expression distributions along the *x*-axis (row) and *y*-axis (column) also being independent and random. Conversely,

suppose the gene exhibits a spatial expression pattern, both its spatial expression and marginal expression should have serial correlation along the single direction (row or column) or both. Therefore, a non-SVG demonstrates low autocorrelations, while an SVG has high autocorrelations (Fig. 1, Derivations and more details are provided in the "Methods" section and Supplementary).

HEARTSVG uses the semi-pooling process to transform the gene's two-dimensional spatial expression to one-dimensional marginal expression serials along the single direction (row or column) (Fig. 1a, Supplementary Fig. S9, more details are provided in the "Methods" section and the Supplementary). This process aims to extract information and reduce data noise and sparsity from gene spatial expression data. The Portmanteau test[19,20] is then performed to test serial autocorrelations of the gene's marginal expression series. The non-SVG's marginal expressions show constant variance, zero autocorrelation, and no trend or periodic fluctuations across locations (More details are provided in the "Methods" section and the Supplementary). Conversely, marginal expressions of SVGs have high autocorrelations (Fig. 1b). We obtained multiple *p*-values by conducting the Portmanteau test to evaluate four marginal expression series with different semi-pooling parameters (More details in the Supplementary). We then combined all four *p*-values into a single *p*-value using Stouffer's method[21,22]. We applied Holm's method to adjust the final *p*-values of all genes, enabling the identification of statistically significant SVGs at a genome-wide scale. The Portmanteau test is one-sided, the Stouffer's method is two-sided. In addition, HEARTSVG provides an auto-clustering module for SVGs in the software, which is complementary to SVG detection for further biological investigations. The auto-clustering module (More details in Methods) comprises functionalities for predicting spatial domains, conducting functional studies, and visualization based on SVGs.

### Simulation
We conducted extensive simulations to evaluate the performance of HEARTSVG and compared it with five other methods: SpatialDE, SPARK, SPARK-X, scGCO, and Squidpy. Simulation data were generated with 22 spatial expression patterns that varied in different aspects, including spatial shape, percentages of the marked area, and spatial position (More details are provided in Tab. S9 in the Supplementary). The gene expression distribution in spatial transcriptomics data is complex, and no single model fits all genes. To comprehensively characterize the expression properties and ensure fair comparisons, we generated gene expression data using four distributions—Poisson (Pois), Zero-Inflated Poisson (ZIP), Negative Binomial (NB), and Zero-Inflated Negative Binomial (ZINB)—which represents different data characteristics and are widely used across spatial transcriptomics studies[23]. We used the $F_1$ score to assess the performance of HEARTSVG and the other methods in identifying SVGs. In noise-free simulated data, HEARTSVG showed higher $F_1$ scores (average $F_1$ score = 0.948) than the other methods across 22 different spatial patterns, four different data generations, and varying numbers of cells (Fig. 2a, Supplementary Fig. S62). The identification performance was influenced by the percentage of the marked area of SVGs and the number of cells/spots (Fig. 2b, Supplementary Fig. S1–S3, S62). When the number of cells and the percentage of the SVG marked area were small, HEARTSVG was able to identify more SVGs, while SPARK-X missed some SVGs, Squidpy had more false positives, and SPARK, scGCO, and SpatialDE performed poorly overall (Big Triangles vs. Small Triangles, Big Circles vs. Small Circles, Big Squares vs. Small Squares, Supplementary Fig. S62). For example, on the simulated data of Big Circles and Small Circles patterns with 3000 cells, HEARTSVG achieved higher $F_1$ scores (average $F_1$ score = 1.000, 0.992) than the other methods, while SPARK-X achieved only 0.926 and 0.710, Squidpy achieved 0.925 and 0.855, respectively, and SPARK, scGCO, and SpatialDE were close to zero. SpatialDE and SPARK performed poorly on sparse spatial

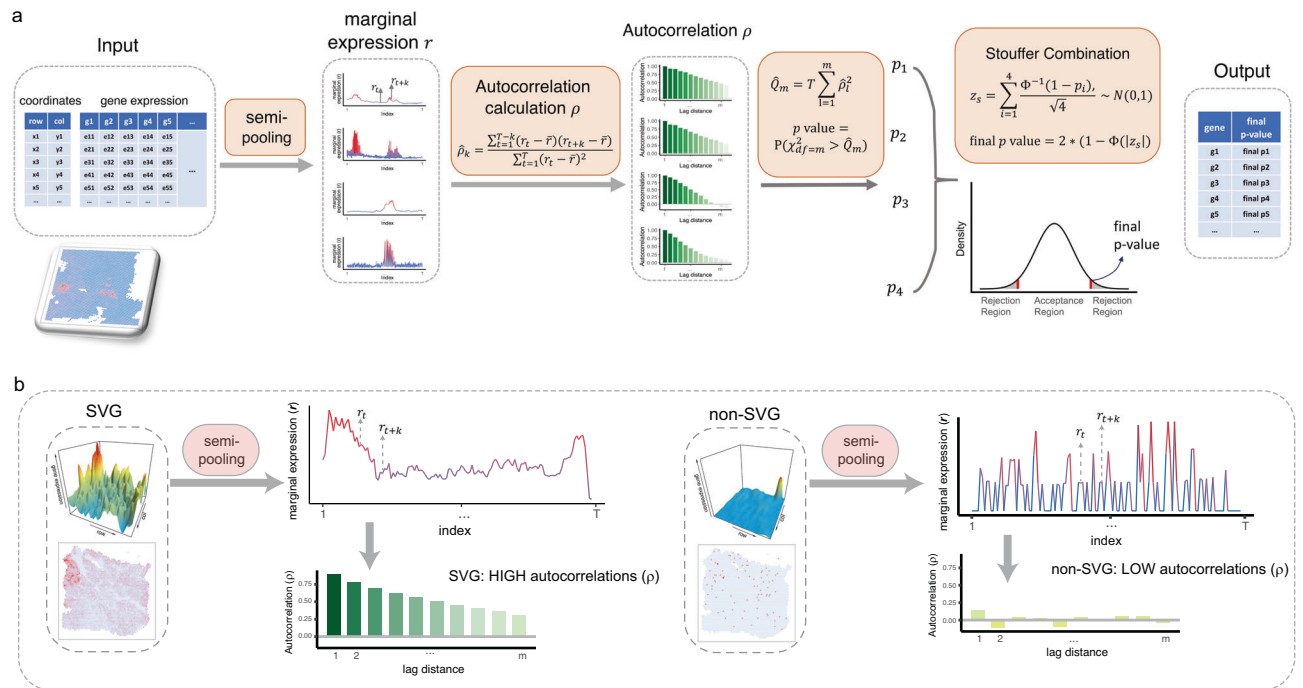

**Fig. 1 | Schematic representation of the HEARTSVG. a** HEARTSVG utilizes the semi-pooling process to convert spatial gene expressions into marginal expressions (**r**) and calculates autocorrelations (**ρ**) of marginal expressions. HEARTSVG calculates the sum of the squared autocorrelations ($\hat{Q}_m = T\sum_{l=1}^{m}\hat{\rho}_l^2$) for each marginal expression series and obtains a *p*-value by testing $\hat{Q}_m$. All these *p*-values are combined into a final *p*-value through the Stouffer combination

($z_s = \sum_{i=1}^{4}\frac{z_i}{\sqrt{4}} \sim N(0,1)$, final p $-$ value $= 2(1-\Phi(|z_s|))$). HEARTSVG distinguishes between SVGs and non-SVGs by the final *p*-value. **b** The autocorrelations (**ρ**) of marginal expressions (**r**) for the SVG and non-SVG exhibit different level scales. Representative autocorrelation estimator plots are plotted below the corresponding marginal expression plots for the SVG and non-SVG. The color depth represents the magnitude of autocorrelation.

expression data, possibly because they used a Gaussian data-generative model, which was inappropriate for ST data. Therefore, SpatialDE and SPARK used normalization mechanisms to make the ST data approximate a normal distribution. However, this normalization process removed excessive heterogeneity, including the signals from SVGs, and thus limited their ability to identify SVGs (Supplementary Fig. S87–S88). scGCO failed to identify many SVGs in highly sparse datasets because it could not detect the candidate regions for SVGs accurately. Furthermore, HEARTSVG performed stably across different spatial expression patterns of SVGs (Fig. S62).

To evaluate the robustness of HEARTSVG, we generated simulated data with three different noise generation approaches: Gaussian noise, the noise of "randomly exchanging expression values of selected nodes", and mixture noise (More details are provided in the Methods and Supplementary). We compared HEARTSVG with three other methods: SPARK-X, scGCO, and Squidpy in noisy simulations. In simulated data with Gaussian noise, HEARTSVG showed the best performance (average $F_1$ score = 0.849 at Gaussian noise strength of 0.3) among the four methods and was the most robust to increasing Gaussian noise strength. SPARK-X and Squidpy achieved the second-best identification performance (Fig. 2b, Supplementary Fig. S63–S65). For simulated data with the noise of "randomly exchanging expression values of selected nodes", we randomly selected some cells of the SVG's marked area and non-marked area and then exchanged their expression values. All methods had a substantial decline in $F_1$ score when the percentage of randomly exchanged cells increased. HEARTSVG still had the highest accuracy (average $F_1$ score = 0.618 at percentages of exchanging cells of 30%) and the lowest false positive rates (average FPR < 0.001 at percentages of exchanging cells of 30%) among the methods (Fig. 2c, Supplementary Fig. S66–S86). For simulations with mixture noise, HEARTSVG performed the highest $F_1$ scores (average $F_1$ score = 0.931) and TPRs of each gene set (average

TPR = 0.901) than the other methods (Supplementary Fig. S39–S60). To account for the uncertainty regarding the number of SVGs in real data, we generated additional simulation datasets with varying percentages of SVGs. We specifically compared the performance of HEARTSVG and SPARK-X, which showed better results in the previous simulations. As the percentage of SVGs increased, the false positive rates (FPR) of SPARK-X grew, while HEARTSVG maintained low FPRs (Fig. 2d, Supplementary Fig. S4–S6). The $F_1$ scores of HEARTSVG and SPARK-X were similar, but SPARK-X exhibited large variations of $F_1$ scores (Supplementary Fig. S4–S6). Data characteristics of different distributions significantly affect the performance of various methods in identifying SVGs. To assess the suitability of each method for different data characteristics, we conducted sensitivity analyses regarding varying data characteristics (Supplementary Fig. S89–S97). Our results indicate that scGCO is significantly affected by increased data dispersion, while HEARTSVG, SPARK-X, and Squidpy remain robust under such conditions. Increased data sparsity and lower overall expression levels generally diminish the efficiency of SVG identification. The low count of cells/spots consistently impairs all methods' ability to identify SVGs. Additionally, comparing the average false discovery proportion (FDP) across methods shows that HEARTSVG effectively controls the false discovery rate (FDR) in given simulations (Supplementary Fig. S98–S97).

Furthermore, HEARTSVG demonstrated good scalability and computational performance (Fig. 2e). HEARTSVG, SPARK-X, and scGCO can scale to datasets with one million cells. HEARTSVG and SPARK-X outperformed other methods noticeably. For simulated data with 1,000,000 cells and 10,000 genes, HEARTSVG demonstrated the fastest performance and small memory usage(13.45 mins and 416 GB), while SPARK-X required 16.43 mins and 344 GB. In contrast, scGCO demanded 21.83 hours and 924.4 GB, and Squidpy necessitated 4.93 days and 367 GB. We also evaluated the scalability using several real

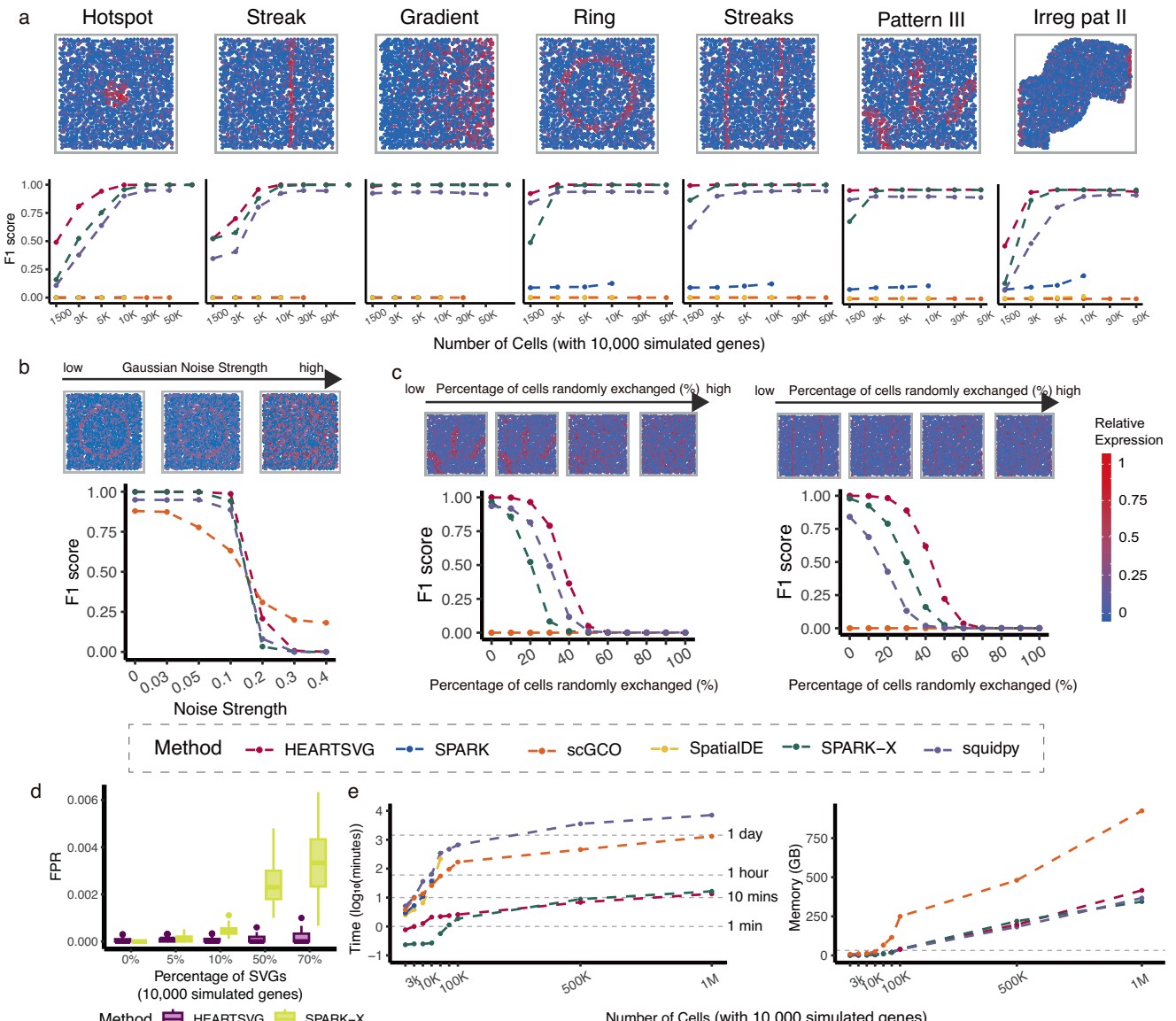

**Fig. 2 | Simulation results show that HEARTSVG has high accuracy and good scalability and computational efficiency. a** Visualization of seven representative spatial expression patterns. $F_1$ score plots compare the accuracy (y-axis) of HEARTSVG, SpatialDE, SPARK, SPARK-X, scGCO and Squidpy in simulation data with varying numbers of cells (x-axis). The $F_1$ score for each method at each noise level represents the average of 10 replications. Each replicate simulation involved $n = 3,000$ cells with 10,000 simulated genes generated using a ZINB distribution. **b** $F_1$ score plots compare four different methods across varying levels of Gaussian noise (x-axis). The $F_1$ score for each method at each noise level represents the average of 10 replications. Each replicate simulation involved $n = 3,000$ cells with 10,000 simulated genes generated using a Poisson distribution. **c** $F_1$ score plots compare four different methods across varying percentages of randomly exchanged expression values of cells (x-axis). The $F_1$ score for each method at each noise level represents the average of 10 replications. Each replicate simulation involved

$n = 3,000$ cells with 10,000 simulated genes generated using a ZINB distribution. Figure 2a, b, and c share a common legend. **d** False positive rate (FPR, y-axis) boxplots of HEARTSVG and SPARK-X in simulation data of hotspot pattern with different percentages of SVGs(x-axis). Boxplots were generated from 50 replications for each percentage of SVGs. Each replicate simulation involved $n = 5,000$ cells with 10,000 simulated genes generated using a ZINB distribution. In each boxplot, the lower hinge, upper hinge, and center line represent the 25th percentile (first quartile), 75th percentile (third quartile), and 50th percentile (median value), respectively. Whiskers extend no further than ±1.5 times the inter-quartile range. Data beyond the end of the whiskers are considered outliers and are plotted individually. **e** Plot shows time consumption in $\log_{10}$(minutes) (y-axis) and memory usage in GB (y-axis) of each method for analyzing 10,000 genes with different numbers of cells (x-axis). Source data are provided with this paper.

spatial transcriptomics datasets. On the mouse hypothalamus data, comprising 1,027,848 cells and 161 genes, HEARTSVG required 1.43 mins and 7.31 GB, scGCO needed a runtime of 112 mins and 14.72 GB, SPARK-X took 0.62 mins and 5.78 GB, and Squidpy took 3.73 mins, and 7.78 GB (Supplementary Fig. S7). Moreover, we attempted to compare the performance of HEARTSVG, scGCO, SPARK-X and Squidpy on simulated data with 2 million cells and 1000 genes. HEARTSVG completed the computation in 4.5 minutes and 82.70 GB, Squidpy took 188.5 minutes and 343.7 GB, while SPARK-X and scGCO

failed to scale to the dataset with 2 million cells. On the other hand, SPARK and SpatialDE were limited to sample sizes of 20,000 and 30,000 spots, respectively. SPARK necessitated over 3 hours for a 20,000-spot dataset, and SpatialDE took nearly 4 hours for a 30,000-spot dataset (Fig. 2e).

## Applications to ST datasets from different spatial technologies
Spatial transcriptomic technologies have various sequencing methods and yield different data characteristics. Therefore, in addition to

large-scale simulations, we evaluated the accuracy, robustness, and generality of HEARTSVG on several real ST datasets from different ST technologies, comprising three next-generation sequencing (NGS)-based spatial technologies (10X Visium[5,24–26], Slide-seqV2[3] and HDST[4]), and one imaging-based spatial technology (MERFISH[6]).

## HEARTSVG identifies SVGs and predicts spatial domains

10X Visium is the most widely used commercial spatial transcriptomics technology in cancer research. We applied HEARTSVG to a human colorectal cancer (CRC) dataset[24] generated using 10X Visium technology, involving 4174 spots and 15,427 genes. We performed unsupervised clustering and cell type annotation on this dataset, incorporating information from the Wu et al. study[24] and the hematoxylin and eosin-stained (H&E) tissue image (Fig. 3a). This tissue contains five main cell types: tumor cells, smooth muscle cells, normal epithelium, lamina propria, and fibroblast, with the tumor cells located in two distinct regions (Fig. 3a). HEARTSVG identified 8,020 SVGs, and SpatialDE, SPARK, SPARK-X scGCO, and Squidpy identified 11,190, 12,198, 13,946, 1244 and 6849 SVGs, respectively, at an adjusted p-value cutoff of 0.05. scGCO missed many SVGs with clear spatial expression patterns comparing with other methods (Supplementary Fig. S11). For instance, RPS20, RPS29, ARPC3, and GAS5 exhibited clear and similar spatial expression patterns. scGCO only identified RPS20, HEARTSVG and other three methods successfully identified all four genes. The top 10 genes ranked by HEARTSVG SPARK-X and Squidpy shown more pronounced spatial expression patterns compared to SpatialDE SPARK, and scGCO (Supplementary Fig. S12). SpatialDE's top 10 selected SVGs displayed minimal spatial patterns, while SPARK and scGCO outperformed SpatialDE to some extent. Notably, SPARK-X demonstrated a preference for selecting SVGs with large stripe patterns, aligning with previous simulation findings. The SVGs identified by HEARTSVG exhibited significant biological relevance, as confirmed by pathway enrichment analyzes conducted for each method. The enrichment analysis results (Fig. 3b) showed that HEARTSVG displayed smaller p-values and larger gene intersection sizes compared to the other five methods across 19 tumor-related KEGG pathways, including Cancer: Overview and Signal transduction. Using single-cell level common gene modules linked with tumor microenvironments[27,28] and consensus molecular markers of colorectal cancer subtypes[29,30] as reference standards for true SVGs, HEARTSVG demonstrates the highest AUCs (AUC = 0.843, 0.727, respectively), underscoring its biological interpretability. The former gene list has been widely applied in pan-cancer studies of tumor microenvironments[31–36], while the latter has found broad applications in CRC patient classification[37–43] and has been validated by various studies[37,40].

For the identified SVGs, we utilized the auto-clustering module to predict six primary spatial domains and performed enrichment analyses of the SVGs in each spatial domain (Fig. 3d–f, Supplementary Fig. S13). Some spatial domains were correlated with specific cell types, consistent with the unsupervised spatial clustering results. The SVGs in spatial domain 4 expressed highly in the muscle cell region and identified many GO (Gene Ontology) terms and KEGG pathways associated with smooth muscle cells (Fig. 3d–f). The representative genes of spatial domain 4, DES[44,45], MYL9[46], and ACTB[47,48], were essential for the functions of smooth muscle cells. The SVGs of spatial domains 1, 2, 3, and 5 showed high-expression patterns in the tumor cell regions. However, we identified some spatial domains beyond explained cell types. The SVGs of spatial domains 1 and 2 showed high expression in the left and right tumor cell regions, respectively (Fig. 3d). The spatial domain 1 was enriched in immune-associated GO terms and KEGG pathways (Fig. 3f, Supplementary Fig. S14). Several representative SVGs in spatial domain 1, such as IGKC, IGHG4, and CD28, are associated with immune infiltration[49–51] (Fig. 3e). The spatial domain 2 were enriched in the GO terms and KEGG pathways of cell differentiation[52–54]

(Fig. 3e, Supplementary Fig. S15), and included cell markers, such as EPCAM, KRT8, and CLDN3, which are connected with epithelial carcinogenesis, epithelial-mesenchymal transition (EMT) or cancer enhancement. The spatial domain 3 corresponded to the location of tumor cells. Some SVGs of the spatial domain 3, such as B2M[55–58] and FTL[57–59] are important encoding antigen genes in many cancers, as well as FTH1[60–62] and FTL[59,63,64], which are closely related to iron metabolism in cancer cells. The functional differences between the left and right tumor cells could explain why the spatial expression pattern in the right tumor cell region has clearer boundaries than the left tumor cell region.

We applied HEARTSVG on two other colorectal cancer ST datasets and corresponding liver metastasis ST datasets from the same cohort. HEARTSVG had a higher AUC (average AUC = 0.792) than other methods (Fig. S13). In the six colorectal cancer and liver metastasis spatial transcriptomic (ST) datasets, we detected higher expressions of numerous mitochondrial-encoded genes in the tumor cells compared to the non-tumor region within the colorectal tumor samples. However, this phenomenon was not observed in the liver metastasis samples (Fig. S20). We supposed that tumor cells at the primary site of colorectal cancer have higher oxidative phosphorylation (OXPHOS) activity than metastatic liver cancer sites, in line with recent studies showing OXPHOS upregulates in colorectal cancer[65–68]. Overall, HEARTSVG successfully detected SVGs with visually distinct patterns. The auto-clustering module effectively predicted spatial functional domain based on the distinguished SVG patterns positioned in and beyond the cell types.

## HEARTSVG detects SVGs explained by cell types

Slide-seqV2[3] is a spatial transcriptomics technology that achieves transcriptome-wide measurements at near-cellular resolution. We applied HEARTSVG to mouse cerebellum data generated by Slide-seqV2, consisting of 20,141 genes measured on 11,626 spots. The cerebellum plays a crucial role in sensorimotor control[69–71] and consists of the cortex, white matter, and cerebellar nuclei[72]. The cerebellar cortex comprises three cortical layers[70] from the outside to the inside: the molecular layer (ML), the Purkinje layer (PCL), and the granular layer (GL). Purkinje cells are a unique kind of neuron in the cerebellar cortex and constitute a slight, convoluted monolayer.

HEARTSVG, SpatialDE, SPARK, SPARK-X scGCO, and Squidpy detected 710, 1,086, 421, 586, 68, and 1564 SVGs, respectively. We supported the validity of SVGs detected by HEARTSVG in two pieces of evidence. First, HEARTSVG identified marker genes of specified cell types with spatially restricted expression patterns (Fig. 4a–d). For example, Mbp (adjusted p-value = 0) in oligodendrocytes, Car8 (adjusted p-value = 0) in Purkinje cells, and Clbn1(adjusted p-value = 0) in granule cells. Notably, HEARTSVG detected the marker genes of Purkinje cells (Fig. 4a–c, Supplementary Fig. S20, and Supplementary Table S3), Car8 (adjusted p-value = 0), Pcp2 (adjusted p-value = 0) and Pcp4 (adjusted p-value = 0), whereas SPARK- failed to identify them. scGCO failed to detect several SVGs with distinct spatial expression patterns, including Calm1, Calm2, Itm2b, among others, which were successfully identified by the other four methods (Supplementary Fig. S20–S21, and Supplementary Table S3). Second, we performed tissue specificity enrichment analysis for the SVGs identified by each method. HEARTSVG, SpatialDE, SPARK, SPARK-X, scGCO, and Squidpy enriched 40, 51, 97, 50, 26, and 56 tissue specificity pathways (Fig. 4d and Supplementary Table S2). The enriched tissue-specific pathways identified by HEARTSVG and scGCO were all related to the brain, with high percentages (87.5%, 35 cerebellar pathways and 92.31%, 24 cerebellar pathways) of enriched tissue-specific pathways in the cerebellum, and the remaining pathways associated with the cerebral cortex and hippocampus. Although SPARK identified the highest number of pathways (97 pathways), over 40% of these pathways were unrelated to the brain, including 36 skin-specific pathways (37.11%) and

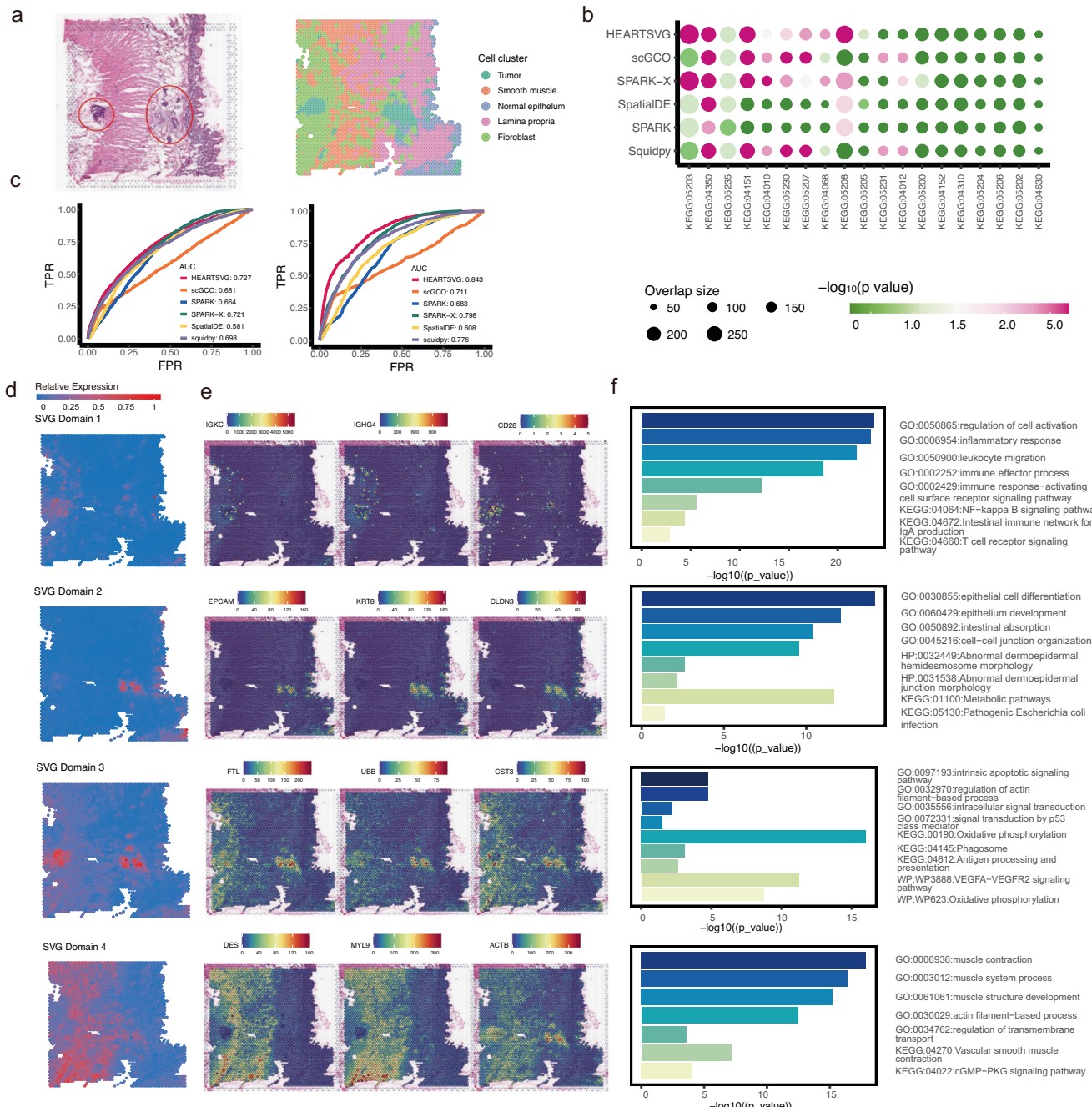

**Fig. 3 | HEARTSVG identifies tumor related SVGs and predicts spatial functional domains with distinct biological functions in the 10X Visium colorectal cancer data. a** Original hematoxylin and eosin stained (H&E) tissue image (left) and results of unsupervised spatial clustering (right). The red-circled areas in the HE image represent the tumor regions. **b** The bubble plot illustrates the results of KEGG pathway enrichment analysis for 19 tumor-related pathways (*x*-axis) across different methods. Each bubble represents a pathway, and its size corresponds to the overlap gene size of the pathway and SVGs detected by each method. The *x*-axis and *y*-axis of the plot represent different methods and their significance ($-\log_{10}(\text{p}-\text{value})$). *P*-values were calculated by g:Profiler. **c** The ROC curves

illustrate the TPR and FPR of six different methods when using common gene modules of tumor microenvironments (left) and consensus molecular markers of colorectal cancer subtypes (right) and as gold standards for true SVGs. AUC is the area under the ROC curve. **d** HEARTSVG predicts four spatial domains based on SVGs and graphed the average expression of SVGs in each spatial domain. **e** Representative SVGs correspond to the four predicted spatial domains in Fig. 3d. **f** Enrichment analysis corresponds to the four predicted spatial domains. The length of bars represents the enrichment using $-\log_{10}(\text{p}-\text{value})$. *P*-values were calculated by g:Profiler. Source data are provided with this paper.

three rectum-specific pathways (3.09%). SpatialDE, SPARK-X and Squidpy also identified some enriched pathways that were not associated with the brain. SpatialDE identified one rectum pathway (1.96% of the total pathways), SPARK-X identified three rectum pathways (6%) and three skin pathways (6%), and Squidpy identified one endometrium pathway (1.79%). The heatmap (Fig. 4e) of SVGs detected by

HEARTSVG corresponding to the molecular, Purkinje and granule layers of the cerebellum confirmed the biological interpretability of the SVGs detected by the HEARTSVG. These findings demonstrated that HEARTSVG is a reliable method for detecting SVGs exhibiting arbitrary spatial patterns in structurally complex tissues, such as the brain.

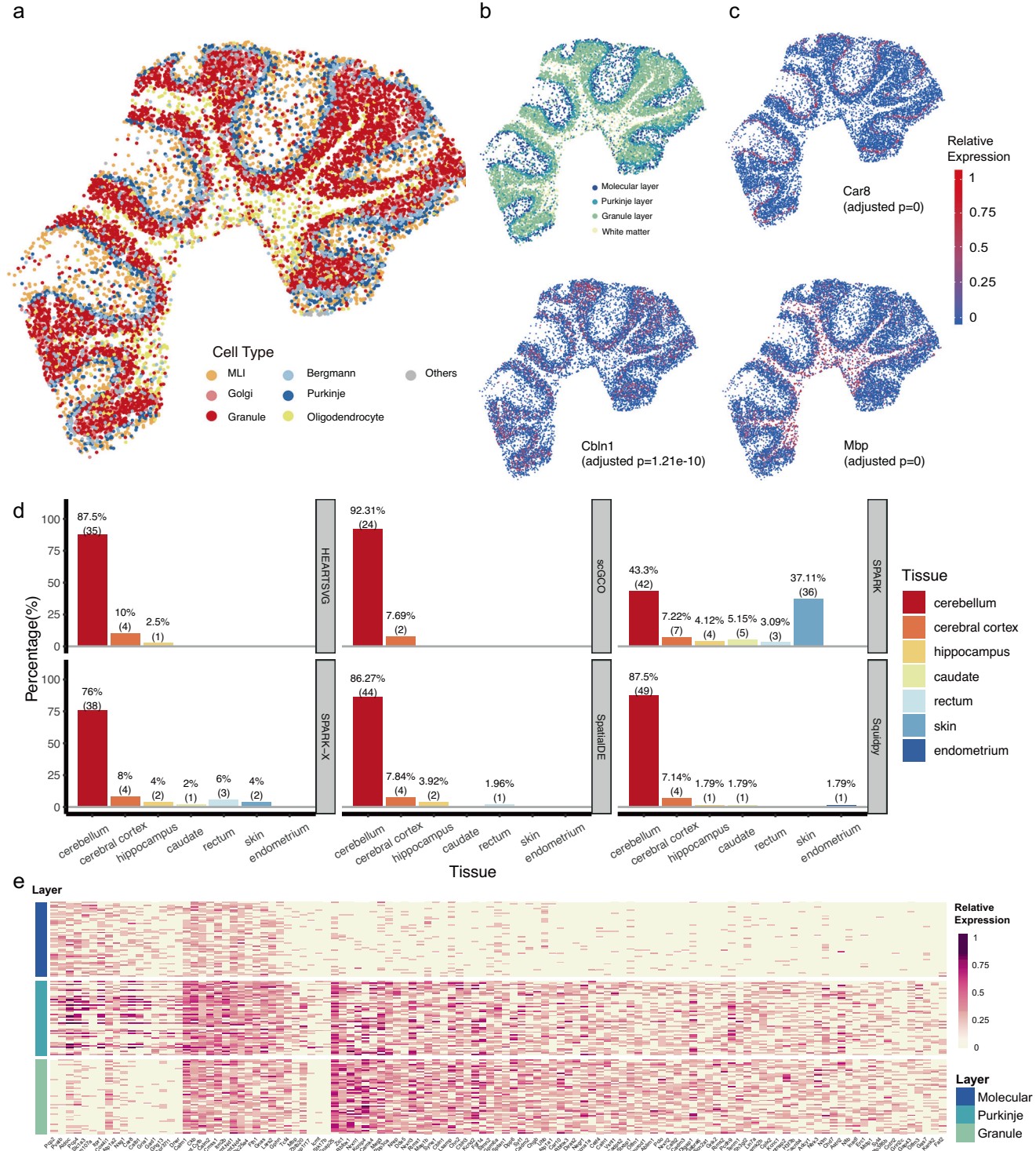

**Fig. 4 | HEARTSVG detects biologically meaningful SVGs in the Slide-seqV2 cerebellum data. a** Visualization of unsupervised spatial clustering results. **b** Visualization of cerebellum layer annotations. **c** Visualizations of spatial expressions and adjusted *p*-values of representative SVGs of specified cell types detected by HEARTSVG. Car8 (adjusted *p*-value = 0) in Purkinje cells, Clbn1(adjusted *p*-value = 1.21e-10) in granule cells and Mbp (adjusted *p*-value = 0) in oligodendrocytes. *P*-values were calculated by HEARTSVG and adjusted using Holm's method. **d** The tissue-specific enrichment analysis results for each method, where the *x*-axis represents different tissues, and the *y*-axis represents the percentage of tissue-specific pathways. Each panel corresponds to each method. **e** Heatmap of SVGs expressions in Molecular layer, Purkinje layer, and Granule layer. Source data are provided with this paper.

## HEARTSVG identifies marker genes with spatial patterns

We analyzed two datasets of mouse preoptic hypothalamus generated by multiplexed error-robust fluorescence in situ hybridization[73] (MERFISH). MERFISH enabled spatially resolved RNA analysis of individual cells with high accuracy and high detection efficiency[5]. The data generated through MERFISH were moderately sparse, with more than 40% of the genes detected in more than half of the cells. The first dataset involved 6,112 cells and 155 genes and consisted of eight cell types (Fig. 5a). The second dataset consisted of 10 cell types (Fig. 5b) involving 5,665 cells and 161 features (156 genes and five blank

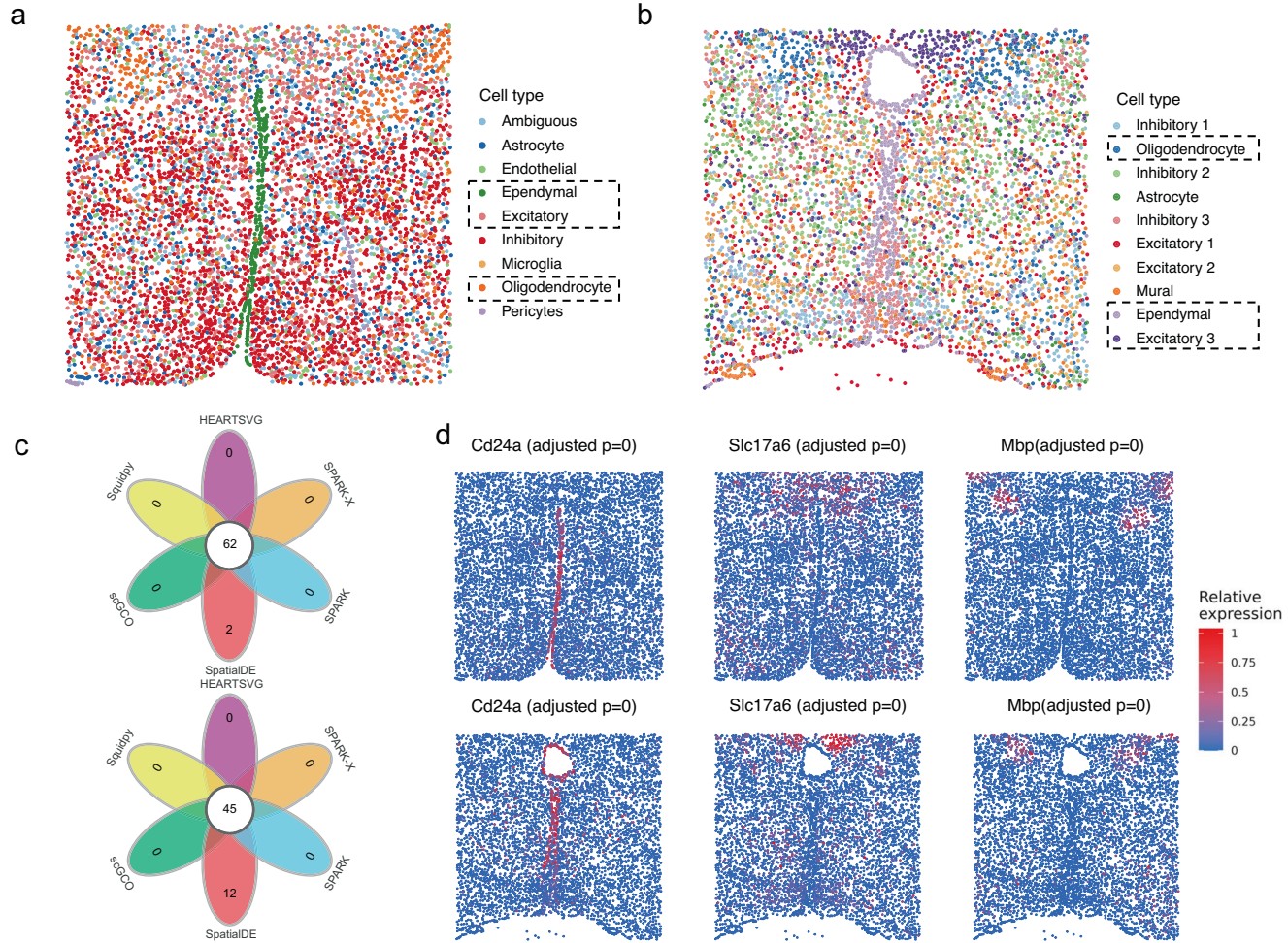

**Fig. 5 | HEARTSVG identifies cell type-specific marker genes with distinct spatial patterns across different MERFISH datasets. a** Visualization of known cell annotations of the MERFISH data 1 (6,112 cells with 155 genes). **b** Visualization of unsupervised spatial clustering results of the MERFISH data 2 (5665 cells with 156 genes and 5 blank controls). **c** Venn diagrams of SVGs identified by HEARTSVG, SpatialDE, SPARK, SPARK-X, scGCO and Squidpy. **d** Visualizations of representative marker genes for ependymal, excitatory, and oligodendrocyte cell types, along with corresponding adjusted *p*-values obtained by HEARTSVG in two MERFISH datasets. *P*-values were calculated by HEARTSVG, and adjusted using Holm's method. The panels above of Fig. 5c and Fig. 5d correspond to the MERFISH data 1. The panels below correspond to the MERFISH data 2. Source data are provided with this paper.

controls). HEARTSVG, SpatialDE, SPARK, SPARK-X, scGCO and Squidpy identified 133, 154,149, 141, 65, and 145 genes in the first MERFISH dataset and 128, 161,145, 132, 46, and 144 genes in the second MERFISH dataset. The results of all methods were highly consistent (Fig. 5c, Supplementary Fig. S22). However, SpatialDE misclassified five blank controls as SVGs with top gene ranks. HEARTSVG, SPARK, SPARK-X and Squidpy reported one blank control as false positive with low ranks and no false positives reported by scGCO. However, scGCO missed some SVGs with clear spatial expression patterns, such as Mbp in the MERFISH data 2 (Fig. 5d), Nnat in in these two MERFISH data (Supplementary Fig. S22).

In both datasets, HEARTSVG efficiently identified SVGs associated with cell types spatially located in specific regions (Fig. 5c, S20). For example, HEARTSVG detected Cd24a (adjusted *p*-value = 0), Mlc1 (adjusted *p*-value = 0), and Nnat (adjusted *p*-value = 0) as significantly associated SVGs in ependymal[74,75], Slc17a6 (adjusted *p*-value = 0)[76–78], Cbln2 (adjusted *p*-value = 0), Necab1 (adjusted *p*-value = 0), and Ntng1 (adjusted *p*-value = 0) in excitatory neurons[79,80]. The oligodendrocyte (OD) markers[74,75,81], including Mbp (adjusted *p*-value = 0), Ermn (adjusted *p*-value = 0), Ndrg1, and Sgk1 (adjusted *p*-value = 0), were also accurately identified. We utilized the auto-clustering module to obtain multiple spatial domains. The resulting spatial domains consistently matched their corresponding cell types (Fig. S22). For example, in the first data, we predicted two spatial domains corresponding to Oligodendrocyte and Excitatory 3 neurons, respectively. Overall, the auto-clustering module highlighted the usefulness of the software HEARTSVG.

## HEARTSVG has general applicability across various datasets

To evaluate the generality of HEARTSVG, we applied it to a more comprehensive range of datasets, including mouse olfactory bulb data generated by high-definition spatial transcriptomics (HDST) and ST datasets of two different cancers using 10X Visium. The HDST dataset[4] was huge and sparse, consisting of 181,367 spots and 19,950 genes, with more than 98% of spots detecting less than 50 genes. Only HEARTSVG, SPARK-X, scGCO, and Squidpy could flawlessly be operated on the HDST data and detected 447, 89, 0, and 248 SVGs, respectively. scGCO failed to identify any SVGs in this sparse HDST dataset. HEARTSVG identified top-ranked SVGs (Gm42418, mt-Rnr1, mt-Rnr2, Cmss1, Gphn) that showed pronounced spatial expression patterns (Supplementary Fig. S23), although visual-spatial expression patterns of genes were challenging to observe in such sparse data.

10X Visium is the most popular ST technology in cancer research. Therefore, we applied our method to analyze additional ST data generated by 10X Visium, including a primary liver cancer (PLC) ST dataset[25], and a renal clear cell carcinoma with brain metastasis (RCC-BM) ST dataset[26], aiming to showcase the superior performance of HEARTSVG. Consistent with previous applications, the tumor cells in these datasets exhibited complexity and high heterogeneity, encompassing multiple tumor cell types with diverse functions within the same tissue. HEARTSVG effectively identified tumor-related SVGs in cancer ST data and predicted several spatial domains with different functionalities. For example, the PLC ST data contained three distinct tumor cell types. The identification of tumor-associated SVGs by HEARTSVG, along with the prediction of their corresponding spatial domains, revealed a potential synergistic function among these cell types (Supplementary Fig. S24). In the RCC-BM ST data, we found two spatial domains showing different high SVG expressions, corresponding to tumor small nests and tumor medium/big nests[26] (Supplementary Fig. S25), respectively. The regions of tumor small nests and tumor medium/big nests in this sample were adjacent. Some immune-related genes, such as CD44[82,83] and CD14[82,84,85] are highly expressed in the tumor small nest region. Moreover, we found that many tumor-related genes showed higher expression in the "small nests" of tumors than in the "large nests" (Supplementary Fig. S25), which is consistent with the study of Sudmeier et al. [26]. SVG detection contributed to providing further insights into intertumoral and intratumoral genetic heterogeneity and complex tumor microenvironments (TME) and cancer mechanisms, which is critical to understanding tumor progression and response to therapy.

## Discussion

We proposed HEARTSVG, a distribution-free, test-based method, for rapid and precise detection of SVGs in large-scale spatial transcriptomic data. Different from existing SVG detection methods[11,12,15–17], HEARTSVG uses an alternative strategy that employs the exclusion of non-SVG genes to infer the existence of SVGs, allowing it to identify SVGs of any spatial expression patterns with high accuracy, robustness, and generalizability across various ST datasets from different spatial technologies. Benefiting from the test framework and absence of underlying data-generative models, HEARTSVG has superior computational efficiency and scalability, highly suitable for large-scale spatial transcriptomics data. Moreover, the HEARTSVG software offers various functionalities for advanced analysis of SVGs, including auto-clustering, enrichment analysis, and visualization tools.

Our study evaluated the performance of HEARTSVG on both simulated and real ST data, demonstrating its accuracy, robustness, and generality in various scenarios, including varying numbers of cells, percentages of marked area of SVGs, spatial patterns, and spatial transcriptomic sequencing technologies. HEATSVG had the highest $F_1$ scores in most simulation scenarios and had good scalability and computational efficiency. HEARTSVG, SPARK-X, scGCO and Squidpy were able to successfully run on a dataset of one million cells. However, HEARTSVG and SPARK-X exhibited lower time consumption than scGCO and Squidpy. scGCO achieves excellent FPR control, but its performance is hampered by overlooking a substantial number of SVGs in sparse simulated datasets, due to inaccuracies in candidate region identification. Other studies have also revealed limitations of scGCO in identification of SVGs[86–90]. In the simulated datasets with increasing percentages of SVGs, SPARK-X had increasing FPRs while HEARTSVG maintained low FPRs. Besides, HEARTSVG can detect SVGs with diverse spatial patterns, while SPARK-X has pattern preferences in recognizing SVGs and has difficulty detecting some non-striped patterns and small percentages of marked areas of SVGs.

We implement HEARTSVG on twelve datasets from four different spatial transcriptome sequencing technologies (10X Visium, Slide-seqV2, HDST, and MERFISH) across three different tissues (colorectal, liver, and brain). HEARTSVG exhibited the highest AUC (average AUC = 0.792), demonstrating its accuracy and robustness across datasets with diverse data characteristics. The brain is a complex organ with intricate structures and a wide variety of cell types in constrained regions[71,91–93]. HEARTSVG can sensitively identify cell-type markers that are restricted to specific brain regions. For example, HEARTSVG identified the markers, Car8, Pcp2, and Pcp4 of the thin and curly Purkinje cell layer, which SPARK-X failed to identify. Despite favorable FPRs, scGCO's inaccurate identification of candidate regions limits its capacity to fully recognize SVGs with similar spatial expression patterns. SpatialDE misidentified five blank control genes as SVGs with small adjusted $p$-values in the MERFISH preoptic hypothalamus data. We performed tissue-specific enrichment analysis of the SVGs identified by each method and illustrated the biological benefits of HEARTSVG. The enriched tissue-specific pathways identified by HEARTSVG and scGCO were all related to the brain. In contrast, SPARK identified more than 40% of enriched tissue-specific pathways that were unrelated to the brain in the Slide-seqV2 cerebellum data. This indicates that the reliability of the SVGs identified by SPARK was limited. SpatialDE (1.96%) and SPARK-X (14%) also had pathways unrelated to the brain. Only HEARTSVG, SPARK-X and scGCO can identify SVGs on the huge HDST dataset (180 K- spots and 19,000- genes) and mouse hypothalamus MERFISH data (1 million cells and 161 genes), demonstrating HEARTSVG's excellent computing efficiency and scalability.

In this study, we conducted analyses of ST datasets for three different types of cancer (colorectal cancer, primary liver cancer, and renal cell carcinoma brain metastasis), which were generated using 10X Visium - a widely used commercial ST technology in cancer research. The ST data of tumors contained few cell types and their SVGs are primarily associated with tumor cells. HEARTSVG performed well on different cancer ST datasets, and pathway analysis results demonstrated its ability to identify many tumor-related SVGs. HEARTSVG (8 significant pathways), SPARK-X (8 significant pathways), scGCO (7 significant pathways), and Squidpy (7 significant pathways) identified more cancer-related KEGG pathways than SpatialDE (2 significant pathways) and SPARK (2 significant pathways) in the 10X Visium colorectal cancer data. Furthermore, the SVG auto-clustering module of the software HEARTSVG facilitated the prediction of different tumor-associated spatial domains with distinct spatial expression patterns. In the colorectal cancer ST data, tumor cells were located in two non-adjacent regions of the sample. We discovered that two tumor-associated spatial domains had high expression patterns in only one tumor cell region instead of both, as shown in Fig. 3. Enrichment analysis revealed distinct biological processes and functions associated with the two spatial domains. We observed similar phenomena in the ST datasets of primary liver cancer and renal clear cell carcinoma with brain metastasis. In the PLC ST data, many SVGs were highly expressed in both tumor cell subtypes 1 and 3, constituting a common spatial functional domain. In the RCC-BM ST dataset, we identified two adjacent spatial domains based on different SVG clusters, corresponding to tumor small nests and tumor medium/big nests, respectively. Spatial domain prediction based on SVGs has revealed tumors' intricate functional diversity and synergistic interactions beyond cellular classifications, shedding new light on the biological complexity of tumor tissues.

Overall, HEARTSVG is a powerful method for detecting spatially variable genes with the ability to identify spatial expression patterns of arbitrary shapes. Moreover, the inclusion of an auto-clustering module in the HEARTSVG software enhances the understanding of the

biological process, demonstrating the versatility and potential of HEARTSVG in spatial transcriptomics data analysis. However, HEARTSVG has such limitations as relying solely on spatial coordinates. In future studies combing gene expression with corresponding H&E tissue images, incorporating information from H&E tissue images will provide a more comprehensive understanding of the cellular mechanism in disease progression.

## Methods

### Identification of spatially variable genes

In spatial transcriptomics (ST) data, each gene can be represented by a vector containing three elements: the gene $\mathbf{g} = (x, y, e)^T$, where $x$, $y$, and $e$ correspond to the row coordinates, column coordinates, and the expression counts of the gene on the spot at the $(x, y)$ (Fig. S7). To simplify notation, we assume in the following proof that there is only one gene. HEARTSVG tested for each gene, so the "only one gene" assumption does not affect the derivation and conclusion. We determine whether g is an SVG by testing whether the expression of the gene is randomly distributed in the ST data. In practice, we assume that the expression counts of the non-SVG gene at a given location $(x_i, y_i)$ are independent of expressions at nearby locations. Therefore, we applied the Portmanteau test to test several autocorrelations of $r_t$ that are simultaneously at zero to determine whether the gene is an SVG. $r_t$ is the gene marginal expression series after the semi-pooling step. The null and alternative hypotheses are:

$$H_0 : \rho_1 = , \ldots, = \rho_m = 0, H_A : \exists\, k \in \{1, \ldots, m\}, \rho_k \neq 0 \quad (1)$$

To simplify the symbolic representation, we rewrite the subscript of the marginal expression series as $\mathbf{r} = (r_1, .., r_t, \ldots, r_T)^T$, define the autocovariance of order $k$ as:

$$\gamma_k = Cov(r_t, r_{t-k}) = Cov(r_t, r_{t+k}) \text{ for all } k \geq 0 \quad (2)$$

and the $k-$th order autocorrelation (ACF) as

$$\rho_k = \frac{\gamma_k}{\gamma_0} \quad (3)$$

If the gene is non-SVG without a spatial pattern in ST data, our purpose is to test the null hypothesis: $H_0 : \rho_1 = \cdots = \rho_m = 0$. The test statistic is defined as $Q_m = T \sum_{l=1}^{m} \hat{\rho}_l^2$ followed by chi-distribution with $m$ degree of freedom, where $\hat{\gamma}_k = \frac{1}{T-k} \sum_{t=1+k}^{T} (r_t - \bar{r})(r_{t-k} - \bar{r}), k = 0, \ldots, T-1$, $\bar{r}$ is the mean of $\mathbf{r}$, $m = \ln(T)$, and introduce $\hat{\rho}_k = \frac{\hat{\gamma}_k}{\hat{\gamma}_0}$. The $p$-value for testing the null hypothesis can be calculated by

$$p = P\left(\chi^2(df = m) > Q_m \mid H_0 \text{ is true}\right) \quad (4)$$

We combined all individual $p$-values into a single $p$-value by Stouffer's method. Stouffer's method is a classic $p$-value combination method that tends to pick up consistent effects and is more robust in the presence of rare outliers[94]. The Stouffer's statistic is defined as

$$z_{stouffer} = \sum_{i=1}^{4} \frac{z_i}{\sqrt{4}} \sim N(0, 1) \quad (5)$$

where $z_i = \Phi^{-1}(1 - p_i)$, $\Phi^{-1}(\cdot)$ is the inverse of the cumulative distribution function of a standard normal distribution. Hence, the combined $p$-value of four $p$-values is calculated by $p_c = 1 - \Phi(z)$. We use the continuously adjusted combined $p$-value to determine whether a gene is an SVG. The final $p$-values of all genes were adjusted with the Holm's method. If the adjusted p-value of a gene is less than 0.05, it is recognized as an SVG.

### Auto-clustering module

The auto-clustering module utilizes the hierarchical clustering algorithm and includes the following steps.

Step 1: Calculate the similarity between each pair of genes based on spatial expression and generation of the distance matrix.

Step 2: Construct a clustering tree based on the distance matrix using the complete linkage criterion. The resulting hierarchy of clusters can be visualized as a dendrogram.

Step 3: Determination of the final clustering results by cutting the dendrogram at a certain height or distance threshold. The cutting height is chosen using the maximum breakpoint of all breakpoints selected by the Yamamoto test[95,96].

We predicted spatial domains based on each SVG cluster's regions and expression levels.

### Simulation Design

We generated extensive simulation scenarios to evaluate the performances of HEARTSVG and five other existing SVG methods. Each scenario had 20 replications. For spatial expression pattern settings, we set 22 different spatial expression patterns (More details in Supplementary Table S1 and S5). We generated the spatial locations of spots by the random-point-pattern Poisson process (intensity parameter lambda in the noise of "Randomly Exchanging Nodes" is 0.7, others are 0.5). The expression counts are generated from the zero-inflated negative binomial (ZINB) distribution, negative binomial (NB) distribution, Poisson distribution, and zero-inflated Poisson (ZIP) distribution (More details of distribution parameters in Tab. S1). In noise-free simulated data, we simulated 10,000 genes (1000 SVGs and 9000 non-SVGs) and varied the number of spots from 1500, 3000, 5000, 10,000, 30,000, and 50,000 (Supplementary Fig. S1–S3 and S62). Furthermore, we compared the false positive rates and $F_1$ scores of HEARTSVG and SPARK-X with the variation in the percentages of SVGs. We varied the percentages of SVGs from 0%, 5%, 10%, 50%, and 70%, and the number of spots from 3000, 5000, and 10,000. Other simulation settings were similar to noise simulations of from ZINB distribution.

Regarding the simulations with noises, we generated simulated data with three different noise generation approaches: Gaussian noise, the noise of "randomly exchanging expression values of selected nodes" and mixture noise. We added six different levels of Gaussian noise to simulated data with four different distributions and 22 spatial patterns and created six noisy simulated data of each spatial pattern (more details in Supplementary-Section 10.3). For simulated data with noise of "randomly exchanging expression values of selected nodes", we followed the procedures described by scGCO. We randomly selected varying percentages spots from the marked and non-marked areas of the SVGs and swapped their expression values. For simulated data with mixture noise, we generated 1000 simulated SVGs and randomly rearranged the gene expressions to generate non-SVGs. Then, we mix their expression to create non-SVGs, SVGs with noise, and non-SVGs with noise (Fig. S38).

Like other SVG detection methods, we use the continuously adjusted $p$-value to determine whether a gene is an SVG. If the adjusted $p$-value of a gene is less than 0.05, it is identified as an SVG. Based on this criterion, we converted continuously adjusted $p$-values to binary results and calculated performance indices. The $F_1$ score is a measurement of accuracy that balances precision and recall. The calculations of performance indices were as follows.

TP: True positive.

FN: False negative.

FP: False positive.

TN: Ture negative.

$$\text{Precision} = \frac{\text{TP}}{\text{TP} + \text{FP}}$$

$$\text{TPR} = \frac{\text{TP}}{\text{TP} + \text{FN}}$$

$$\text{FPR} = \frac{\text{FP}}{\text{FP} + \text{TN}}$$

$$F_1 \text{ score} = \frac{2 * \text{Precision} * \text{Recall}}{\text{Precision} + \text{Recall}}$$

## Statistics & Reproducibility

In this study, no statistical method was used to predetermine sample size. All data used in this study were collected from public resources and used to demonstrate the performance of HEARTSVG. We performed quality control of spatial transcriptomics data based on the commonly used and pre-established criteria in this field. For real data, low-quality genes detected in less than 1% of spots were excluded from the analysis. The experiments were not randomized. Analyzes were conducted exclusively on published data, as documented in their original publications, precluding blinding by investigators during reanalysis.

## Reporting summary

Further information on research design is available in the Nature Portfolio Reporting Summary linked to this article.

## Data availability

All data analyzed in this manuscript are available in their raw form from the respective original authors. (1) The 10X Visium data of colorectal cancer are available at the Single-Cell Colorectal Cancer Liver Metastases (CRLM) Atlas [http://www.cancerdiversity.asia/scCRLM]; (2) The Slide-seqV2 data are available at the Single Cell Portal [https://singlecell.broadinstitute.org/single_cell/study/SCP815]; (3) The MERFISH datasets are available in the Dryad Digital Repository from [https://doi.org/10.5061/dryad.8t8s248]; (4) The mouse olfactory bulb data generated by high-definition spatial transcriptomics (HDST) are available at the NCBI Gene Expression Omnibus (GEO) database repository under accession code GSE130682; (5) The 10X Visium data of primary liver cancer are available at the Genome Sequence Archive (GSA) under accession code HRA000437; (6) The 10X Visium data of renal clear cell cancer brain metastasis are available at the NCBI Gene Expression Omnibus (GEO) database repository under accession code GSE179572. Source data are provided in this paper.

## Code availability

The HEARTSVG is implemented in R, and is available on GitHub (https://github.com/cz0316/HEARTSVG) and Zenodo[97].

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

## Acknowledgements

The computations in this paper were run on the Siyuan-1 cluster supported by the Center for High-Performance Computing at Shanghai Jiao Tong University. We express our gratitude to Ms. Yudi Chen and Dr. Panpan Zhang for their valuable feedback on the manuscript. We sincerely appreciate the helpful advice on codes provided by Ph.D. students Jie Zhou and Mr. Zhaochang Yang. Additionally, we are grateful to Ms. Kaiqi Zhang, Ms. Xiwen Sun, Ms. Congwen Xiao, and Ms. Xiaoya Sun for their helpful discussions. This study was supported by grants from the National Natural Science Foundation of China (Grant No. 12171318 to Z.Y.), the Shanghai Science and Technology Commission (Grant No. 21ZR1436300, 23XD1401900, and 23DZ2290600 to Z.Y.), the Shanghai Jiao Tong University STAR Grant (Grant No. 20190102 to Z.Y.), the Medical Engineering Cross Fund of Shanghai Jiao Tong University (Grant No. YG2023ZD21 to Z.Y.), the Fundamental Research Funds for the Central Universities (Grant No. YG2023QNA01 to S.-Y.M.), the Shanghai Rising-Star Program (Grant No. 23YF1421000 to S.-Y.M.), the Clinical Research Project of Shanghai Municipal Health Commission in Health Industry (Grant No. 20234Y0285 to S.-Y.M.), the Shanghai Science and Technology Commission (Grant No. 20JC1410100), and Yu Lab.

## Author contributions

X.Y., Z.Y., and S.-G.M. designed the HEARTSVG algorithm and the simulation framework. X.Y. implemented the HEARTSVG software, performed data analyses, and conducted comparisons. Z.Y. and S.-Y.M. secured funding for the study. Y.M., R.G., and T.W. were responsible for dataset preprocessing. Y.W., S.C., and B.F. contributed to figure design and performed analyses using real data. X.Y. and Z.Y. wrote this paper. All authors reviewed and approved the final manuscript.

## Competing interests

The authors declare no competing interests.
