## [Peer Review File · Nature Communications]

HEARTSVG: a fast and accurate method for identifying spatially variable genes in large-scale spatial transcriptomicsReviewer #1 (Remarks to the Author):

1. One of the key advantage claimed by the authors is scalability. However, their evaluation stopped at 50,000 cells. In comparison, the authors of scGCO evaluated performance up to 1 million cells. Thus additional evaluation is necessary. I would suggest the authors to evaluate both simulated data up to 1 million cells and real biological sample with millions of cells. Furthermore, in addition to running time, it is essential to profile the memory footprint of different methods to gain a holistic picture of these methods' scalability.

2. I am strong concerned about how the authors calculated performance metrics (such as AUC, F1 etc) on real biological samples. Unlike fields such as image classification in computer vision, where the class label of each image in the data set is authentic and known, our current understanding of SV genes is rather limited and for each biological sample, the true set of SV genes are unknown. The authors selected the top 500 overlaps between HEARTSVG and SPARK-X to form a set of true SVGs. This is a very biased approach and will strongly favor HEARTSVG and SPARK-X. Using genes selected by HEARTSVG to evaluate the performance of HEARTSVG is the same as evaluate a method's performance using the training data, it is fundamentally flawed.

3. Given that the true set of SV genes are generally unknown in real biological samples, the evaluation based on simulated data is critical. Unfortunately the authors only simulate three simple patterns. Additional patterns are necessary to gain an unbiased picture. The authors should demonstrate that the patterns are comprehensive enough to capture major SV trends. Furthermore, because all methods will perform extremely well on noise-free simulated data, varying level of noises should be added to evaluate the robustness of different methods.

Reviewer #2 (Remarks to the Author):

The manuscript entitled, "HEARTSVG: a fast and accurate method for spatially variable gene identification in large-scale spatial transcriptomic data" by Yuan et al. presents an interesting new methodology to identify spatial variable genes (SVGs). This work aims to identify SVGs without using pre-determined assumptions about gene expression spatial distribution. The authors focus on identifying non-SVGs genes to define a background spatial expression and then infer SVGs using both semi-pooling and autocorrelation. The authors have analyzed a significant number of datasets (n=12) across different sequencing technologies and investigated data sparsity, distinct spatial patterns, number of spots, SVGs abundance and ZINB parameters on simulated data. Additionally, in the revised version the authors include a comparison with other SVGs detection methods such as scGCO, spatialDE, SPARK and SPARK-X. The authors show that HEARTSVG provides a considerable improvement of their methodology in their capability to accurately identify SVGs and provide scalability of the tool. To be accepted the authors will need to thoroughly revise the manuscript and edit figures to increase their quality.

Major Comments.

Which parameters (steps) were used for the different semi-pooling schema? This information is not clearly available in the manuscript. What is the reasoning behind choosing a Stouffer's method to derive a combined p-value? Showing the distribution of p-values across a set of samples would be beneficial. Also, the authors should describe which spatial autocorrelation measure is used in the HEARTSVG and explain how distinct is for example from Moran'I autocorrelation measure or other. What autocorrelation threshold was used for genes to be considered an SVG (high autocorrelation) versus a non-SVG (low autocorrelation)?

Also, how is the proposed method distinct from Squidpy by Palla et al. Nature Methods, 19, 171-178 (2022)?

Major Comments.

Please, consider revising Figure 1 and clearly illustrate inputs (used resources) and the outputs, metrics as it fails to show all the analysis steps to accurately quantify the final ST profiles.

The description of the auto-clustering method used in the manuscript is rather limited. I recommend adding additional details in the Supplementary Method section.

On Figure 2, the x-axis label should be labeled Percentage of SVGs. Additionally, the authors mentioned sample size, but it seems more like they are refereeing to the number of spots.

On Figure 3 panel C, revise the scale in which p-values are shown ($-\log_{10}(\text{p-value})$).

Consider revising manuscript and figures to consistently use FPR (false positive rate) and TPR (true positive rate).

On line 136: "Spot coordinates of each spot were generated using a Poisson random point process and gene expression counts were generated from the zero-inflated negative binomial distribution (ZINB)". In the revised version of the manuscript the authors used negative binomial (NB), zero-inflated negative binomial (ZINB), Poisson, and zero-inflated Poisson (ZIP) and this information is not clearly included in the result section.

On lines 322-325 and Figure 4. It seems quite odd the identification of "Rectum specific pathways" in cerebellum data. Please, confirm this is correct and share the GO ID and associated p-value. Additionally have the authors selected an expression threshold above which genes are considered to be expressed? And what is the minimum number of genes to be considered for a pathway to be enriched?

On Figure S10, 1236 genes were identified as common SVGs between different methods HEARTSVG, scGCO and SPARK-X, etc. It would be of interest to show the expression patterns of other genes that overlap across the different methods.

Minor Comments.

I recommend improving the quality of panels shown in Figure S10 and increase the font size. The same applies to Figure 4B labels.

On Figure S7 legend is not clear to what: "large pattern size" refers too. "Figure S7 Simulation results of hotspot pattern with low sparsity and large pattern size, (a) F1 score, (b) recall, (c) precision."

On Figure S19 legend what are: "MT" genes? "Figure S19 'MT' genes in primary colorectal cancer tissue and liver metastasis cancer tissue."

On Figures S23-S24 add which scale is being used to display the expression values.

Comments on the Quality of English Language

The paper is easy to follow and understand, but I would recommend editing by an English-speaking editor, in particular the Result section, to remove sentence duplications, and select the appropriate plural form, prepositions and articles and any other orthographic typos.

Consider uniformly using GO (vs Go) for Gene Ontology terms and uppercase gene symbols throughout the manuscript.

Line 140: Consider correcting this sentence: The $F1$ score was used to assess the performances of identification SVGs of HEARTSVG and three other SVG detection methods in identifying SVGs."

Line 144: Consider reframing this sentence: "SpatialDE, SPARK struggled to identify SVGs in sparse data possibly due to their adoption of the Gaussian data-generative model, which is not well-suited for gene spatial expression distribution." Why is it not well suited?

Line 147: Consider reframing this sentence: "scGCO exhibited many false negatives in due to its difficulty in accurately identifying candidate regions for SVGs in highly sparse datasets."

Lines 147-8: Remove the repetition from sentence: "On the low sparsity simulated data, scGCO showed improved $F1$ scores on simulated data with lower sparsity ($F1$ score=0.333).

Line 149: Consider re-writing this sentence to clarify what it means: "Identification performance was influenced by the pattern sizes of SVGs and sample sizes (Fig.2b, S1-3). What sample sizes were used? and what are pattern size? Described in Methods?"

Figure 1 legend: Edit inflaation to inflation

Dear Reviewers:

Thank you for the reviewers' comments concerning our manuscript entitled "HEARTSVG: a fast and accurate method for spatially variable gene identification in large-scale spatial transcriptomic data"(ID: NCOMMS-23-29155). Those comments are all valuable and very helpful for revising and improving our paper, as well as the important guiding significance to our research. We have carefully addressed your comments and made corrections. The detailed responses and the corresponding changes in the manuscript are shown in the following pages. Representative results have been highlighted in our response, and comprehensive tables and figures presenting all results are attached at the end of this reply. The contents are outlined below for your convenience.

We hope these revisions align with your expectations and improve the overall quality of the manuscript.

Contents

CONTENTS	1
RESPONSE TO REVIEWER #1:	2
RESPONSE TO REVIEWER #2	21
REFERENCES.....	52
FIGURES AND TABLES	53
1. RESULTS OF SIMULATIONS WITH MIXTURE NOISE.....	55
2. RESULTS OF NOISE-FREE SIMULATIONS	79
3. RESULTS OF SIMULATIONS WITH GAUSSIAN NOISE.....	83
4. RESULTS OF SIMULATIONS WITH NOISE OF 'RANDOMLY EXCHANGING EXPRESSION VALUES OF SELECTED NODES'	105

Response to Reviewer #1:

1. One of the key advantages claimed by the authors is scalability. However, their evaluation stopped at 50,000 cells. In comparison, the authors of scGCO evaluated the performance up to 1 million cells. Thus, additional evaluation is necessary. I would suggest the authors to evaluate both simulated data up to 1 million cells and real biological samples with millions of cells. Furthermore, in addition to running time, it is essential to profile the memory footprint of different methods to gain a holistic picture of these methods' scalability.

Thank you for your valuable comment on scalability. We appreciate the insights and understand the importance of extending our evaluation to larger datasets, both in simulated and real biological scenarios. We optimized the code using the 'dplyr' package for data manipulation and achieved reductions in runtime and memory usage. The updated version is available on GitHub. HEARTSVG computes faster in large-scale datasets than all other methods (13.45 mins for 1 million cells). Taking into consideration the comment from Reviewer 2, we have included Squidpy¹ for comparison. **Squidpy**¹ is a tool for the analysis and visualization of spatial molecular data and uses Moran's I statistics to detect spatially variable genes.).

Specifically, we expanded the assessment of simulated data (with 10,000 simulated genes) to scenarios with up to one million cells (**Figure 2 in the revised manuscript, Page 10**). In terms of runtime, HEARTSVG and SPARK-X outperformed other methods noticeably. For datasets with 500,000 and 1,000,000 cells, HEARTSVG demonstrated the fastest performance (6.83 mins and 13.45 mins), while SPARK-X required 8.82 mins and 16.43 mins. In contrast, scGCO demanded 7.58 hours and 21.83 hours, and Squidpy necessitated 2.47 days and 4.93 days. Regarding memory requirements, HEARTSVG, SPARK-X, and Squidpy exhibited similar memory needs, with values of 196.87 GB, 219.12 GB, and 182.17 GB for datasets of 500,000 cells, and 416 GB, 344 GB, and 367 GB for datasets of 1 million cells, respectively. Notably, scGCO had the highest memory demand, requiring 480.42 GB and 924.4 GB for the respective datasets of 500,000 and 1,000,000 cells.

Additionally, we have evaluated time consumption and memory requirements of four methods (HEARTSVG, SPARK-X, scGCO, and Squidpy) on real biological samples from mouse hypothalamus, comprising 1027,848 cells and 161 genes as show in the Figure S1 in this reply (**Figure S7, Page 14 in Supplementary**). HEARTSVG required 1.43 mins and 7.31 GB, scGCO needed a runtime of 112 mins and 14.72 GB, SPARK-X took 0.62 mins and 5.78 GB, and Squidpy took 3.73 mins, and 7.78 GB. We attempted to compare the performance of HEARTSVG, scGCO, SPARK-X and Squidpy on simulated data with 2 million cells (1000 simulated genes). HEARTSVG completed the computation in 4 to .5 minutes and 82.70 GB, Squidpy took 188.5 minutes and 343.7 GB, while SPARK-X and scGCO failed to scale to the dataset with 2 million cells. The new results are presented below and depicted in both the revised manuscript and the supplementary materials.

Figure 2e in the revised manuscript Plot shows the time consumption in $\log_{10}(\text{minutes})$ (y-axis) for each method analyzing on simulated data (10,000 simulated genes) with different sample sizes (x-axis). **Figure 2e** Plot shows memory requirements in GB (y-axis) for each method analyzing on simulated data (10,000 simulated genes) with different sample sizes (x-axis). Considering the limitation of scalability, we did not apply SpatialDE to datasets with samples exceeding 30,000 and did not apply SPARK to datasets with sample sizes exceeding 20,000. Considering the limitation of scalability, we did not apply SpatialDE to datasets with samples exceeding 30,000 and did not apply SPARK to datasets with sample sizes exceeding 20,000.

Figure S1 (Figure S7 in the supplementary) **a**, Bar diagram shows time consumption (y-axis) of four methods on mouse hypothalamus data (1027,848 cells and 161 genes) by MERFISH technology. **b**, Bar diagram shows memory requirements (y-axis) of four methods.

2. I am strong concerned about how the authors calculated performance metrics (such as AUC, F1 etc) on real biological samples. Unlike fields such as image classification in computer vision, where the class label of each image in the data set is authentic and known, our current understanding of SV genes is rather limited and for each biological sample, the true set of SV genes are unknown. The authors selected the top 500 overlaps between HEARTSVG and SPARK-X to form a set of true SVGs. This is a very biased approach and will strongly favor HEARTSVG and SPARK-X. Using genes selected by HEARTSVG to evaluate the performance of HEARTSVG is the same as evaluate a method's performance using the training data, it is fundamentally flawed.

Thank you for your comment. You are right. It is unfair to form a set of true SVGs by selecting the top 500 overlaps between HEARTSVG and SPARK-X. To avoid bias, we now selected the top 500 overlaps from the results of all six methods as the set of true SVGs and generated mixture simulations (The schematic of simulated data with mixture noise is shown in **Figure S2 in this reply**). Based on the top 500 overlaps from the results of all six methods, we constructed four artificial genesets as benchmarks to assess the accuracy and robustness of all methods. Among these four genesets, HEARTSVG exhibited the highest TPRs (average TPR = 0.987) (**Figure S3-S6, listed below**). The other five methods exhibited lower TPRs on the genesets with noise, especially in non-SVGs with noise. In addition, we generated 1000 noise-free simulated SVGs and employed the same approach to create non-SVGs, SVGs with noise, and non-SVGs with noise. According to our simulation results, HEARTSVG performed the highest F1 scores (average

F1 score = 0.931) and TPRs of each gene set (average TPR = 0.901) than the other methods across datasets with different spatial patterns (**Figure S23, Page23 in this reply**) In contrast, scGCO, SpatialDE, SPARK, and SPARK-X had lower precision values, with both false negatives in the SVGs with noise and false positives in the non-SVGs with noise. These findings suggest that HEARTSVG is a robust and effective method for detecting SVGs in noisy datasets.

Regarding the calculation of the indices (AUC, F1, etc.) in the section of “Application to 10X Visium colorectal cancer data”, we chose two widely used genesets, single-cell level common gene modules linked with tumor microenvironments^{2,3} and consensus molecular markers of colorectal cancer subtypes^{4,5}, as reference standards for true SVGs. The first geneset, single-cell level common gene modules linked with tumor microenvironments has been widely applied in pan-cancer studies of tumor microenvironments⁶⁻¹¹. Zhang,et.al³ used these gene modules to study the tumor micro-environment of liver cancer. The second geneset, consensus molecular markers of colorectal cancer subtypes, has found broad applications CRC patient classification¹²⁻¹⁸ and has been validated by various studies^{12,15}.

The representative results (**Figure S3-S6**) are listed below, and all the results are attached at the end of this response (**Figure S23 in this reply**). We have also updated the manuscript and the Supplementary accordingly (**Figure 3 , Page 15 the manuscript, Figure S13, S39-S61, Page 24,48-70 on Supplementary**).

Figure S2 The schematic of simulated data with mixture noise. Set 1 represents the true SVGs set, derived by selecting the top 500 overlaps from the results of all six methods. Set 3 is the non-SVGs set, created by randomly rearranging gene expressions within Set 1. Set 2 and Set 4 are generated by using a mixture of SVGs and non-SVGs to simulate data with noise.

Figure S3 a, Visualization of SVG, SVG with noise, non-SVG, non-SVG with noise. b, The heatmap shows the comparison of TPR values among four gene sets on three colorectal cancer ST datasets and three corresponding liver metastasis ST datasets. Set1: SVGs. Set2: SVGs with noise. Set3: non-SVGs. Set4: non-SVGs with noise.

Figure S4 Simulation results of SVGs identification using simulated data with mixture noise. **a**, Visualization of Pattern: Hotspot with mixture noise. **b-e**, Simulation results of six different methods (HEARTSVG, scGCO, SPARK, SPARK-X, Squidpy) on simulated data generated by four distinct distributions (ZINB, ZIP, NB, Poisson). The bar diagram (sub-panel (1)) shows F1 scores, TPRs, precisions, and FPRs. The heatmap (sub-panel (2)) depicts the comparison of TPR values among six genesets.

Figure S5 Simulation results of SVGs identification using simulated data with mixture noise. **a**, Visualization of Pattern: Streak with mixture noise. **b-e**, Simulation results of six different methods (HEARTSVG, scGCO, SPARK, SPARK-X, Squidpy) on simulated data generated by four distinct distributions (ZINB, ZIP, NB, Poisson). The bar diagram (sub-panel (1)) shows F1 scores, TPRs, precisions, and FPRs. The heatmap (sub-panel (2)) depicts the comparison of TPR values among six genesets.

Figure S6 Simulation results of SVGs identification using simulated data with mixture noise. **a**, Visualization of Pattern: Big triangles with mixture noise. **b-e**, Simulation results of six different methods (HEARTSVG, scGCO, SPARK, SPARK-X, Squidpy) on simulated data generated by four distinct distributions (ZINB, ZIP, NB, Poisson). The bar diagram (sub-panel (1)) shows F1 scores, TPRs, precisions, and FPRs. The heatmap (sub-panel (2)) depicts the comparison of TPR values among six genesets.

3. Given that the true set of SV genes are generally unknown in real biological samples, the evaluation based on simulated data is critical. Unfortunately, the authors only simulate three simple patterns. Additional patterns are necessary to gain an unbiased picture. The authors should demonstrate that the patterns are comprehensive enough to capture major SV trends. Furthermore, because all methods will perform extremely well on noise-free simulated data, varying levels of noise should be added to evaluate the robustness of different methods.

Thank you very much for your comments. Following your comments, we have extended our simulations with 19 additional spatial patterns that cover various scenarios of gene expression changes. We have also assessed the robustness of different methods in the presence of varying levels of noise.

In noise-free simulated data, HEARTSVG showed higher F1 scores (average F1 score=0.948) than other methods across 22 different spatial patterns and varying numbers of cells (**Figure S7**). Regarding the simulations with noises, we generated simulated data with three different noise generation approaches:

- a) Gaussian noise
- b) The noise of 'Randomly Exchanging Expression Values of Selected Nodes'
- c) Mixture noise

We added **Gaussian noise** to simulated data with four different distributions and 22 spatial patterns, following a similar approach as scGCO. We applied six different levels of noise to 10,000 simulated genes (both SVGs and non-SVGs) and created six datasets with noises. HEARTSVG showed the best performance (average F1 score = 0.849 at Gaussian noise strength of 0.3) among the four methods (HEARTSVG, scGCO, SPARK-X and Squidpy), and was the most robust to increasing Gaussian noise strength. SPARK-X was the second-best method in terms of these indices. Squidpy had a significant drop in Precision and F1 scores when the Gaussian noise strength was higher than 0.2, indicating that it had many false positives in noisy data. For simulated data with **noise of 'Randomly Exchanging Expression Values of Selected Nodes'**, we followed the procedures described by scGCO to generate noisy datasets. We randomly selected some spots of the SVG's marked area and non-marked area and then exchanged their expression values. All methods had a substantial decline in F1 score when the percentage of

randomly exchanged cells increased. HEARTSVG still had the highest accuracy (average F1 score = 0.618 at percentages of exchanging cells of 30%) and the lowest false positive rates (average FPR < 0.001 at percentages of exchanging cells of 30%) among the methods. For simulations with **mixture noise**, we have described the noise data generation methods and simulation results in detail in our reply to **comment 2**. The representative results (**Figure S3-S6**) are listed below, and all the results are attached at the end of this response (**Figure S24-S66, Page83 to Page122 in this reply**). We have also updated the manuscript and the Supplementary accordingly (**Figure 2, Page 10 on the manuscript, Figure S62-S105, Page 71-117 on the Supplementary**).

Figure S7 a, Visualization of 23 representative spatial expression patterns: Hotspot, Streak, Gradient, Ring, Nested rings, Streaks, Curve, Rectangles, Big triangles, Big circles, Big squares, Small triangles, Small circles, Small squares, Big circles II, Small triangles II, Pattern I, Pattern II, Pattern III, Irreg pat I, Irreg pat II, Irreg pat III. b-e, F1 score plots, TPR plots, Precision plots, and FPR plots compare the index values (y-axis) of HEARTSVG (red), scGCO (orange), SpatialIDE

(yellow), SPARK (blue), SPARK-X (green), and Squidpy (purple) in simulation data. The comparison is based on varying sample sizes (x-axis) at an adjusted p-value cutoff of 0.05. Each plot corresponds to the left spatial patterns in sub-figure a.

Figure S8 Simulation results for identifying SVGs using simulated data with Gaussian noise. **a**, Visualization of Pattern: Hotspot with varying level of Gaussian noise. **b-e**, Simulation results of four different methods (HEARTSVG, scGCO, SPARK-X and Squidpy) on simulated data generated by four distinct distributions (ZINB, ZIP, NB, Poisson). F1 score plots, TPR plots,

Precision plots, and FPR plots compare the index values (y-axis) of HEARTSVG (red), scGCO (orange), SPARK-X (green), and Squidpy (purple) across varying levels of Gaussian noise strength (x-axis). Index values were calculated at the adjusted p-value cutoff of 0.05.

Figure S9 Simulation results for identifying SVGs using simulated data with Gaussian noise. **a**, Visualization of Pattern: Streaks with varying level of Gaussian noise. **b-e**, Simulation results of four different methods (HEARTSVG, scGCO, SPARK-X and Squidpy) on simulated data generated by four distinct distributions (ZINB, ZIP, NB, Poisson). F1 score plots, TPR plots, Precision plots, and FPR plots compare the index values (y-axis) of HEARTSVG (red), scGCO (orange), SPARK-X (green), and Squidpy (purple) across varying levels of Gaussian noise strength (x-axis). Index values were calculated at the adjusted p-value cutoff of 0.05.

(orange), SPARK-X (green), and Squidpy (purple) across varying levels of Gaussian noise strength (x-axis). Index values were calculated at the adjusted p-value cutoff of 0.05.

Figure S10 Simulation results for identifying SVGs using simulated data with Gaussian noise. **a**, Visualization of Pattern: Big circles with varying level of Gaussian noise. **b-e**, Simulation results of four different methods (HEARTSVG, scGCO, SPARK-X and Squidpy) on simulated data generated by four distinct distributions (ZINB, ZIP, NB, Poisson). F1 score plots, TPR plots, Precision plots, and FPR plots compare the index values (y-axis) of HEARTSVG (red), scGCO

(orange), SPARK-X (green), and Squidpy (purple) across varying levels of Gaussian noise strength (x-axis). Index values were calculated at the adjusted p-value cutoff of 0.05.

Figure S11 Simulation results for identifying SVGs using simulated data with noise of 'Randomly Exchanging Expression Values of Selected Nodes.' **a**, visualization of Pattern: Streaks with different percentage of cells random exchanges (%). **b-e**, Simulation results of four different methods (HEARTSVG, scGCO, SPARK-X and Squidpy) on simulated data generated by four distinct distributions (ZINB, ZIP, NB, Poisson). F1 score plots, TPR plots, Precision plots, and FPR plots compare the index values (y-axis) of HEARTSVG (red), scGCO (orange), SPARK-X (green), and Squidpy (purple) across different percentage of cells random exchanges (x-axis). Index values were calculated at the adjusted p-value cutoff of 0.05.

Figure S12 Simulation results for identifying SVGs using simulated data with noise of 'Randomly Exchanging Expression Values of Selected Nodes.' **a**, Visualization of Pattern: Big circles with different percentage of cells random exchanges (%). **b-e**, Simulation results of four different methods (HEARTSVG, scGCO, SPARK-X and Squidpy) on simulated data generated by four distinct distributions (ZINB, ZIP, NB, Poisson). F1 score plots, TPR plots, Precision plots, and FPR plots compare the index values (y-axis) of HEARTSVG (red), scGCO (orange), SPARK-X (green), and Squidpy (purple) across different percentage of cells random exchanges (x-axis). Index values were calculated at the adjusted p-value cutoff of 0.05.

Response to Reviewer #2

The manuscript entitled, “HEARTSVG: a fast and accurate method for spatially variable gene identification in large-scale spatial transcriptomic data” by Yuan et al. presents an interesting new methodology to identify spatial variable genes (SVGs). This work aims to identify SVGs without using pre-determined assumptions about gene expression spatial distribution. The authors focus on identifying non-SVGs genes to define a background spatial expression and then infer SVGs using both semi-pooling and autocorrelation. The authors have analyzed a significant number of datasets (n=12) across different sequencing technologies and investigated data sparsity, distinct spatial patterns, number of spots, SVGs abundance and ZINB parameters on simulated data. Additionally, in the revised version the authors include a comparison with other SVGs detection methods such as scGCO, spatialDE, SPARK and SPARK-X. The authors show that HEARTSVG provides a considerable improvement of their methodology in their capability to accurately identify SVGs and provide scalability of the tool. To be accepted the authors will need to thoroughly revise the manuscript and edit figures to increase their quality.

Thank you very much for the valuable feedback. We will thoroughly revise the manuscript and improve the quality of the figures to better meet your requirements and enhance the overall quality of the paper. Your suggestions are greatly appreciated.

Major Comments.

1. Which parameters (steps) were used for the different semi-pooling schema? This information is not clearly available in the manuscript.

Thank you very much for comments. Follow your comments, we added more information about the semi-pooling process to the Supplementary (**Figure S10, Page 19 in the Supplementary**) and presented below.

The semi-pooling process needs two parameters: direction parameter and feature map parameter. For each gene, the spatial expression data was averaged according to the given direction and step parameters, and the mean value was used as the new marginal expression value (**Figure S13 in this reply**). HEARTSVG used four sets of different semi-pooling parameters, which are:

- 1) Direction: row direction, feature map: $1 \times n_{row}$;
- 2) Direction: row direction, step: feature map: $1 \times [\ln (n_{row})]$;
- 3) Direction: column direction, feature map: $1 \times n_{col}$;
- 4) Direction: column direction, step: feature map: $1 \times [\ln (n_{col})]$

where n_{row} is the number of rows in the spatial transcriptome data, n_{col} is the number of columns in the spatial transcriptome data, and $[\cdot]$ means rounding to the nearest integer.

Figure S13 (Figure S10 in the Supplementary) Illustration of the semi-pooling process. **a**, Spatial expression data schematic diagram. We assume the expression value of the gene in each cell is the order value of the column it belongs to. **c-f**, The new marginal expression series based on four sets of parameters.

2. What is the reasoning behind choosing a Stouffer's method to derive a combined p-value? Showing the distribution of p-values across a set of samples would be beneficial.

Thank you for your comment. We chose Stouffer's method because it is a classic p-value combination method that tends to pick up consistent effects and is more robust in the presence of rare outliers¹⁹. We also demonstrated the rationality of choosing Stouffer's method through the following simulation. In the 10X Visium colorectal cancer data, we selected the top 500 overlaps from the results of all six methods as SVGs and created non-SVGs by randomly rearranging gene expressions of SVGs. Then, we investigated the distributions of p-values obtained from the autocorrelation test applied to the marginal expression time series (**Figure S14 a-d**). Moreover, we compared two common combination methods: Fisher's combination method and Stouffer's method. Fisher's method ($Z_{Fisher} = -2 \sum_{i=1}^4 \ln(p_i) \sim \chi^2(df = 8)$) directly processed the p-values, while Stouffer's method ($Z_{Stouffer} = \sum_{i=1}^4 \frac{\Phi^{-1}(1-p_i)}{\sqrt{4}} \sim N(0,1)$) first transformed the p-values into Z-scores, and then combined them. We calculated the combined p-value statistics of the two methods, plotted the density histograms (**Figure S14e, S14f in this reply**), and compared them with their theoretical distribution density curves (the black solid lines in the figure), as shown in **Figure S14 (in this reply)**. It can be seen that the distribution of Stouffer's statistic was closer to the theoretical distribution, indicating that choosing Stouffer's method was reasonable. We added more information about the Stouffer's method to the Supplementary (**Figure S9, Page 18 in the Supplementary**) and presented below.

Figure S14 a–d, (Figure S9 in the Supplementary) The density distributions of p-values obtained from the autocorrelation test applied to the marginal expression time series. **e,** The density distributions of Stouffer's statistic. The black solid line shows the theoretical distribution ($N(0,1)$) of the Stouffer's statistic. **f,** The density distributions of Fisher's statistic. The black solid line shows the theoretical distribution ($\chi^2(df = 8)$) of the Fisher's statistic.

3. Also, the authors should describe which spatial autocorrelation measure is used in the HEARTSVG and explain how distinct is for example from Moran's I autocorrelation measure or other. What autocorrelation threshold was used for genes to be considered an SVG (high autocorrelation) versus a non-SVG (low autocorrelation)?

Also, how is the proposed method distinct from Squidpy by Palla et al. Nature Methods, 19, 171-178 (2022)?

Thank you for your comments. Because Squidpy uses Moran's I to identify SVGs, we will respond to these two comments together. Both HEARTSVG and Moran's I measure the spatial correlation between gene expression values and their neighboring values, but they differ significantly in their specific approaches.

1) How to measure the spatial correlation?

Moran's I is a statistical measure of spatial autocorrelation, assessing whether gene expression at spatial locations correlates with that at neighboring locations. The formula for Moran's I is as follows: $Moran's\ I = \frac{N \sum_{i=1}^N \sum_{j=1}^N w_{ij} (x_i - \bar{x})(x_j - \bar{x})}{W \sum_{i=1}^N (x_i - \bar{x})^2}$, where, N is the number of cells/spatial spots, x_i, x_j are the gene expressions of the cell i and j , and \bar{x} is the mean expression of all cells, $W = \sum \sum w_{ij}$ is the sum of all w_{ij} , w_{ij} is the spatial weight of between the cell i and the cell j . The calculation of Moran's I depends on a predefined spatial weight matrix (if $w_{ij} = 0$, thus $w_{ij}(x_i - \bar{x})(x_j - \bar{x}) = 0$, 0 suggests spatial randomness). Thus, the reliability of the predefined spatial weight matrix is questionable due to the diversity and complexity of real biological samples.

HEARTSVG uses the autocorrelation of the marginal expressions as a measure of the spatial correlation, rather than directly calculating the spatial correlation of gene expressions. We measure the spatial correlation of gene expression levels and neighboring expressions by calculating autocorrelations of the marginal expression series (r): $\hat{\rho}_k = \frac{\sum_{t=1}^{T-k} (r_t - \bar{r})(r_{t+k} - \bar{r})}{\sum_{t=1}^T (r_t - \bar{r})^2}$, k is the interval between two marginal expressions. If $\hat{\rho}_k = 0$, it means no correlation between gene expression and neighboring expressions; otherwise, there is a correlation. HEARTSVG assumes that the two-dimensional distribution of non-SVGs is random, which is a weak assumption. If a

gene is random in two dimensions, its marginal distribution should also be random (The derivation process is in the Section of Methods in the Supplementary).

2) How to identify SVGs?

The values of Moran's I range from -1 to 1. Positive values indicate positive spatial autocorrelation (similarity in expressions at adjacent cells). Negative values indicate negative spatial autocorrelation (dissimilarity in expressions at neighboring cells). 0 suggests spatial randomness. However, there is no fixed threshold for Moran's I to determine whether there is spatial correlation. To determine whether the calculated Moran's I statistic is significant, the Monte Carlo simulation is used to test whether there is spatial autocorrelation in the data. It shuffles the gene expression values randomly or generates new random gene expression values, and recalculates the Moran's I statistic multiple times, thus establishing a significance level based on the random distribution. The p-value is computed by comparing the originally calculated Moran's I statistic with the Monte Carlo simulation results. The formula for the test statistic is: $Z = \frac{I - E(I)}{\sqrt{Var(I)}} \sim N(0,1)$. If the p-value is very small, it indicates that there is significant spatial autocorrelation in the data. If the p-value is very large, it indicates that there is no significant spatial autocorrelation in the data. The more times you shuffle, the higher the reliability of the results, but the more time-consuming it is.

HEARTSVG does not use a fixed autocorrelation threshold to determine SVGs. It tests whether multiple autocorrelations of multiple marginal expressions against zero then uses the p-values of the statistic as the threshold to determine whether a gene is an SVG. For each marginal expression series, HEARTSVG tests whether multiple autocorrelations are zero ($H_0: \hat{\rho}_1 = \dots = \hat{\rho}_m = 0$), the statistic is $Q_m = T \sum_{k=1}^m \hat{\rho}_k^2$, T is the length of the marginal expression series, m is the max interval of the between two marginal expressions. m is an empirical parameter and is usually set to: $m = \ln(T)$. Finally, Stouffer's method was used to test whether multiple marginal expressions are random. This procedure makes our method more robust and can capture a wider range of patterns.

HEARTSVG and Squidpy use different measures of spatial autocorrelations. HEARTSVG uses the autocorrelation of the marginal expression series as a measure of the spatial correlations. The

Squidpy uses Moran's I and calculates p-value based on standard normal approximation from 100 random permutations, which is very time-consuming. Therefore, Squidpy only evaluates the subset of highly variable genes. We added the Squidpy as a comparison in our revised manuscript. In the noise-free simulations, Squidpy can effectively identify SVGs, but in the noisy simulations, Squidpy's ability to identify SVGs will decline significantly (**Figure S24-S66, Page83 in this reply**). **Figure S15** shows some genes that were not identified by HEARTSVG, but not by Squidpy in the 10X Visium colorectal cancer data"

Figure S15 SVGs selected by HEARTSVG, but not by Squidpy.

4. Please, consider revising Figure 1 and clearly illustrate inputs (used resources) and the outputs, metrics as it fails to show all the analysis steps to accurately quantify the final ST profiles.

Thank you for your comments. We have modified the Figure 1 in the manuscript to show all analysis steps. The revised figure is shown below.

Figure 1 (in the manuscript) Schematic representation of the HEARTSVG. a, HEARTSVG utilizes the semi-pooling process to convert the spatial gene expression into marginal expressions (r) and calculates autocorrelations (ρ) of marginal expressions. HEARTSVG calculates the sum of the squared autocorrelations ($\hat{Q}_m = T \sum_{i=1}^m \hat{\rho}_i^2$) for each marginal expression series and obtains a p-value by testing \hat{Q}_m . All these p-values are combined into a final p-value through the Stouffer combination ($z_s = \sum_{i=1}^4 \frac{z_i}{\sqrt{4}} \sim N(0,1)$, final p – value = $2(1 - \Phi(|z_s|))$). HEARTSVG distinguishes between SVGs and non-SVGs by the final p-value. **b**, The autocorrelations (ρ) of marginal expressions (r) for the SVG and non-SVG exhibit different level scales. Representative autocorrelation estimator plots are plotted below the corresponding marginal expression plots for the SVG and non-SVG. The color depth represents the magnitude of autocorrelation.

5. The description of the auto-clustering method used in the manuscript is rather limited. I recommend adding additional details in the Supplementary Method section.

Thank you for the comment. We have added details of the auto-clustering method in the Supplementary Method section (**Page 20 in the Supplementary**). The auto-clustering module uses the hierarchical clustering algorithm to cluster SVGs into different spatial expression patterns based on their expression and location similarity. The steps of the auto-clustering module are as follows:

Step 1: Calculate the similarity between each pair of genes based on spatial expression and generation of the distance matrix.

We calculated the Euclidean distance between each pair of SVGs based on their expression and positions, serving as a measure of similarity among SVGs.

Step 2: Construct a clustering tree based on the distance matrix using the complete linkage criterion. The resulting hierarchy of clusters can be visualized as a dendrogram.

Initially, each gene is assigned to its own cluster. Then, in each iteration, the two closest clusters are merged into a new cluster using the complete linkage method, which is a method that determines the distance between two clusters by the maximum distance between any two data points from different clusters, to ensure significant dissimilarity between clusters. This process repeats until all data points are eventually merged into a single cluster (a clustering tree).

Step 3: Determination of the final clustering results by cutting the dendrogram at a certain height or distance threshold. The cutting height is chosen using the maximum breakpoint of all breakpoints selected by the Yamamoto test.

After forming the hierarchical structure, we arrange the heights in ascending order and utilize the Yamamoto test to identify breakpoints at different height thresholds. The maximum breakpoint value is employed as the cutting height to determine the final number of clusters. The Yamamoto test employs the rolling window method to calculate the difference of heights

before and after each point. If the difference exceeds the threshold, the i -th point is considered a breakpoint. The details were listed below.

We had a serial: $h_1, \dots, h_i, \dots, h_N$, set window width: $2 * n_{period}$. Then we calculated the difference of heights before and after the i -th point: $D(h_i) = \frac{|\text{mean}(H_{i\text{-before}}) - \text{mean}(H_{i\text{-after}})|}{\text{sd}(H_{i\text{-before}}) + \text{sd}(H_{i\text{-after}})}$, where, $H_{i\text{-before}} = (h_{i-1}, \dots, h_{i-n_{period}})$, $H_{i\text{-after}} = (h_{i+1}, \dots, h_{i+n_{period}})$, $\text{mean}(\cdot)$ and $\text{sd}(\cdot)$ are the mean and standard error function, the $\text{threshold} = \frac{t_{1-\alpha-(df=n_{period}-1)}}{\sqrt{n_{period}}}$. If $D(h_i) > \text{threshold}$, the i -th point is a breakpoint. We identified all breakpoints of the serial and chose the max breakpoint as the cutting height of the cluster tree.

The rationale behind the Yamamoto test lies in the characteristics of hierarchical clustering. SVG clusters with similar spatial expression patterns have smaller distances and form clusters at lower heights in the tree structure, while SVG clusters with different patterns have larger distances and form clusters at higher heights in the tree structure. Thus, when merging clusters of SVGs with different patterns, there is a noticeable jump in height.

6. On Figure 2, the x-axis label should be labeled Percentage of SVGs. Additionally, the authors mentioned sample size, but it seems more like they are referring to the number of spots.

Thank you for your comments. Following your comment, we have updated the x-axis label to "Percentage of SVGs" to better align with the content of **Figure 2d (Page 10 in the manuscript)**.

You are right, it is more accurate to refer to the number of spots. We have changed the x-axis label to "Number of Cells (with 10,000 simulated genes) " of **Figure 2a, 2e (Page 10 in the manuscript)**. The revised figure is shown below.

Figure 2 (in the manuscript) **Simulation results show that HEARTSVG has high accuracy and good scalability and computational efficiency.** **a**, Visualization of seven representative spatial expression patterns: hotspot, streak, gradient, rind, streaks, pattern III, and irregular pattern II. F_1 score plots compare the accuracy (y-axis) of HEARTSVG (red), SpatialDE (yellow), SPARK (blue), SPARK-X (green), scGCO (orange) and Squidpy (purple) in simulation data with varying numbers of cells (x-axis) at an adjusted p-value cutoff of 0.05. Each plot corresponds to the upper spatial patterns. Simulations were conducted under high sparsity (*zero-inflation parameter* = 0.941, *size* = 0.5, *mu* = 0.5) and a moderate percentage of SVGs

(10%, 1,000 SVGs). The F_1 score for each method in each simulation scenario represents the average of fifty replications. For easy computation, we did not apply SpatialDE and SPARK to datasets with sample sizes of more than 10,000 in these simulations. **b**, Representative simulation results with Gaussian noise. F_1 score plots compare four different methods (HEARTSVG, scGCO, SPARK-X and Squidpy) across varying levels of Gaussian noise strength (x-axis). Index values were calculated at the adjusted p-value cutoff of 0.05. Simulations were conducted 3000 cells (with 10,000 simulated genes) under high sparsity (*zero – inflation parameter* = 0.941, *size* = 0.5, *mu* = 0.5). **c**, Representative simulation results with increasing percentage of randomly exchanged cells. F_1 score plots compare four different methods (HEARTSVG, scGCO, SPARK-X and Squidpy) across varying percentage of randomly exchanged expression values of cells (x-axis). Index values were calculated at the adjusted p-value cutoff of 0.05. Simulations were conducted 3000 cells (with 10,000 simulated genes) under high sparsity (*zero – inflation parameter* = 0.941, *size* = 0.5, *mu* = 0.5). **d**, False positive rate (FPR, y-axis) boxplots of HEARTSVG (red) and SPARK-X (green) in simulation data of hotspot pattern with different percentages of SVGs(x-axis) at an adjusted p-value cutoff of 0.05. Each scenario has 10,000 simulated genes, including both SVGs and non-SVGs. Simulations were conducted under high sparsity (*zero – inflation parameter* = 0.941, *size* = 0.5, *mu* = 0.5) and moderate number of cells(n=5,000). In each boxplot, the lower hinge, upper hinge, and center line represent the 25th percentile (first quartile), 75th percentile (third quartile), and 50th percentile (median value), respectively. Whiskers extend no further than ± 1.5 times the inter-quartile range. Data beyond the end of the whiskers are considered outliers and are plotted individually. **e**, Plot shows time consumption in $\log_{10}(\text{minutes})$ (y-axis) and memory usage in GB (y-axis) of each method for analyzing 10,000 genes with different numbers of cells (x-axis). Considering the limitation of scalability, we did not apply SpatialDE to datasets with sample sizes exceeding 30,000 and did not apply SPARK to datasets with sample sizes exceeding 20,000. Fig.2b and Fig.2d have the common figure legend.

- On Figure 3 panel C, revise the scale in which p-values are shown ($-\log_{10}(\text{p-value})$).
- Consider revising manuscript and figures to consistently use FPR (false positive rate) and TPR (true positive rate).

Thank you for the comment. Following your comments 7-8, we changed the Figure 3. The new figure is presented below and shown in the revised manuscript.

Figure 3 (in the manuscript) HEARTSVG identifies tumor related SVGs and predicts spatial functional domains with distinct biological functions in the 10X Visium colorectal cancer data. **a**, Original hematoxylin and eosin stained (H&E) tissue image (left) and results of unsupervised spatial clustering (right). The red-circled areas in the HE image represent the tumor

regions. **b**, The heatmap depicts the comparison of recall values among four genesets. Set 1 represents the true SVGs set, derived by selecting the top 500 overlaps from the results of all six methods. Set 4 corresponds to the non-SVGs set, obtained by randomly rearranging the gene expressions within Set 1. Set 2 and Set 3 are generated by introducing noise to the true SVGs set and non-SVGs set, respectively. (More details of four genesets in **Supplementary**). **c**, The bubble plot illustrates the results of KEGG pathway enrichment analysis for 19 tumor-related pathways (x-axis) across different methods. Each bubble represents a pathway, and its size corresponds to the overlap gene size of the pathway and SVGs detected by each method. The x-axis and y-axis of the plot represent different methods and the significance ($-\log_{10}(\text{p-value})$). **d**, The ROC curves illustrate the TPR and FPR of six different methods when using common gene modules of tumor microenvironments (left) and consensus molecular markers of colorectal cancer subtypes (right) and as gold standards for true SVGs. AUC is the area under the ROC curve. **e, f**, HEARTSVG predicts four spatial domains based on SVGs and graphed the average expression of SVGs in each spatial domain. Three tumor-related spatial domains (spatial domains 1, 2, and 3) exhibit different spatial average expression patterns. **f**, Representative SVGs correspond to the four predicted spatial domains in Fig. **3e, 3g**, Enrichment analysis corresponds to the four predicted spatial domains in Fig. **3e**. The length of bars represents the enrichment using $-\log_{10}(\text{p-value})$.

9. On line 136: “Spot coordinates of each spot were generated using a Poisson random point process and gene expression counts were generated from the zero-inflated negative binomial distribution (ZINB)”. In the revised version of the manuscript the authors used negative binomial (NB), zero-inflated negative binomial (ZINB), Poisson, and zero-inflated Poisson (ZIP) and this information is not clearly included in the result section.

Thank you for the comment. Following your comment, we added more simulated data generating by negative binomial (NB), zero-inflated negative binomial (ZINB), Poisson, and zero-inflated Poisson (ZIP). The new results show that HEARTSVG can effectively identify SVGs (average F1 scores =0.958 in noise-free data with 3000 simulated cells) under different distributions and spatial patterns. These results demonstrate the robustness and flexibility of

HEARTSVG in dealing with different types of spatial transcriptomic data. The new results are presented below and shown in the revised manuscript and supplementary (**Figure 2, Page 10 on the manuscript, Figure S62-S105, Page 71-117 on the Supplementary**). More details about the distribution parameters are attached at the end of this response (**Table S2, Page 52 on this reply**).

Figure S16 a, Visualization of 22 spatial expression patterns: Hotspot, Streak, Gradient, Ring, Nested rings, Streaks, Curve, Rectangles, Big triangles, Big circles, Big squares, Small triangles, Small circles, Small squares, Big circles II, Small triangles II, Pattern I, Pattern II, Pattern III, Irreg pat I, Irreg pat II, Irreg pat III. **b**, F1 scores (y-axis) of HEARTSVG (red), scGCO (orange), SpatialDE (yellow), SPARK (blue), SPARK-X (green), and Squidpy (purple) in simulation data from four different distributions.

10. On lines 322-325 and Figure 4. It seems quite odd the identification of “Rectum specific pathways” in cerebellum data. Please, confirm this is correct and share the GO ID and associated p-value. Additionally have the authors selected an expression threshold above which genes are considered to be expressed? And what is the minimum number of genes to be considered for a pathway to be enriched?

Thank you for your comments. We have verified our results again. We used g:Profiler²¹ to do the enrichment analysis for Figure 4, and compared it with HPA (Human Protein Atlas) database. The HPA database is often used for enrichment analysis of tissue-specificity. **Table 1** lists the HPA ID, tissue, p-value, intersection size, and TPR for the “Rectum specific pathways”. We did not use an expression threshold. We enriched the SVGs chosen by each method (i.e., the genes with adjusted p-value ≤ 0.05 in each method). We required at least 5 genes for a pathway to be enriched and used an adjusted p-value < 0.01 as the significance threshold.

Table 1 Rectum and endometrium specific pathways

Method	HPA: ID	tissue	p_value	Intersection size	term_size	TPR
SPARK	HPA:0400242	rectum	1.22E-10	28	192	0.146
SPARK	HPA:0400241	rectum	8.01E-09	30	260	0.115
SPARK	HPA:0400243	rectum	8.46E-09	20	113	0.177
SPARK-X	HPA:0400242	rectum	1.25E-06	25	192	0.13
SPARK-X	HPA:0400241	rectum	2.61E-06	29	260	0.112
SPARK-X	HPA:0400243	rectum	8.50E-06	18	113	0.159
SpatialDE	HPA:0400241	rectum	0.001901336	34	260	0.131
Squidpy	HPA:0641531	endometrium	0.006927202	14	49	0.286

11. On Figure S10, 1236 genes were identified as common SVGs between different methods HEARTSVG, scGCO and SPARK-X, etc. It would be of interest to show the expression patterns of other genes that overlap across the different methods.

Thank you for your valuable and useful comments. We have performed exploratory analysis and visualization on this part of the data. Since many genes were identified as SVGs in this dataset, we focused on the Top 5000 SVG identified by any of the methods (except scGCO). We found that some SVGs shared by HEARTSVG, Squidpy, and SPARK-X (excluding commend SVGs) were not significant or had large adjusted p-values in the SPARK and SpatialDE results (**Figure S17**). Overlaps between SPARK-X and Squidpy (excluding commend SVG) exhibited patterns like the gradient spatial expression pattern (**Figure S19**). This is consistent with our simulation results, where SPARK-X and Squidpy had high F1 scores on simulation data with gradient patterns (**Figure S23, Page 79 in this response**). However, some SVGs shared by SPARK and SpatialDE (excluding commend SVGs) do not have clear spatial expression patterns (**Figure S18**). The new results are presented below.

Figure S17 SVGs selected by HEARTSVG, Squidpy, and SPARK-X, but not by SPARK, SpatialDE, and scGCO.

Figure S18 SVGs selected by SPARK and SpatialDE, but not by the other 4 methods.

Figure S19 SVGs selected by Squidpy, and SPARK-X, but not by other 4 methods.

Minor Comments.

12. I recommend improving the quality of panels shown in Figure S10 and increase the font size. The same applies to Figure 4B labels.

Thank you for your comments. We have improved the quality of the panels in **original Figure S10 (Figure S11 in the Supplementary)** and increased the font size. We have also enlarged the labels in **Figure 4b**. The revised figures are shown below.

Original Figure S10 (Figure S11 in the Supplementary) scGCO missed SVGs (RPS29, ARPC3, GAS5) with clear spatial expression patterns comparing with other methods. **a**, Visualizations of spatial expressions of gene RPS20, RPS29, ARPC3, and GAS5. **b**, Venn diagrams of SVGs in the colorectal cancer data identified by HEARTSVG, SpatialDE, SPARK, SPARK-X, scGCO, and Squidpy. **c**, Marginal expression plots of gene RPS20, and GAS5 by HEARTSVG. **d**, Visualizations of graph cuts by scGCO with different initial smooth factor of gene RPS20, and GAS5 by HEARTSVG

Figure 4 (in the manuscript) HEARTSVG detects biologically meaningful SVGs in the Slide-seqV2 cerebellum data. **a**, Visualization of unsupervised spatial clustering results. **b**, Visualization of cerebellum layer annotations. **c**, Visualizations of spatial expressions and adjusted p-values of representative SVGs of specified cell types detected by HEARTSVG. Car8 (adjusted p=1.21 e-10) in Purkinje cells, Cbln1 (adjusted p=0) in granule cells and Mbp (adjusted p=0) in oligodendrocytes. **d**, The tissue-specific enrichment analysis results for each method, where the x-axis represents different tissues, and the y-axis represents the percentage of tissue-

specific pathways. Each panel corresponds to each method. **e**, Heatmap of SVGs expressions in Molecular layer, Purkinje layer, and Granule layer.

13. On Figure S7 legend is not clear to what: “large pattern size” refers too. “Figure S7 Simulation results of hotspot pattern with low sparsity and large pattern size, (a) F1 score, (b) recall, (c) precision.”

Thank you for your comment. By “pattern size”, we mean the percentages of the marked area of SVG. In Figure S7, the “large pattern size” refers to the gradient pattern. We realize that this was not clear enough. In the revised manuscript and the appendix, we have specified the spatial pattern name and the corresponding percentages of marked area of the SVGs (**Table S1 in this reply, Page 53**) for each simulation. We have deleted original Figure S7 and instead performed simulations with more spatial patterns and different distributions (**Figure S1-S66 in this reply, Page 56-125**), representative simulation results are shown in the reply to Comment 10.

14. On Figure S19 legend what are: “MT” genes? “Figure S19 ‘MT’ genes in primary colorectal cancer tissue and liver metastasis cancer tissue.”

Thank you for your comment. By “MT” genes, we mean the mitochondrial-encoded genes that start with “MT”. We have changed the legend of **Figure S19 in the original Supplementary (Figure 20 in the revised Supplementary, Page 28)** to “Mitochondrial-encoded (MT-) genes in primary colorectal cancer tissue and liver metastasis cancer tissue.”

15. On Figures S23-S24 add which scale is being used to display the expression values.

Thank you for your comment. We have added the expression scale legend to Figures S23-S24 (**Figures S23-S24 in the Supplementary, Page 34-35**). The revised figures are shown below.

a

b

Figure S23 (in the Supplementary) (a) Cell annotations of HDST data. (b) Representative svgs identified by HEARTSVG.

Figure S24 (in the Supplementary) (a) Original hematoxylin and eosin stained (H&E) tissue image. (b) Unsupervised spatial clustering results. (c) SVGs cluster patterns. (d) Representative genes of six SVG clusters.

Comments on the Quality of English Language

16. The paper is easy to follow and understand, but I would recommend editing by an English-speaking editor, in particular the Result section, to remove sentence duplications, and select the appropriate plural form, prepositions and articles and any other orthographic typos.

Thank you for your comments on our paper. We have revised our paper according to your comments, and we have also asked an English-speaking editor to proofread our paper. We hope that our paper meets your expectations and standards now.

17. Consider uniformly using GO (vs Go) for Gene Ontology terms and uppercase gene symbols throughout the manuscript.

Thank you for your comments. We have carefully revised the manuscript again to ensure consistent use of 'GO' as the Gene Ontology term throughout the entire document. Moreover, we capitalized the gene symbols fully in uppercase for human samples and used only the first letter in uppercase for mouse samples.

18. Line 140: Consider correcting this sentence: The *F1* score was used to assess the performances of identification SVGs of HEARTSVG and three other SVG detection methods in identifying SVGs."

Thank you for your comment. We have modified this sentence to "We used the F_1 score to assess the performance of HEARTSVG and the other methods in identifying SVGs."

19. Line 144: Consider reframing this sentence: "SpatialDE, SPARK struggled to identify SVGs in sparse data possibly due to their adoption of the Gaussian data-generative model, which is not well-suited for gene spatial expression distribution." Why is it not well suited?

Thank you for your comment. We have modified this sentence to "SpatialDE and SPARK performed poorly on sparse spatial expression data, possibly because they used a Gaussian data-generative model. The sparse and skewed spatial expression data do not satisfy the assumptions of the Gaussian data-generative model, which requires the data to be continuous, normally distributed, and independent."

20. Line 147: Consider reframing this sentence: “scGCO exhibited many false negatives in due to its difficulty in accurately identifying candidate regions for SVGs in highly sparse datasets.”

Thank you for your comment. We have modified this sentence to “scGCO failed to identify many SVGs in highly sparse datasets because it could not detect the candidate regions for SVGs accurately.”

21. Lines 147-8: Remove the repetition from sentence: “On the low sparsity simulated data, scGCO showed improved *F1* scores on simulated data with lower sparsity (*F1* score=0.333).

Thank you for your comment. Due to the revision of the manuscript content, we deleted this sentence in the revised manuscript.

22. Line 149: Consider re-writing this sentence to clarify what it means: “Identification performance was influenced by the pattern sizes of SVGs and sample sizes (Fig.2b, S1-3). What sample sizes were used? and what are pattern size? Described in Methods?

Thank you for your comment. We have modified this sentence to “Identification performance was influenced by the percentage of the marked area of SVGs and the number of cells/spots (Fig.S1-3).” By “pattern size”, we mean the percentage of the marked area of the SVGs. “Sample size” means the number of the cells/spots. We realize that the previous sentence lacked clarity. In the revised manuscript and the supplementary, we used “the percentage of the marked area of SVGs” and added the information of percentages of marked area of SVGs in **Table S1** in this reply.

23. Figure 1 legend: Edit inflaation to inflation

Thank you for your comment. We have revised Figure 1 and its legend, as indicated in our response to Comment 4.

References

1. Palla, G. *et al.* Squidpy: a scalable framework for spatial omics analysis. *Nat. Methods* **19**, 171–178 (2022).
2. Wu, S. Z. *et al.* A single-cell and spatially resolved atlas of human breast cancers. *Nat. Genet.* **53**, 1334–1347 (2021).
3. Xue, R. *et al.* Liver tumour immune microenvironment subtypes and neutrophil heterogeneity. *Nature* **612**, 141–147 (2022).
4. Guinney, J. *et al.* The consensus molecular subtypes of colorectal cancer. *Nat. Med.* **21**, 1350–1356 (2015).
5. Dienstmann, R. *et al.* Colorectal Cancer Subtyping Consortium (CRCSC) Identifies Consensus of Molecular Subtypes. *Ann. Oncol.* **25**, ii115 (2014).
6. Zhang, Y. *et al.* MetaTiME integrates single-cell gene expression to characterize the meta-components of the tumor immune microenvironment. *Nat. Commun.* **14**, 2634 (2023).
7. Zilbauer, M. *et al.* A Roadmap for the Human Gut Cell Atlas. *Nat. Rev. Gastroenterol. Hepatol.* **20**, 597–614 (2023).
8. Bied, M., Ho, W. W., Ginhoux, F. & Blériot, C. Roles of macrophages in tumor development: a spatiotemporal perspective. *Cell. Mol. Immunol.* **20**, 983–992 (2023).
9. Zheng, X. *et al.* Single-cell analyses implicate ascites in remodeling the ecosystems of primary and metastatic tumors in ovarian cancer. *Nat. Cancer* **4**, 1138–1156 (2023).
10. Tokura, M. *et al.* Single-Cell Transcriptome Profiling Reveals Intratumoral Heterogeneity and Molecular Features of Ductal Carcinoma *In Situ*. *Cancer Res.* **82**, 3236–3248 (2022).
11. Hsieh, W.-C. *et al.* Spatial multi-omics analyses of the tumor immune microenvironment. *J. Biomed. Sci.* **29**, 96 (2022).
12. Jung, G., Hernández-Illán, E., Moreira, L., Balaguer, F. & Goel, A. Epigenetics of colorectal cancer: biomarker and therapeutic potential. *Nat. Rev. Gastroenterol. Hepatol.* **17**, 111–130 (2020).
13. Li, J., Ma, X., Chakravarti, D., Shalpour, S. & DePinho, R. A. Genetic and biological hallmarks of colorectal cancer. *Genes Dev.* **35**, 787–820 (2021).
14. Lee, H.-O. *et al.* Lineage-dependent gene expression programs influence the immune landscape of colorectal cancer. *Nat. Genet.* **52**, 594–603 (2020).
15. Joanito, I. *et al.* Single-cell and bulk transcriptome sequencing identifies two epithelial tumor cell states and refines the consensus molecular classification of colorectal cancer. *Nat. Genet.* **54**, 963–975 (2022).
16. Qi, J. *et al.* Single-cell and spatial analysis reveal interaction of FAP+ fibroblasts and SPP1+ macrophages in colorectal cancer. *Nat. Commun.* **13**, 1742 (2022).
17. Pelka, K. *et al.* Spatially organized multicellular immune hubs in human colorectal cancer. *Cell* **184**, 4734–4752.e20 (2021).
18. Schmitt, M. & Greten, F. R. The inflammatory pathogenesis of colorectal cancer. *Nat. Rev. Immunol.* **21**, 653–667 (2021).
19. Heard, N. A. & Rubin-Delanchy, P. Choosing between methods of combining p -values. *Biometrika* **105**, 239–246 (2018).

Figures and Tables

Table S1 Spatial patterns and corresponding percentages of marked area for SVGs(%)

Pattern	Percentages of marked area of the SVGs (%)
Hotspot	5
Streak	5
Gradient	15
Ring	15
Nested rings	15
Streaks	10
Curve	7.5
Rectangles	5
Big triangles	15
Big circles	15
Big squares	15
Small triangles	7.5
Small circles	7.5
Small squares	7.5
Big circles II	15
Small triangles II	15
Pattern I	20
Pattern II	20
Pattern III	20
Irreg pat I	10
Irreg pat II	5
Irreg pat III	5

Table S2 Simulation settings of simulated data

Non-SVG and SVG of non-marked cells/spots				SVG of marked cells/spots			
	zero proportion	mu (NB, ZINB) / lambda (Poisson, ZIP)	size	zero proportion	mu (NB, ZINB) / lambda (Poisson, ZIP)	size	
ZINB	0.941	0.5	0.5	0.5	1	1	
ZIP	0.654	2	-	0.202	6	-	
NB	0.5	0.5	1.5	0.364	1.5	1.5	
Poisson	0.607	0.5	-	0.223	1.5	-	

1. Results of simulations with mixture noise

We simulated 3000 cells with 6000 genes (each set has 1000 genes) in each scenario. We generated 1000 simulated SVGs and randomly rearranged the gene expressions to generate non-SVGs. Then, we mix their expression in different proportions to create non-SVGs, SVGs with noise, and non-SVGs with noise (**Figure S2, Table S3 in this reply**).

Table S3 Mix proportions.

	percentage of SVGs' expression (%)	percentage of SVGs' expression (%)
Set2: SVGs with low noise	90	10
Set3: SVGs with medium noise	80	20
Set5: non-SVGs with low noise	10	90
Set5: non-SVGs with medium noise	20	80

Figure S1 Simulation results of SVGs identification using simulated data with mixture noise. (a) Visualization of Pattern: Hotspot with mixture noise. (b-e) Simulation results of six different methods (HEARTSVG, scGCO, SPARK, SPARK-X, Squidpy) on simulated data generated by four distinct distributions (ZINB, ZIP, NB, Poisson). The bar diagram (sub-panel (1)) shows F1 scores, TPRs, precisions, and FPRs. The heatmap (sub-panel (2)) depicts the comparison of TPR values among six genesets.

Figure S2 Simulation results of SVGs identification using simulated data with mixture noise.

(a) Visualization of Pattern: Streak with mixture noise. (b-e) Simulation results of six different methods (HEARTSVG, scGCO, SPARK, SPARK-X, Squidpy) on simulated data generated by four distinct distributions (ZINB, ZIP, NB, Poisson). The bar diagram (sub-panel (1)) shows F1 scores, TPRs, precisions, and FPRs. The heatmap (sub-panel (2)) depicts the comparison of TPR values among six genesets.

Figure S3 Simulation results of SVGs identification using simulated data with mixture noise.

(a) Visualization of Pattern: Gradient with mixture noise. (b-e) Simulation results of six different methods (HEARTSVG, scGCO, SPARK, SPARK-X, Squidpy) on simulated data generated by four distinct distributions (ZINB, ZIP, NB, Poisson). The bar diagram (sub-panel (1)) shows F1 scores, TPRs, precisions, and FPRs. The heatmap (sub-panel (2)) depicts the comparison of TPR values among six genesets.

Figure S4 Simulation results of SVGs identification using simulated data with mixture noise.

(a) Visualization of Pattern: Ring with mixture noise. (b-e) Simulation results of six different methods (HEARTSVG, scGCO, SPARK, SPARK-X, Squidpy) on simulated data generated by four distinct distributions (ZINB, ZIP, NB, Poisson). The bar diagram (sub-panel (1)) shows F1 scores, TPRs, precisions, and FPRs. The heatmap (sub-panel (2)) depicts the comparison of TPR values among six genesets.

Figure S5 Simulation results of SVGs identification using simulated data with mixture noise.

(a) Visualization of Pattern: Nested rings with mixture noise. (b-e) Simulation results of six different methods (HEARTSVG, scGCO, SPARK, SPARK-X, Squidpy) on simulated data generated by four distinct distributions (ZINB, ZIP, NB, Poisson). The bar diagram (sub-panel (1)) shows F1 scores, TPRs, precisions, and FPRs. The heatmap (sub-panel (2)) depicts the comparison of TPR values among six genesets.

Figure S6 Simulation results of SVGs identification using simulated data with mixture noise.

(a) Visualization of Pattern: Streaks with mixture noise. (b-e) Simulation results of six different methods (HEARTSVG, scGCO, SPARK, SPARK-X, Squidpy) on simulated data generated by four distinct distributions (ZINB, ZIP, NB, Poisson). The bar diagram (sub-panel (1)) shows F1 scores, TPRs, precisions, and FPRs. The heatmap (sub-panel (2)) depicts the comparison of TPR values among six genesets.

Figure S7 Simulation results of SVGs identification using simulated data with mixture noise.

(a) Visualization of Pattern: Curve with mixture noise. (b-e) Simulation results of six different methods (HEARTSVG, scGCO, SPARK, SPARK-X, Squidpy) on simulated data generated by four distinct distributions (ZINB, ZIP, NB, Poisson). The bar diagram (sub-panel (1)) shows F1 scores, TPRs, precisions, and FPRs. The heatmap (sub-panel (2)) depicts the comparison of TPR values among six genesets.

Figure S8 Simulation results of SVGs identification using simulated data with mixture noise.

(a) Visualization of Pattern: Rectangles with mixture noise. (b-e) Simulation results of six different methods (HEARTSVG, scGCO, SPARK, SPARK-X, Squidpy) on simulated data generated by four distinct distributions (ZINB, ZIP, NB, Poisson). The bar diagram (sub-panel (1)) shows F1 scores, TPRs, precisions, and FPRs. The heatmap (sub-panel (2)) depicts the comparison of TPR values among six genesets.

Figure S9 Simulation results of SVGs identification using simulated data with mixture noise.

(a) Visualization of Pattern: Big triangles with mixture noise. (b-e) Simulation results of six different methods (HEARTSVG, scGCO, SPARK, SPARK-X, Squidpy) on simulated data generated by four distinct distributions (ZINB, ZIP, NB, Poisson). The bar diagram (sub-panel (1)) shows F1 scores, TPRs, precisions, and FPRs. The heatmap (sub-panel (2)) depicts the comparison of TPR values among six genesets.

Figure S10 Simulation results of SVGs identification using simulated data with mixture noise.

(a) Visualization of Pattern: Big circles with mixture noise. (b-e) Simulation results of six different methods (HEARTSVG, scGCO, SPARK, SPARK-X, Squidpy) on simulated data generated by four distinct distributions (ZINB, ZIP, NB, Poisson). The bar diagram (sub-panel (1)) shows F1 scores, TPRs, precisions, and FPRs. The heatmap (sub-panel (2)) depicts the comparison of TPR values among six genesets.

Figure S11 Simulation results of SVGs identification using simulated data with mixture noise.

(a) Visualization of Pattern: Big squares with mixture noise. (b-e) Simulation results of six different methods (HEARTSVG, scGCO, SPARK, SPARK-X, Squidpy) on simulated data generated by four distinct distributions (ZINB, ZIP, NB, Poisson). The bar diagram (sub-panel (1)) shows F1 scores, TPRs, precisions, and FPRs. The heatmap (sub-panel (2)) depicts the comparison of TPR values among six genesets.

Figure S12 Simulation results of SVGs identification using simulated data with mixture noise.

(a) Visualization of Pattern: Small triangles with mixture noise. (b-e) Simulation results of six different methods (HEARTSVG, scGCO, SPARK, SPARK-X, Squidpy) on simulated data generated by four distinct distributions (ZINB, ZIP, NB, Poisson). The bar diagram (sub-panel (1)) shows F1 scores, TPRs, precisions, and FPRs. The heatmap (sub-panel (2)) depicts the comparison of TPR values among six genesets.

Figure S13 Simulation results of SVGs identification using simulated data with mixture noise.

(a) Visualization of Pattern: Small circles with mixture noise. (b-e) Simulation results of six different methods (HEARTSVG, scGCO, SPARK, SPARK-X, Squidpy) on simulated data generated by four distinct distributions (ZINB, ZIP, NB, Poisson). The bar diagram (sub-panel (1)) shows F1 scores, TPRs, precisions, and FPRs. The heatmap (sub-panel (2)) depicts the comparison of TPR values among six genesets.

Figure S14 Simulation results of SVGs identification using simulated data with mixture noise.

(a) Visualization of Pattern: Small squares with mixture noise. (b-e) Simulation results of six different methods (HEARTSVG, scGCO, SPARK, SPARK-X, Squidpy) on simulated data generated by four distinct distributions (ZINB, ZIP, NB, Poisson). The bar diagram (sub-panel (1)) shows F1 scores, TPRs, precisions, and FPRs. The heatmap (sub-panel (2)) depicts the comparison of TPR values among six genesets.

Figure S15 Simulation results of SVGs identification using simulated data with mixture noise.

(a) Visualization of Pattern: Big circles II with mixture noise. (b-e) Simulation results of six different methods (HEARTSVG, scGCO, SPARK, SPARK-X, Squidpy) on simulated data generated by four distinct distributions (ZINB, ZIP, NB, Poisson). The bar diagram (sub-panel (1)) shows F1 scores, TPRs, precisions, and FPRs. The heatmap (sub-panel (2)) depicts the comparison of TPR values among six genesets.

Figure S16 Simulation results of SVGs identification using simulated data with mixture noise.

(a) Visualization of Pattern: Small triangles II with mixture noise. (b-e) Simulation results of six different methods (HEARTSVG, scGCO, SPARK, SPARK-X, Squidpy) on simulated data generated by four distinct distributions (ZINB, ZIP, NB, Poisson). The bar diagram (sub-panel (1)) shows F1 scores, TPRs, precisions, and FPRs. The heatmap (sub-panel (2)) depicts the comparison of TPR values among six genesets.

Figure S17 Simulation results of SVGs identification using simulated data with mixture noise.

(a) Visualization of Pattern: Pattern I with mixture noise. (b-e) Simulation results of six different methods (HEARTSVG, scGCO, SPARK, SPARK-X, Squidpy) on simulated data generated by four distinct distributions (ZINB, ZIP, NB, Poisson). The bar diagram (sub-panel (1)) shows F1 scores, TPRs, precisions, and FPRs. The heatmap (sub-panel (2)) depicts the comparison of TPR values among six genesets.

Figure S18 Simulation results of SVGs identification using simulated data with mixture noise.

(a) Visualization of Pattern: Pattern II with mixture noise. (b-e) Simulation results of six different methods (HEARTSVG, scGCO, SPARK, SPARK-X, Squidpy) on simulated data generated by four distinct distributions (ZINB, ZIP, NB, Poisson). The bar diagram (sub-panel (1)) shows F1 scores, TPRs, precisions, and FPRs. The heatmap (sub-panel (2)) depicts the comparison of TPR values among six genesets.

Figure S19 Simulation results of SVGs identification using simulated data with mixture noise.

(a) Visualization of Pattern: Pattern III with mixture noise. (b-e) Simulation results of six different methods (HEARTSVG, scGCO, SPARK, SPARK-X, Squidpy) on simulated data generated by four distinct distributions (ZINB, ZIP, NB, Poisson). The bar diagram (sub-panel (1)) shows F1 scores, TPRs, precisions, and FPRs. The heatmap (sub-panel (2)) depicts the comparison of TPR values among six genesets.

Figure S20 Simulation results of SVGs identification using simulated data with mixture noise.

(a) Visualization of Pattern: Irreg pat I with mixture noise. (b-e) Simulation results of six different methods (HEARTSVG, scGCO, SPARK, SPARK-X, Squidpy) on simulated data generated by four distinct distributions (ZINB, ZIP, NB, Poisson). The bar diagram (sub-panel (1)) shows F1 scores, TPRs, precisions, and FPRs. The heatmap (sub-panel (2)) depicts the comparison of TPR values among six genesets.

Figure S21 Simulation results of SVGs identification using simulated data with mixture noise.

(a) Visualization of Pattern: Irreg pat II with mixture noise. (b-e) Simulation results of six different methods (HEARTSVG, scGCO, SPARK, SPARK-X, Squidpy) on simulated data generated by four distinct distributions (ZINB, ZIP, NB, Poisson). The bar diagram (sub-panel (1)) shows F1 scores, TPRs, precisions, and FPRs. The heatmap (sub-panel (2)) depicts the comparison of TPR values among six genesets.

Figure S22 Simulation results of SVGs identification using simulated data with mixture noise.

(a) Visualization of Pattern: Irreg pat III with mixture noise. (b-e) Simulation results of six different methods (HEARTSVG, scGCO, SPARK, SPARK-X, Squidpy) on simulated data generated by four distinct distributions (ZINB, ZIP, NB, Poisson). The bar diagram (sub-panel (1)) shows F1 scores, TPRs, precisions, and FPRs. The heatmap (sub-panel (2)) depicts the comparison of TPR values among six genesets.

Figure S23 Simulation results of SVGs identification using simulated data with mixture noise.

(a) Visualization of Pattern: Irreg pat IV with mixture noise. (b-e) Simulation results of six different methods (HEARTSVG, scGCO, SPARK, SPARK-X, Squidpy) on simulated data generated by four distinct distributions (ZINB, ZIP, NB, Poisson). The bar diagram (sub-panel (1)) shows F1 scores, TPRs, precisions, and FPRs. The heatmap (sub-panel (2)) depicts the comparison of TPR values among six genesets.

2. Results of noise-free simulations

Figure S23 a, Visualization of 23 representative spatial expression patterns: Hotspot, Streak, Gradient, Ring, Nested rings, Streaks, Curve, Rectangles, Big triangles, Big circles, Big squares, Small triangles, Small circles, Small squares, Big circles II, Small triangles II, Pattern I, Pattern II, Pattern III, Irreg pat I, Irreg pat II, Irreg pat III. b-e, F1 score plots, TPR plots, Precision plots, and FPR plots compare the index values (y-axis) of HEARTSVG (red), scGCO (orange), SpatialDE (yellow), SPARK (blue), SPARK-X (green), and Squidpy (purple) in simulation data. The

comparison is based on varying sample sizes (x-axis) at an adjusted p-value cutoff of 0.05. Each plot corresponds to the left spatial patterns in sub-figure **a**.

3. Results of simulations with Gaussian noise

We simulated 3000 cells with 10,000 genes (1000 SVGs and 9000 non-SVGs) in each scenario. We added varying levels (ranging from 0 to 0.6) of noises to noise-free data to create simulated datasets with different degrees of noise. The parameters of the four distributions we used were shown in Tables S1 and S2.

Figure S24 Simulation results for identifying SVGs using simulated data with Gaussian noise. (a) Visualization of Pattern: Hotspot with varying level of Gaussian noise. (b-e) Simulation results of four different methods (HEARTSVG, scGCO, SPARK-X and Squidpy) on simulated data generated by four distinct distributions (ZINB, ZIP, NB, Poisson). F1 score plots, TPR plots, Precision plots, and FPR plots compare the index values (y-axis) of HEARTSVG (red), scGCO (orange), SPARK-X (green), and Squidpy (purple) across varying levels of Gaussian noise strength (x-axis). Index values were calculated at the adjusted p-value cutoff of 0.05.

Figure S25 Simulation results for identifying SVGs using simulated data with Gaussian noise. (a) Visualization of Pattern: Streak with varying level of Gaussian noise. (b-e) Simulation results of four different methods (HEARTSVG, scGCO, SPARK-X and Squidpy) on simulated data generated by four distinct distributions (ZINB, ZIP, NB, Poisson). F1 score plots, TPR plots, Precision plots, and FPR plots compare the index values (y-axis) of HEARTSVG (red), scGCO (orange), SPARK-X (green), and Squidpy (purple) across varying levels of Gaussian noise strength (x-axis). Index values were calculated at the adjusted p-value cutoff of 0.05.

Figure S26 Simulation results for identifying SVGs using simulated data with Gaussian noise. (a) Visualization of Pattern: Gradient with varying level of Gaussian noise. (b-e) Simulation results of four different methods (HEARTSVG, scGCO, SPARK-X and Squidpy) on simulated data generated by four distinct distributions (ZINB, ZIP, NB, Poisson). F1 score plots, TPR plots, Precision plots, and FPR plots compare the index values (y-axis) of HEARTSVG (red), scGCO (orange), SPARK-X (green), and Squidpy (purple) across varying levels of Gaussian noise strength (x-axis). Index values were calculated at the adjusted p-value cutoff of 0.05.

Figure S27 Simulation results for identifying SVGs using simulated data with Gaussian noise. (a) Visualization of Pattern: Ring with varying level of Gaussian noise. (b-e) Simulation results of four different methods (HEARTSVG, scGCO, SPARK-X and Squidpy) on simulated data generated by four distinct distributions (ZINB, ZIP, NB, Poisson). F1 score plots, TPR plots, Precision plots, and FPR plots compare the index values (y-axis) of HEARTSVG (red), scGCO (orange), SPARK-X (green), and Squidpy (purple) across varying levels of Gaussian noise strength (x-axis). Index values were calculated at the adjusted p-value cutoff of 0.05.

Figure S28 Simulation results for identifying SVGs using simulated data with Gaussian noise. (a) Visualization of Pattern: Nested rings with varying level of Gaussian noise. (b-e) Simulation results of four different methods (HEARTSVG, scGCO, SPARK-X and Squidpy) on simulated data generated by four distinct distributions (ZINB, ZIP, NB, Poisson). F1 score plots, TPR plots, Precision plots, and FPR plots compare the index values (y-axis) of HEARTSVG (red), scGCO (orange), SPARK-X (green), and Squidpy (purple) across varying levels of Gaussian noise strength (x-axis). Index values were calculated at the adjusted p-value cutoff of 0.05.

Figure S29 Simulation results for identifying SVGs using simulated data with Gaussian noise.

(a) Visualization of Pattern: Streaks with varying level of Gaussian noise. (b-e) Simulation results of four different methods (HEARTSVG, scGCO, SPARK-X and Squidpy) on simulated data generated by four distinct distributions (ZINB, ZIP, NB, Poisson). F1 score plots, TPR plots, Precision plots, and FPR plots compare the index values (y-axis) of HEARTSVG (red), scGCO (orange), SPARK-X (green), and Squidpy (purple) across varying levels of Gaussian noise strength (x-axis). Index values were calculated at the adjusted p-value cutoff of 0.05.

Figure S30 Simulation results for identifying SVGs using simulated data with Gaussian noise.

(a) Visualization of Pattern: Curve with varying level of Gaussian noise. (b-e) Simulation results of four different methods (HEARTSVG, scGCO, SPARK-X and Squidpy) on simulated data generated by four distinct distributions (ZINB, ZIP, NB, Poisson). F1 score plots, TPR plots, Precision plots, and FPR plots compare the index values (y-axis) of HEARTSVG (red), scGCO (orange), SPARK-X (green), and Squidpy (purple) across varying levels of Gaussian noise strength (x-axis). Index values were calculated at the adjusted p-value cutoff of 0.05.

Figure S31 Simulation results for identifying SVGs using simulated data with Gaussian noise. (a) Visualization of Pattern: Rectangles with varying level of Gaussian noise. (b-e) Simulation results of four different methods (HEARTSVG, scGCO, SPARK-X and Squidpy) on simulated data generated by four distinct distributions (ZINB, ZIP, NB, Poisson). F1 score plots, TPR plots, Precision plots, and FPR plots compare the index values (y-axis) of HEARTSVG (red), scGCO (orange), SPARK-X (green), and Squidpy (purple) across varying levels of Gaussian noise strength (x-axis). Index values were calculated at the adjusted p-value cutoff of 0.05.

Figure S32 Simulation results for identifying SVGs using simulated data with Gaussian noise. (a) Visualization of Pattern: Big triangles with varying level of Gaussian noise. (b-e) Simulation results of four different methods (HEARTSVG, scGCO, SPARK-X and Squidpy) on simulated data generated by four distinct distributions (ZINB, ZIP, NB, Poisson). F1 score plots, TPR plots, Precision plots, and FPR plots compare the index values (y-axis) of HEARTSVG (red), scGCO (orange), SPARK-X (green), and Squidpy (purple) across varying levels of Gaussian noise strength (x-axis). Index values were calculated at the adjusted p-value cutoff of 0.05.

Figure S33 Simulation results for identifying SVGs using simulated data with Gaussian noise.

(a) Visualization of Pattern: Big circles with varying level of Gaussian noise. (b-e) Simulation results of four different methods (HEARTSVG, scGCO, SPARK-X and Squidpy) on simulated data generated by four distinct distributions (ZINB, ZIP, NB, Poisson). F1 score plots, TPR plots, Precision plots, and FPR plots compare the index values (y-axis) of HEARTSVG (red), scGCO (orange), SPARK-X (green), and Squidpy (purple) across varying levels of Gaussian noise strength (x-axis). Index values were calculated at the adjusted p-value cutoff of 0.05.

Figure S34 Simulation results for identifying SVGs using simulated data with Gaussian noise. (a) Visualization of Pattern: Big squares with varying level of Gaussian noise. (b-e) Simulation results of four different methods (HEARTSVG, scGCO, SPARK-X and Squidpy) on simulated data generated by four distinct distributions (ZINB, ZIP, NB, Poisson). F1 score plots, TPR plots, Precision plots, and FPR plots compare the index values (y-axis) of HEARTSVG (red), scGCO (orange), SPARK-X (green), and Squidpy (purple) across varying levels of Gaussian noise strength (x-axis). Index values were calculated at the adjusted p-value cutoff of 0.05.

Figure S35 Simulation results for identifying SVGs using simulated data with Gaussian noise.

(a) Visualization of Pattern: Small triangles with varying level of Gaussian noise. (b-e) Simulation results of four different methods (HEARTSVG, scGCO, SPARK-X and Squidpy) on simulated data generated by four distinct distributions (ZINB, ZIP, NB, Poisson). F1 score plots, TPR plots, Precision plots, and FPR plots compare the index values (y-axis) of HEARTSVG (red), scGCO (orange), SPARK-X (green), and Squidpy (purple) across varying levels of Gaussian noise strength (x-axis). Index values were calculated at the adjusted p-value cutoff of 0.05.

Figure S36 Simulation results for identifying SVGs using simulated data with Gaussian noise. (a) Visualization of Pattern: Small circles with varying level of Gaussian noise. (b-e) Simulation results of four different methods (HEARTSVG, scGCO, SPARK-X and Squidpy) on simulated data generated by four distinct distributions (ZINB, ZIP, NB, Poisson). F1 score plots, TPR plots, Precision plots, and FPR plots compare the index values (y-axis) of HEARTSVG (red), scGCO (orange), SPARK-X (green), and Squidpy (purple) across varying levels of Gaussian noise strength (x-axis). Index values were calculated at the adjusted p-value cutoff of 0.05.

Figure S37 Simulation results for identifying SVGs using simulated data with Gaussian noise. (a) Visualization of Pattern: Small squares with varying level of Gaussian noise. (b-e) Simulation results of four different methods (HEARTSVG, scGCO, SPARK-X and Squidpy) on simulated data generated by four distinct distributions (ZINB, ZIP, NB, Poisson). F1 score plots, TPR plots, Precision plots, and FPR plots compare the index values (y-axis) of HEARTSVG (red), scGCO (orange), SPARK-X (green), and Squidpy (purple) across varying levels of Gaussian noise strength (x-axis). Index values were calculated at the adjusted p-value cutoff of 0.05.

Figure S38 Simulation results for identifying SVGs using simulated data with Gaussian noise. (a) Visualization of Pattern: Big circles II with varying level of Gaussian noise. (b-e) Simulation results of four different methods (HEARTSVG, scGCO, SPARK-X and Squidpy) on simulated data generated by four distinct distributions (ZINB, ZIP, NB, Poisson). F1 score plots, TPR plots, Precision plots, and FPR plots compare the index values (y-axis) of HEARTSVG (red), scGCO (orange), SPARK-X (green), and Squidpy (purple) across varying levels of Gaussian noise strength (x-axis). Index values were calculated at the adjusted p-value cutoff of 0.05.

Figure S39 Simulation results for identifying SVGs using simulated data with Gaussian noise. (a) Visualization of Pattern: Small triangles II with varying level of Gaussian noise. (b-e) Simulation results of four different methods (HEARTSVG, scGCO, SPARK-X and Squidpy) on simulated data generated by four distinct distributions (ZINB, ZIP, NB, Poisson). F1 score plots, TPR plots, Precision plots, and FPR plots compare the index values (y-axis) of HEARTSVG (red), scGCO (orange), SPARK-X (green), and Squidpy (purple) across varying levels of Gaussian noise strength (x-axis). Index values were calculated at the adjusted p-value cutoff of 0.05.

Figure S40 Simulation results for identifying SVGs using simulated data with Gaussian noise. (a) Visualization of Pattern: Pattern I with varying level of Gaussian noise. (b-e) Simulation results of four different methods (HEARTSVG, scGCO, SPARK-X and Squidpy) on simulated data generated by four distinct distributions (ZINB, ZIP, NB, Poisson). F1 score plots, TPR plots, Precision plots, and FPR plots compare the index values (y-axis) of HEARTSVG (red), scGCO (orange), SPARK-X (green), and Squidpy (purple) across varying levels of Gaussian noise strength (x-axis). Index values were calculated at the adjusted p-value cutoff of 0.05.

Figure S41 Simulation results for identifying SVGs using simulated data with Gaussian noise. (a) Visualization of Pattern: Pattern II with varying level of Gaussian noise. (b-e) Simulation results of four different methods (HEARTSVG, scGCO, SPARK-X and Squidpy) on simulated data generated by four distinct distributions (ZINB, ZIP, NB, Poisson). F1 score plots, TPR plots, Precision plots, and FPR plots compare the index values (y-axis) of HEARTSVG (red), scGCO (orange), SPARK-X (green), and Squidpy (purple) across varying levels of Gaussian noise strength (x-axis). Index values were calculated at the adjusted p-value cutoff of 0.05.

Figure S42 Simulation results for identifying SVGs using simulated data with Gaussian noise. (a) Visualization of Pattern: Pattern III with varying level of Gaussian noise. (b-e) Simulation results of four different methods (HEARTSVG, scGCO, SPARK-X and Squidpy) on simulated data generated by four distinct distributions (ZINB, ZIP, NB, Poisson). F1 score plots, TPR plots, Precision plots, and FPR plots compare the index values (y-axis) of HEARTSVG (red), scGCO (orange), SPARK-X (green), and Squidpy (purple) across varying levels of Gaussian noise strength (x-axis). Index values were calculated at the adjusted p-value cutoff of 0.05.

Figure S43 Simulation results for identifying SVGs using simulated data with Gaussian noise. (a) Visualization of Pattern: Irreg pat I with varying level of Gaussian noise. (b-e) Simulation results of four different methods (HEARTSVG, scGCO, SPARK-X and Squidpy) on simulated data generated by four distinct distributions (ZINB, ZIP, NB, Poisson). F1 score plots, TPR plots, Precision plots, and FPR plots compare the index values (y-axis) of HEARTSVG (red), scGCO (orange), SPARK-X (green), and Squidpy (purple) across varying levels of Gaussian noise strength (x-axis). Index values were calculated at the adjusted p-value cutoff of 0.05.

Figure S44 Simulation results for identifying SVGs using simulated data with Gaussian noise. (a) Visualization of Pattern: Irreg pat II with varying level of Gaussian noise. (b-e) Simulation results of four different methods (HEARTSVG, scGCO, SPARK-X and Squidpy) on simulated data generated by four distinct distributions (ZINB, ZIP, NB, Poisson). F1 score plots, TPR plots, Precision plots, and FPR plots compare the index values (y-axis) of HEARTSVG (red), scGCO (orange), SPARK-X (green), and Squidpy (purple) across varying levels of Gaussian noise strength (x-axis). Index values were calculated at the adjusted p-value cutoff of 0.05.

Figure S45 Simulation results for identifying SVGs using simulated data with Gaussian noise. (a) Visualization of Pattern: Irreg pat III with varying level of Gaussian noise. (b-e) Simulation results of four different methods (HEARTSVG, scGCO, SPARK-X and Squidpy) on simulated data generated by four distinct distributions (ZINB, ZIP, NB, Poisson). F1 score plots, TPR plots, Precision plots, and FPR plots compare the index values (y-axis) of HEARTSVG (red), scGCO (orange), SPARK-X (green), and Squidpy (purple) across varying levels of Gaussian noise strength (x-axis). Index values were calculated at the adjusted p-value cutoff of 0.05.

4. Results of simulations with noise of 'Randomly Exchanging Expression Values of Selected Nodes'

Due to the different ways of generating noise, we created new simulation data (both expression and coordinates) with noise, instead of transforming the noise-free data. We used the same parameter settings as in Section 1-3 to generate the spatial patterns and simulated 3000 cells with 10,000 genes (1000 SVGs and 9000 non-SVGs) in each scenario. The parameters of the four distributions we used are shown in Tables S1 and S2.

Figure S46 Simulation results for identifying SVGs using simulated data with noise of 'Randomly Exchanging Expression Values of Selected Nodes.' (a) Visualization of Pattern: Hotspot with different percentage of cells random exchanges (%). (b-e) Simulation results of four different methods (HEARTSVG, scGCO, SPARK-X and Squidpy) on simulated data generated by

four distinct distributions (ZINB, ZIP, NB, Poisson). F1 score plots, TPR plots, Precision plots, and FPR plots compare the index values (y-axis) of HEARTSVG (red), scGCO (orange), SPARK-X (green), and Squidpy (purple) across different percentage of cells random exchanges (x-axis). Index values were calculated at the adjusted p-value cutoff of 0.05.

Figure S47 Simulation results for identifying SVGs using simulated data with noise of 'Randomly Exchanging Expression Values of Selected Nodes.' (a) Visualization of Pattern: Streak with different percentage of cells random exchanges (%). (b-e) Simulation results of four different methods (HEARTSVG, scGCO, SPARK-X and Squidpy) on simulated data generated by four distinct distributions (ZINB, ZIP, NB, Poisson). F1 score plots, TPR plots, Precision plots, and FPR plots compare the index values (y-axis) of HEARTSVG (red), scGCO (orange), SPARK-X (green), and Squidpy (purple) across different percentage of cells random exchanges (x-axis). Index values were calculated at the adjusted p-value cutoff of 0.05.

Figure S48 Simulation results for identifying SVGs using simulated data with noise of 'Randomly Exchanging Expression Values of Selected Nodes.' (a) Visualization of Pattern: Ring with different percentage of cells random exchanges (%). (b-e) Simulation results of four different methods (HEARTSVG, scGCO, SPARK-X and Squidpy) on simulated data generated by four distinct distributions (ZINB, ZIP, NB, Poisson). F1 score plots, TPR plots, Precision plots, and FPR plots compare the index values (y-axis) of HEARTSVG (red), scGCO (orange), SPARK-X (green), and Squidpy (purple) across different percentage of cells random exchanges (x-axis). Index values were calculated at the adjusted p-value cutoff of 0.05.

Figure S49 Simulation results for identifying SVGs using simulated data with noise of 'Randomly Exchanging Expression Values of Selected Nodes.' (a) Visualization of Pattern: Nested rings with different percentage of cells random exchanges (%). (b-e) Simulation results of four different methods (HEARTSVG, scGCO, SPARK-X and Squidpy) on simulated data generated by four distinct distributions (ZINB, ZIP, NB, Poisson). F1 score plots, TPR plots, Precision plots, and FPR plots compare the index values (y-axis) of HEARTSVG (red), scGCO (orange), SPARK-X (green), and Squidpy (purple) across different percentage of cells random exchanges (x-axis). Index values were calculated at the adjusted p-value cutoff of 0.05.

Figure S50 Simulation results for identifying SVGs using simulated data with noise of 'Randomly Exchanging Expression Values of Selected Nodes.' (a) Visualization of Pattern: Streaks with different percentage of cells random exchanges (%). (b-e) Simulation results of four different methods (HEARTSVG, scGCO, SPARK-X and Squidpy) on simulated data generated by four distinct distributions (ZINB, ZIP, NB, Poisson). F1 score plots, TPR plots, Precision plots, and FPR plots compare the index values (y-axis) of HEARTSVG (red), scGCO (orange), SPARK-X (green), and Squidpy (purple) across different percentage of cells random exchanges (x-axis). Index values were calculated at the adjusted p-value cutoff of 0.05.

Figure S51 Simulation results for identifying SVGs using simulated data with noise of 'Randomly Exchanging Expression Values of Selected Nodes.' (a) Visualization of Pattern: Curve with different percentage of cells random exchanges (%). (b-e) Simulation results of four different methods (HEARTSVG, scGCO, SPARK-X and Squidpy) on simulated data generated by four distinct distributions (ZINB, ZIP, NB, Poisson). F1 score plots, TPR plots, Precision plots, and FPR plots compare the index values (y-axis) of HEARTSVG (red), scGCO (orange), SPARK-X (green), and Squidpy (purple) across different percentage of cells random exchanges (x-axis). Index values were calculated at the adjusted p-value cutoff of 0.05.

Figure S52 Simulation results for identifying SVGs using simulated data with noise of 'Randomly Exchanging Expression Values of Selected Nodes.' (a) Visualization of Pattern: Rectangles with different percentage of cells random exchanges (%). (b-e) Simulation results of four different methods (HEARTSVG, scGCO, SPARK-X and Squidpy) on simulated data generated by four distinct distributions (ZINB, ZIP, NB, Poisson). F1 score plots, TPR plots, Precision plots, and FPR plots compare the index values (y-axis) of HEARTSVG (red), scGCO (orange), SPARK-X (green), and Squidpy (purple) across different percentage of cells random exchanges (x-axis). Index values were calculated at the adjusted p-value cutoff of 0.05.

Figure S53 Simulation results for identifying SVGs using simulated data with noise of 'Randomly Exchanging Expression Values of Selected Nodes.' (a) Visualization of Pattern: Big triangles with different percentage of cells random exchanges (%). (b-e) Simulation results of four different methods (HEARTSVG, scGCO, SPARK-X and Squidpy) on simulated data generated by four distinct distributions (ZINB, ZIP, NB, Poisson). F1 score plots, TPR plots, Precision plots, and FPR plots compare the index values (y-axis) of HEARTSVG (red), scGCO (orange), SPARK-X (green), and Squidpy (purple) across different percentage of cells random exchanges (x-axis). Index values were calculated at the adjusted p-value cutoff of 0.05.

Figure S54 Simulation results for identifying SVGs using simulated data with noise of 'Randomly Exchanging Expression Values of Selected Nodes.' (a) Visualization of Pattern: Big circles with different percentage of cells random exchanges (%). (b-e) Simulation results of four different methods (HEARTSVG, scGCO, SPARK-X and Squidpy) on simulated data generated by four distinct distributions (ZINB, ZIP, NB, Poisson). F1 score plots, TPR plots, Precision plots, and FPR plots compare the index values (y-axis) of HEARTSVG (red), scGCO (orange), SPARK-X (green), and Squidpy (purple) across different percentage of cells random exchanges (x-axis). Index values were calculated at the adjusted p-value cutoff of 0.05.

Figure S55 Simulation results for identifying SVGs using simulated data with noise of 'Randomly Exchanging Expression Values of Selected Nodes.' (a) Visualization of Pattern: Big squares with different percentage of cells random exchanges (%). (b-e) Simulation results of four different methods (HEARTSVG, scGCO, SPARK-X and Squidpy) on simulated data generated by four distinct distributions (ZINB, ZIP, NB, Poisson). F1 score plots, TPR plots, Precision plots, and FPR plots compare the index values (y-axis) of HEARTSVG (red), scGCO (orange), SPARK-X (green), and Squidpy (purple) across different percentage of cells random exchanges (x-axis). Index values were calculated at the adjusted p-value cutoff of 0.05.

Figure S56 Simulation results for identifying SVGs using simulated data with noise of 'Randomly Exchanging Expression Values of Selected Nodes.' (a) Visualization of Pattern: Small triangles with different percentage of cells random exchanges (%). (b-e) Simulation results of four different methods (HEARTSVG, scGCO, SPARK-X and Squidpy) on simulated data generated by four distinct distributions (ZINB, ZIP, NB, Poisson). F1 score plots, TPR plots, Precision plots, and FPR plots compare the index values (y-axis) of HEARTSVG (red), scGCO (orange), SPARK-X (green), and Squidpy (purple) across different percentage of cells random exchanges (x-axis). Index values were calculated at the adjusted p-value cutoff of 0.05.

Figure S57 Simulation results for identifying SVGs using simulated data with noise of 'Randomly Exchanging Expression Values of Selected Nodes.' (a) Visualization of Pattern: Small circles with different percentage of cells random exchanges (%). (b-e) Simulation results of four different methods (HEARTSVG, scGCO, SPARK-X and Squidpy) on simulated data generated by four distinct distributions (ZINB, ZIP, NB, Poisson). F1 score plots, TPR plots, Precision plots, and FPR plots compare the index values (y-axis) of HEARTSVG (red), scGCO (orange), SPARK-X (green), and Squidpy (purple) across different percentage of cells random exchanges (x-axis). Index values were calculated at the adjusted p-value cutoff of 0.05.

Figure S58 Simulation results for identifying SVGs using simulated data with noise of 'Randomly Exchanging Expression Values of Selected Nodes.' (a) Visualization of Pattern: Small squares with different percentage of cells random exchanges (%). (b-e) Simulation results of four different methods (HEARTSVG, scGCO, SPARK-X and Squidpy) on simulated data generated by four distinct distributions (ZINB, ZIP, NB, Poisson). F1 score plots, TPR plots, Precision plots, and FPR plots compare the index values (y-axis) of HEARTSVG (red), scGCO (orange), SPARK-X (green), and Squidpy (purple) across different percentage of cells random exchanges (x-axis). Index values were calculated at the adjusted p-value cutoff of 0.05.

Figure S59 Simulation results for identifying SVGs using simulated data with noise of 'Randomly Exchanging Expression Values of Selected Nodes.' (a) Visualization of Pattern: Big circles II with different percentage of cells random exchanges (%). (b-e) Simulation results of four different methods (HEARTSVG, scGCO, SPARK-X and Squidpy) on simulated data generated by four distinct distributions (ZINB, ZIP, NB, Poisson). F1 score plots, TPR plots, Precision plots, and FPR plots compare the index values (y-axis) of HEARTSVG (red), scGCO (orange), SPARK-X (green), and Squidpy (purple) across different percentage of cells random exchanges (x-axis). Index values were calculated at the adjusted p-value cutoff of 0.05.

Figure S60 Simulation results for identifying SVGs using simulated data with noise of 'Randomly Exchanging Expression Values of Selected Nodes.' (a) Visualization of Pattern: Small triangles II with different percentage of cells random exchanges (%). (b-e) Simulation results of four different methods (HEARTSVG, scGCO, SPARK-X and Squidpy) on simulated data generated by four distinct distributions (ZINB, ZIP, NB, Poisson). F1 score plots, TPR plots, Precision plots, and FPR plots compare the index values (y-axis) of HEARTSVG (red), scGCO (orange), SPARK-X (green), and Squidpy (purple) across different percentage of cells random exchanges (x-axis). Index values were calculated at the adjusted p-value cutoff of 0.05.

Figure S61 Simulation results for identifying SVGs using simulated data with noise of 'Randomly Exchanging Expression Values of Selected Nodes.' (a) Visualization of Pattern I with different percentage of cells random exchanges (%). (b-e) Simulation results of four different methods (HEARTSVG, scGCO, SPARK-X and Squidpy) on simulated data generated by four distinct distributions (ZINB, ZIP, NB, Poisson). F1 score plots, TPR plots, Precision plots, and FPR plots compare the index values (y-axis) of HEARTSVG (red), scGCO (orange), SPARK-X (green), and Squidpy (purple) across different percentage of cells random exchanges (x-axis). Index values were calculated at the adjusted p-value cutoff of 0.05.

Figure S62 Simulation results for identifying SVGs using simulated data with noise of 'Randomly Exchanging Expression Values of Selected Nodes.' (a) Visualization of Pattern II with different percentage of cells random exchanges (%). (b-e) Simulation results of four different methods (HEARTSVG, scGCO, SPARK-X and Squidpy) on simulated data generated by four distinct distributions (ZINB, ZIP, NB, Poisson). F1 score plots, TPR plots, Precision plots, and FPR plots compare the index values (y-axis) of HEARTSVG (red), scGCO (orange), SPARK-X (green), and Squidpy (purple) across different percentage of cells random exchanges (x-axis). Index values were calculated at the adjusted p-value cutoff of 0.05.

Figure S63 Simulation results for identifying SVGs using simulated data with noise of 'Randomly Exchanging Expression Values of Selected Nodes.' (a) Visualization of Pattern III with different percentage of cells random exchanges (%). (b-e) Simulation results of four different methods (HEARTSVG, scGCO, SPARK-X and Squidpy) on simulated data generated by four distinct distributions (ZINB, ZIP, NB, Poisson). F1 score plots, TPR plots, Precision plots, and FPR plots compare the index values (y-axis) of HEARTSVG (red), scGCO (orange), SPARK-X (green), and Squidpy (purple) across different percentage of cells random exchanges (x-axis). Index values were calculated at the adjusted p-value cutoff of 0.05.

Figure S64 Simulation results for identifying SVGs using simulated data with noise of 'Randomly Exchanging Expression Values of Selected Nodes.' (a) Visualization of Pattern: Irreg pat I with different percentage of cells random exchanges (%). (b-e) Simulation results of four different methods (HEARTSVG, scGCO, SPARK-X and Squidpy) on simulated data generated by four distinct distributions (ZINB, ZIP, NB, Poisson). F1 score plots, TPR plots, Precision plots, and FPR plots compare the index values (y-axis) of HEARTSVG (red), scGCO (orange), SPARK-X (green), and Squidpy (purple) across different percentage of cells random exchanges (x-axis). Index values were calculated at the adjusted p-value cutoff of 0.05.

Figure S65 Simulation results for identifying SVGs using simulated data with noise of 'Randomly Exchanging Expression Values of Selected Nodes.' (a) Visualization of Pattern: Irreg pat II with different percentage of cells random exchanges (%). (b-e) Simulation results of four different methods (HEARTSVG, scGCO, SPARK-X and Squidpy) on simulated data generated by four distinct distributions (ZINB, ZIP, NB, Poisson). F1 score plots, TPR plots, Precision plots, and FPR plots compare the index values (y-axis) of HEARTSVG (red), scGCO (orange), SPARK-X (green), and Squidpy (purple) across different percentage of cells random exchanges (x-axis). Index values were calculated at the adjusted p-value cutoff of 0.05.

Figure S66 Simulation results for identifying SVGs using simulated data with noise of 'Randomly Exchanging Expression Values of Selected Nodes.' (a) Visualization of Pattern: Irreg pat III with different percentage of cells random exchanges (%). (b-e) Simulation results of four different methods (HEARTSVG, scGCO, SPARK-X and Squidpy) on simulated data generated by four distinct distributions (ZINB, ZIP, NB, Poisson). F1 score plots, TPR plots, Precision plots, and FPR plots compare the index values (y-axis) of HEARTSVG (red), scGCO (orange), SPARK-X (green), and Squidpy (purple) across different percentage of cells random exchanges (x-axis). Index values were calculated at the adjusted p-value cutoff of 0.05.

Reviewer #1 (Remarks to the Author):

1. I am concerned with the validity of the simulation results. There are multiple discrepancies.

For example, in figure S11, after exchanged 100% of spots, which should destroy the pattern and result in complete noise, squidpy, surprisingly, get FPR of 1 in 3 out of 4 tests. More surprisingly, the TPR of squidpy is 0 for all cases even when 0% of spots were exchanged. This data indicates that squidpy will identify all negative noise as SVGs while failed to identify any true SVGs. I think these results contradicts with the reported performance of squidpy.

Furthermore, the result in S11 is a direct conflict with data in figure S9, where the same double strip pattern is analysed with Gaussian noise. In S9, squidpy has a TPR of 1 for this pattern at Gaussian noise 0, but squidpy has a TPR of 0 for the same pattern in figure S11 with 0% exchange rate. How could squidpy identified the same pattern perfectly in one test, while completely failed in another?

In Figure S4-S6, multiple methods have 0 TPR, which is of major concern. These simulated patterns are presumedly with strong and clear spatial patterns. But multiple published methods failed to identify such simulated SVGs, indicating that these methods lack the ability to identify SVGs, which directly contradict with multiple published results.

Same concern goes with the simulation with Gaussian noises (Figure S8-S10). With a large amount of Gaussian noises, which also should destroy the underlying spatial patterns. Some method maintained 100% TPR up to Gaussian noise level 0.6. And some methods demonstrated similar patters with these in S11, showing increased FPR as noises increase, suggesting that they identified more SVGs from higher level of noises.

These discrepancies indicate that the data produced are of poor quality. The authors should clarify the mechanisms behind these irregular observations.

I understand that the authors try to demonstrate that their method is superior to ALL available ones in all scenarios. In my opinion this is not necessary as far as they demonstrate substantial improvements, provided that the evaluation of existing methods is reliable and, at least, should reproduce majority of published results.

2. The author should stop calculating metric such as FPR etc for real biological samples. They updated the true positive set to intersection of all methods tested. But the same logic applies, we simply don't know the set of true SVGs genes. Using the intersection of all methods is flawed just as using intersection of two methods.

3. The memory requirement of scGCO is very different from what was reported in the scGCO paper, which should be further investigated.

Reviewer #2 (Remarks to the Author):

The authors have addressed all the previous questions by including new SVG tool comparisons, new significant results and importantly including more detailed information on HEARTSVG method procedures and its rationale. This information was shared in the rebuttal letter but also nicely included in the revised manuscript. Thus, the authors show that HEARTSVG provides a considerable improvement of their methodology in their capability to accurately identify SVGs and provide scalability of the tool. Moreover, the manuscript was extensively revised, and it reads well. This paper is suitable for publication.

Before publication the authors need to address a set of minor comments which will not require

reviewers' re-evaluation.

Minor Comments

In Figure 2, substitute the top title from "Gaussian noise strength" by "Gaussian noise strength". The same applies to Supplemental Figures S63-S84. Additionally, fix the x-axis for Supplemental Figure S64.

In Figure 3E, include the relative expression units in the slide bar.

In Figure 4 panels A include a title for the slide bar and substitute "low/high" expression by a numeric scale.

Include the Yamamoto test reference in the manuscript.

In Supplemental Figures S1-S3, the x-axis is labelled with "Sample size". Would it be best to replace by "Number of cells" as shown in Figure 1 and Supplemental Figures S62-S84?

Dear Reviewers:

Thank you for the reviewers' comments concerning our manuscript entitled "HEARTSVG: a fast and accurate method for spatially variable gene identification in large-scale spatial transcriptomic data" (ID: NCOMMS-23-29155C). Your insights and comments have been invaluable in refining the quality and clarity of our work. We have carefully considered each of your comments and have made the necessary revisions to address them. We provide a detailed point-to-point response for each comment in the following document and make corresponding changes in the manuscript.

We hope these revisions satisfy your comments and improve the quality of our manuscript.

Response to Reviewer #1

1. I am concerned with the validity of the simulation results. There are multiple discrepancies.
 - 1) For example, in figure S11, after exchanged 100% of spots, which should destroy the pattern and result in complete noise, squidpy, surprisingly, get FPR of 1 in 3 out of 4 tests. More surprisingly, the TPR of squidpy is 0 for all cases even when 0% of spots were exchanged. This data indicates that squidpy will identify all negative noise as SVGs while failed to identify any true SVGs. I think these results contradicts with the reported performance of squidpy.
 - 2) Furthermore, the result in S11 is a direct conflict with data in figure S9, where the same double strip pattern is analysed with Gaussian noise. In S9, squidpy has a TPR of 1 for this pattern at Gaussian noise 0, but squidpy has a TPR of 0 for the same pattern in figure S11 with 0% exchange rate. How could squidpy identified the same pattern perfectly in one test, while completely failed in another?
 - 3) In Figure S4-S6, multiple methods have 0 TPR, which is of major concern. These simulated patterns are presumedly with strong and clear spatial patterns. But multiple published methods failed to identify such simulated SVGs, indicating that these methods lack the ability to identify SVGs, which directly contradict with multiple published results.
 - 4) Same concern goes with the simulation with Gaussian noises (Figure S8-S10). With a large amount of Gaussian noises, which also should destroy the underlying spatial patterns. Some method maintained 100% TPR up to Gaussian noise level 0.6. And some methods demonstrated similar patters with these in S11, showing increased FPR as noises increase, suggesting that they identified more SVGs from higher level of noises.

These discrepancies indicate that the data produced are of poor quality. The authors should clarify the mechanisms behind these irregular observations.

I understand that the authors try to demonstrate that their method is superior to ALL available ones in all scenarios. In my opinion this is not necessary as far as they demonstrate substantial improvements, provided that the evaluation of existing methods is reliable and, at least, should reproduce majority of published results.

Thank you very much for your comments. For the issues raised in Comment 1, we will address them point by point.

For points 1), 2), and 4), we appreciate your feedback on the conflict between TPR and FPR of **Figure S8-S11 in the previous response**, which helped us discover an error in our data processing of the newly added simulation results in the revision. Since these three issues are caused by the same reason, we will answer them together.

We noticed that the conflict between TPR and FPR occurred in the Squidpy method under relatively high levels of Gaussian noise and exchanging nodes noise simulation scenarios, but not in mixture noise and noise-free data. Then we carefully checked the code and the original results and found that this was caused by an error in the data processing of the simulation results. Specifically, gene expression values could become negative when adding Gaussian noise to the simulated data (We followed the scGCO method, and added Gaussian noise to the exchanging nodes noise simulated data). Squidpy computes Moran's I using **$\log(1+\text{counts})$** , not raw counts. When Gaussian noise is added to simulated data, there are cases where expression ≤ -1 , which results in $-\text{Inf}$ or NaN for $\log(1 + \text{counts})$. **This will result in NA (not available) values in the Moran's I and the p-value in the calculation by the Squidpy method. Once a p-value is NA, all the adjusted p-values become NA as well (Figure S1). This led to our incorrect calculation of TP=FP=TN=0, and Precision=0, TPR=0, and Specificity=0. We calculated FPR and F1 score based on Precision, TPR, and Specificity, leading to the error where TPR=0, but FPR=1-Specificity=1.**

We have now addressed this issue: if the gene expression value is less than 0 after adding noise, it is considered as 0; otherwise, it is the original value. Then, we reran Squidpy on simulations under Gaussian noise and exchanging nodes noise. The **Figure S8-S12 in the previous Response** have been updated and listed below (**Figure S2-S6 in this response**), and other results have been updated in the Supplementary.

For 1) “For example, in figure S11, after exchanged 100% of spots, which should destroy the pattern and result in complete noise, squidpy, surprisingly, get FPR of 1 in 3 out of 4 tests. More surprisingly, the TPR of squidpy is 0 for all cases even when 0% of spots were exchanged. This data indicates that squidpy will identify all negative noise as SVGs while failed to identify any true SVGs. I think these results contradicts with the reported performance of squidpy.”

In **previous Figure S11** (exchanging noise simulations in the previous response), we added Gaussian noise to the exchanging noise-simulated data following the procedures described by scGCO. Therefore, gene expressions would become negative and the Moran'I became NA. **This caused TPR=Specificity=0, resulting in a conflict between TPR and FPR (TPR=0, but FPR=1-Specificity=1).** After we corrected the simulation results, at 100 % exchanging rate, squidpy's F1 score, TPR, and FPR were close to zero in the four distributions (**Figure S5-S6, corresponding to Figure S11-S12 in the previous response**). The same problem also occurred in other simulations with exchanging nodes noise, and we corrected them all. The new figures are listed below, and other updated results are in the **new Supplementary (Figure S85-S105)**.

For 2) “Furthermore, the result in S11 is a direct conflict with data in figure S9, where the same double strip pattern is analysed with Gaussian noise. In S9, squidpy has a TPR of 1 for this pattern at Gaussian noise 0, but squidpy has a TPR of 0 for the same pattern in figure S11 with 0% exchange rate. How could squidpy identified the same pattern perfectly in one test, while completely failed in another?”

Due to the same reason as the previous question, the result in **previous Figure S11** (with 0% exchange rate) conflicted with the **previous Figure S9** (at Gaussian noise 0). After rectifying this error, in **Figure S5** (corresponding to **Figure S11 in the previous response**) simulations for the four distributions (ZINB, ZIP, NB, Pois) with a 0% exchange rate, squidpy's **TPR was 0.79, 1, 0.99, and 0.98, respectively,** and squidpy's FPRs were close to 0.

For 4) Same concern goes with the simulation with Gaussian noises (**Figure S8-S10**). With a large amount of Gaussian noises, which also should destroy the underlying spatial patterns. Some method maintained 100% TPR up to Gaussian noise level 0.6. And some methods demonstrated similar patters with these in S11, showing increased FPR as noises increase,

suggesting that they identified more SVGs from higher level of noises.

Gaussian noise caused negative expression values in the simulated data, resulting in **all the squidpy's adjusted p-values being NA** and **TPR=Specificity=0**. This resulted in the squidpy's error (TPR=0, but FPR=1-Specificity=1) in **FigureS8-S10 in the previous response**. Because we also added Gaussian noise to the exchanging nodes noise simulated data, and the same problem also occurred in other simulations with exchange node noise (**FigureS11-S12 in the previous response**). We corrected them all.

In our latest simulation results (as shown in **Figure S2-S4, corresponding to Figure S8-S10 in the previous response**), for the simulated data generated by ZIP and Pois distributions, SVG had higher expression levels and stronger signals, so the noise of "Gaussian noise level 0.6" had little impact on the data, and all methods maintained high TPRs and F1 scores, which was consistent with the simulation results of scGCO¹. But for the simulated data generated by ZINB and NB distributions, SVG had lower expression levels and weaker signals, so the noise of "Gaussian noise level 0.6" had a large impact on the data. HEARTSVG, SPARK-X, squidpy's TPRs and F1 scores dropped significantly.

The error (TPR=0, but FPR=1-Specificity=1) in **previous Figure S11** was caused by the same reason. After we corrected this error, squidpy's FPR was close to 0, and TPR increased with noise (as shown in **Figure S5-S6, corresponding to Figure S11-S12 in the previous response**),.

For 3) In Figure S4-S6, multiple methods have 0 TPR, which is of major concern. These simulated patterns are presumedly with strong and clear spatial patterns. But multiple published methods failed to identify such simulated SVGs, indicating that these methods lack the ability to identify SVGs, which directly contradict with multiple published results.

Thank you for your feedback. We appreciate your concern regarding the performance (TPR=0) of multiple methods on simulated data in **Figure S4-S6 in the previous Response**.

We would like to clarify that the simulated datasets in **Figure S4-S6 in the previous Response** were generated **with mixture noise**, which caused the performance of various methods to decrease. The low TTPRs for scGCO, Squidpy, SPARK, and SpatialDE in **previous Figure S4-S6** were due to the following two reasons:

- 1) the low TPRs for scGCO and Squidpy were a result of the mixture noise introduced in the simulations.
- 2) the use of approximate modeling of sparse count data by a Gaussian distribution in SpatialDE and the Gaussian version of SPARK is not ideal and leads to power loss² (Zhu, J., Sun, S. & Zhou, X).

To demonstrate these, we added the following two parts of simulations to illustrate the impact of mixture noise on the performance of scGCO and Squidpy, and the limitation of the inappropriate underlying data model and normalization mechanism on the identification ability of SPARK and SpatialDE, respectively.

Part I: How does the mixture noise impact the performance of scGCO and Squidpy?

We understand your concern about the noticeable decline in the performance (F1 score, TPR, etc.) of scGCO and Squidpy in this particular simulation (mixture noise), which contrasts with their performance in other simulated data: noise-free, Gaussian noise, and exchanging nodes noise simulated data. To further investigate the impact of mixture noise on the performance of scGCO and Squidpy, we generated two additional simulation datasets using the same simulated data with mixture noise (presented in **Figure S6 in the previous response**). The simulated data of **Figure S6 in the previous response**, contained six classes of geneset (each set has 1000 genes):

Set 1: SVGs;

Set 2: SVGs with low noise;

Set 3: SVGs with medium noise;

Set 4: non-SVGs;

Set 5: non-SVGs with low noise;

Set 6: non-SVGs with medium noise.

Using this data, we generated two additional simulation datasets:

Dataset 1 (2000 genes): A noise-free simulated dataset consisting of Set 1: SVGs and Set 4: non-SVGs;

Dataset 2 (4000 genes): A simulated dataset with weak noise, composed of Set 1: SVGs, Set 2: SVGs with low noise, Set 4: non-SVGs, and Set 5: non-SVGs with low noise.

We found that Squidpy's F1 score significantly decreased with the increase in noise (from 0.57 to 0.34

in Poisson distribution, **Figure S7**). Additionally, we observed that the F1 scores of scGCO and Squidpy in **Dataset**, a noisy-free dataset, were still lower than the previous noise-free simulation data, especially scGCO. Therefore, we used **Dataset 1**'s data to generate **a new noise-free simulated dataset with 10,000 simulated genes** (consisting of 1000 SVGs of Set 1 and 9000 non-SVGs resampled from Set 4), as same as the previous simulated data. In the new simulation dataset, scGCO and Squidpy had higher F1 scores (**Figure S7**) than Dataset 1's data and performed similarly to the noise-free data in the previous simulations. **Overall, the mixture noise had a greater impact on the performance of scGCO and Squidpy than expected, and the numbers of SVGs to non-SVGs also influenced their results.**

Part II: The unsuitable underlying data model and normalization mechanism limit the identification ability of SPARK and SpatialDE.

In our simulations, SPARK and SpatialDE perform noticeably weaker compared to other methods. Both SPARK and SpatialDE use the **Gaussian process regression as the underlying data model**, and they use normalization mechanisms to make spatial transcriptomics data approximate a normal distribution. The heterogeneity in actual spatial transcriptomics data is stronger than simulated data and the normalization applied by SPARK and SpatialDE is effective in some real ST datasets. However, the normalization process of SPARK and SpatialDE removes excessive heterogeneity, including the signals from SVGs, limiting their ability to identify SVGs¹. This limitation is evident in the SVG identification performance in our simulations. The normalization mechanisms of SPARK and SpatialDE overcorrected the signals of SVGs. However, we also created **a new simulation with higher heterogeneity** to compare various methods. Specifically, we kept the expression distribution and parameters of the simulated data unchanged, but we considered that some SVG genes had higher expression in the middle circle, while some SVG genes had similar expression in the three circles, as shown in **Figure S8**. **After increasing the heterogeneity of the simulated data, the performance of SPARK and SpatialDE improved, but still lagged behind other methods, especially in datasets with higher sparsity and dispersion (NB and ZINB, which was consistent with the literature conclusion of SPARK-X²**. Compared with simulation results in the previous revision, increasing the heterogeneity of the simulated data did not significantly change the performance of other methods.

I	pval_norm	var_norm	pval_z_sim	pval_sim	var_sim	pval_norm_fdr_bh	pval_z_sim_fdr_bh
0.09436988	0.000000e+00	0.0001142446	0.000000e+00	0.03225806	7.009212e-05	NA	NA
0.09077445	0.000000e+00	0.0001142446	0.000000e+00	0.03225806	7.767311e-05	NA	NA
0.08760758	1.110223e-16	0.0001142446	0.000000e+00	0.03225806	7.759278e-05	NA	NA
0.08732877	1.110223e-16	0.0001142446	0.000000e+00	0.03225806	6.932961e-05	NA	NA
0.08710005	1.110223e-16	0.0001142446	0.000000e+00	0.03225806	9.299862e-05	NA	NA
0.08504473	6.661338e-16	0.0001142446	0.000000e+00	0.03225806	6.593426e-05	NA	NA
0.08457129	9.992007e-16	0.0001142446	1.110223e-16	0.03225806	1.023104e-04	NA	NA
0.08297377	3.219647e-15	0.0001142446	0.000000e+00	0.03225806	7.065855e-05	NA	NA
0.08256707	4.440892e-15	0.0001142446	0.000000e+00	0.03225806	8.502294e-05	NA	NA
0.08254263	4.440892e-15	0.0001142446	4.540812e-14	0.03225806	1.284049e-04	NA	NA
0.08182167	7.549517e-15	0.0001142446	9.992007e-16	0.03225806	1.042324e-04	NA	NA
0.08148981	9.658940e-15	0.0001142446	0.000000e+00	0.03225806	7.395096e-05	NA	NA
0.08122150	1.176836e-14	0.0001142446	5.551115e-16	0.03225806	1.016575e-04	NA	NA
0.08117360	1.210143e-14	0.0001142446	0.000000e+00	0.03225806	7.498224e-05	NA	NA

Figure S1 When expression values were negative, Squidpy encounters **NA** in the **Moran's I** and **p-value** calculation processes, and calculated all the adjusted p-values as **NA** .

Figure S2 (corresponding to Figure S8 in the previous Response) Simulation results for identifying SVGs using simulated data with Gaussian noise. a, Visualization of Pattern: Hotspot with varying level of Gaussian noise. b-e, Simulation results of four different methods (HEARTSVG, scGCO, SPARK-X and Squidpy) on simulated data generated by four distinct distributions (ZINB, ZIP, NB, Poisson). F1 score plots, TPR plots, Precision plots, and FPR plots compare the index values (y-axis) of HEARTSVG (red), scGCO (orange), SPARK-X (green), and Squidpy (purple) across varying levels of Gaussian noise strength (x-axis). Index values were calculated at the adjusted p-value cutoff of 0.05.

Figure S3 (corresponding to Figure S9 in the previous Response) Simulation results for identifying SVGs using simulated data with Gaussian noise. a, Visualization of Pattern: Streaks with varying level of Gaussian noise. b-e, Simulation results of four different methods (HEARTSVG, scGCO, SPARK-X and Squidpy) on simulated data generated by four distinct distributions (ZINB, ZIP, NB, Poisson). F1 score plots, TPR plots, Precision plots, and FPR plots compare the index values (y-axis) of HEARTSVG (red), scGCO (orange), SPARK-X (green), and Squidpy (purple) across varying levels of Gaussian noise strength (x-axis). Index values were calculated at the adjusted p-value cutoff of 0.05.

Figure S4 (corresponding to Figure S10 in the previous Response) Simulation results for identifying SVGs using simulated data with Gaussian noise. a, Visualization of Pattern: Big Circles with varying level of Gaussian noise. b-e, Simulation results of four different methods (HEARTSVG, scGCO, SPARK-X and Squidpy) on simulated data generated by four distinct distributions (ZINB, ZIP, NB, Poisson). F1 score plots, TPR plots, Precision plots, and FPR plots compare the index values (y-axis) of HEARTSVG (red), scGCO (orange), SPARK-X (green), and Squidpy (purple) across varying levels of Gaussian noise strength (x-axis). Index values were calculated at the adjusted p-value cutoff of 0.05.

Figure S5 (corresponding to Figure S11 in the previous Response) Simulation results for identifying SVGs using simulated data with noise of 'Randomly Exchanging Expression Values of Selected Nodes.' a, Visualization of Pattern: Streaks with different percentage of cells random exchanges (%). b-e, Simulation results of four different methods (HEARTSVG, scGCO, SPARK-X and Squidpy) on simulated data generated by four distinct distributions (ZINB, ZIP, NB, Poisson). F1 score plots, TPR plots, Precision plots, and FPR plots compare the index values (y-axis) of HEARTSVG (red), scGCO (orange), SPARK-X (green), and Squidpy (purple) across different percentage of cells random exchanges (x-axis). Index values were calculated at the adjusted p-value cutoff of 0.05.

Figure S6 (corresponding to Figure S12 in the previous Response) Simulation results for identifying SVGs using simulated data with noise of 'Randomly Exchanging Expression Values of Selected Nodes.' a, Visualization of Pattern: Streaks with different percentage of cells random exchanges (%). b-e, Simulation results of four different methods (HEARTSVG, scGCO, SPARK-X and Squidpy) on simulated data generated by four distinct distributions (ZINB, ZIP, NB, Poisson). F1 score plots, TPR plots, Precision plots, and FPR plots compare the index values (y-axis) of HEARTSVG (red), scGCO (orange), SPARK-X (green), and Squidpy (purple) across different percentage of cells random exchanges (x-axis). Index values were calculated at the adjusted p-value cutoff of 0.05.

Figure S6 in the previous Response Simulation results of SVGs identification using simulated data with mixture noise. **a**, Visualization of Pattern: Big triangles with mixture noise. **b-e**, Simulation results of six different methods (HEARTSVG, scGCO, SPARK, SPARK-X, Squidpy) on simulated data generated by four distinct distributions (ZINB, ZIP, NB, Poisson). The bar diagram (sub-panel (1)) shows F1 scores, TPRs, precisions, and FPRs. The heatmap (sub-panel (2)) depicts the comparison of TPR values among six genesets.

Figure S7 Simulation results of scGCO and Squidpy on simulated data with different levels of mixture noise. **Dataset 1** is a noise-free simulated dataset (consisting of 1000 SVGs of Set 1 and 1000 non-SVGs of Set 4). **Dataset 2** is a simulated dataset with weak noise. **Previous Fig S6** is the simulated data of **Figure S6 in the previous response**, with medium noise. **New noise-free** is a noise-free simulated dataset with 10,000 simulated genes (consisting of 1000 SVGs of Set 1 and 9000 non-SVGs resampled from Set 4), as same as the previous simulated data. The bar diagram shows F1 scores, TPRs, precisions, and FPRs. Index values were calculated at the adjusted p-value cutoff of 0.05.

Figure S8 Simulation results for identifying SVGs using simulated data with higher heterogeneity a, Visualization of SVGs and non-SVG. b-e, Simulation results of six different methods (HEARTSVG, scGCO, SPARK, SPARK-X, SpatialDE and Squidpy) on simulated data generated by four distinct distributions (ZINB, ZIP, NB, Poisson). The bar diagram shows F1 scores, TPRs, precisions, and FPRs. Index values were calculated at the adjusted p-value cutoff of 0.05.

2. The author should stop calculating metric such as FPR etc for real biological samples. They updated the true positive set to intersection of all methods tested. But the same logic applies, we simply don't know the set of true SVGs genes. Using the intersection of all methods is flawed just as using intersection of two methods.

Thank you for your comments. We understand your concern and agree with your point of view, and we have removed **Figure 2b** to the Supplementary. However, for the **Figure 2d**, it is quite common to use the reference gene sets as the gold standard for evaluation in single-cell analysis and spatial transcriptomics analysis³⁻⁵. We think that the **Figure 2d** is helpful for showing the application and effect of our method on real data and hope to keep it. However, if really necessary, we can remove it.

3. The memory requirement of scGCO is very different from what was reported in the scGCO paper, which should be further investigated.

Thank you for your careful review and insightful comments. In the scGCO paper, it was indeed reported that the memory requirement for 1 million cells with 100 genes is approximately 8GB. In our simulation, which involved 1 million cells with 10,000 genes, we observed a memory consumption of around 924GB (equivalent to approximately 9.24GB for 100 genes). Furthermore, we conducted tests on mouse hypothalamus data obtained through MERFISH technology, consisting of 1027,848 cells and 161 genes. In this case, the observed memory usage for scGCO was 14.72GB. In summary, while the reported memory requirement in our simulation may seem significantly higher, it aligns with the expected increase given the larger number of genes. The test on real data also supports the reliability of scGCO's performance in practical scenarios.

References

1. Zhang, K., Feng, W. & Wang, P. Identification of spatially variable genes with graph cuts. *Nat. Commun.* **13**, 5488 (2022).
2. Zhu, J., Sun, S. & Zhou, X. SPARK-X: non-parametric modeling enables scalable and robust detection of spatial expression patterns for large spatial transcriptomic studies. *Genome Biol.* **22**, 184 (2021).
3. Finak, G. *et al.* MAST: a flexible statistical framework for assessing transcriptional changes and characterizing heterogeneity in single-cell RNA sequencing data. *Genome Biol.* **16**, 278 (2015).
4. Wu, Z., Zhang, Y., Stitzel, M. L. & Wu, H. Two-phase differential expression analysis for single cell RNA-seq. *Bioinformatics* **34**, 3340–3348 (2018).
5. Xue, R. *et al.* Liver tumour immune microenvironment subtypes and neutrophil heterogeneity. *Nature* **612**, 141–147 (2022).

Response to Reviewer #2:

The authors have addressed all the previous questions by including new SVG tool comparisons, new significant results and importantly including more detailed information on HEARTSVG method procedures and its rationale. This information was shared in the rebuttal letter but also nicely included in the revised manuscript. Thus, the authors show that HEARTSVG provides a considerable improvement of their methodology in their capability to accurately identify SVGs and provide scalability of the tool. Moreover, the manuscript was extensively revised, and it reads well. This paper is suitable for publication.

Before publication the authors need to address a set of minor comments which will not require reviewers' re-evaluation.

Minor Comments

1. In Figure 2, substitute the top title from “Gaussion noise strength” by “Gaussian noise strength”. The same applies to Supplemental Figures S63-S84. Additionally, fix the x-axis for Supplemental Figure S64.

Thank you for your comment. We appreciate your attention to the details. We have made the modifications in Figure 2 in the manuscript (Page 10), and Supplemental Figures S63-S84 in the Supplementary by substituting "Gaussion" with "Gaussian" in the top title. We have fixed the x-axis for Supplemental Figure S64.

2. In Figure 3E, include the relative expression units in the slide bar.

Thank you for your comment. We have addressed your comment by including the relative expression units in the slide bar for Figure 3e the manuscript (Page 15).

3. In Figure 4 panels A include a tittle for the slide bar and substitute “low/high” expression by a numeric scale.

Thank you for your comment. We appreciate your attention to the details. We have addressed your comment by including the relative expression units in the slide bar for Figure 4 in the manuscript (Page 19).

4. Include the Yamamoto test reference in the manuscript.

Thank you for your comment. We have cited the relevant information of the Yamamoto test in the appropriate places in the manuscript (Page 28, Line 570).

5. In Supplemental Figures S1-S3, the x-axis is labelled with "Sample size". Would it be best to replace by "Number of cells" as shown in Figure 1 and Supplemental Figures S62-S84?

Thank you for your comment. We have updated the x-axis labels in Supplemental Figures S1-S3 to "Number of cells," consistent with Figure 1 and Supplemental Figures S62-S84.

Reviewer #1 (Remarks to the Author):

I appreciate the author's efforts. But the simulation results still have obvious conflicts.

1. There is a dramatic discrepancy between performance of methods when different simulation models are used. It appears that ZINB and NB are similar, and ZIP and Pois are similar. It is intriguing that the same method can arrive at very different results when tested on different models. For example, in S4, TPR at Gaussian noise 0.6 is zero for ZINB while TPRs are 1 under ZIP. This observation suggested that model to generate simulation samples can dramatically change performances for tested algorithms. The authors should address roots for this discrepancy, and whether resulting performance measures are valid and fair, when models to simulate data have a huge impact.
2. Gaussian noise level at 0.6 is quite strong as shown by the pictures provided by the authors, then how can all methods maintain a TPR of 1 at Gaussian noise level 0.6 for the ZIP model (And POIS) as shown in S3? This suggests that ZIP and POIS are not good models to evaluate SVG identification because they are intrinsically biased.
3. ScGCO's performance is weird for ZINB and NB models. For example, in S2A, it seems scGCO was unable to identify any genes as SVG, even when noise is zero. This is related to point 1. I pointed out issues with squidpy previously, the author should carefully examine their code and evaluate whether additional errors impacted scGCO.
4. In S2D, TPR becomes zero for all method except scGCO at Gaussian noise 0.6, but corresponding precisions are not zero. The nominators for TPR and precision are the same, both are TP. Then how could precisions be not zero while TPRs are zero? The same error was observed in multiple other graphs as well.

Dear Reviewers:

Thank you for the reviewers' comments concerning our manuscript entitled "HEARTSVG: a fast and accurate method for spatially variable gene identification in large-scale spatial transcriptomic data" (ID: NCOMMS-23-29155C). We sincerely appreciate your time and effort in reviewing our manuscript. However, we respectfully disagree with the assessment provided by you. Detailed responses addressing each of your comments can be found in the attached document.

Comment 1. There is a dramatic discrepancy between performance of methods when different simulation models are used. It appears that ZINB and NB are similar, and ZIP and Pois are similar. It is intriguing that the same method can arrive at very different results when tested on different models. For example, in S4, TPR at Gaussian noise 0.6 is zero for ZINB while TPRs are 1 under ZIP. This observation suggested that model to generate simulation samples can dramatically change performances for tested algorithms. The authors should address roots for this discrepancy, and whether resulting performance measures are valid and fair, when models to simulate data have a huge impact.

Thank you for your comment. However, we disagree with the comment regarding the validity of our simulation mechanism based on the difference in the performance of the method when using different simulation models. We think that this doubt is not justified.

Firstly, in other studies on SVG identification, different methods usually use different assumed distributions to simulate gene expression data, and their performance also varies. The reason is that different distributions have distinct data characteristics, and the same method performs differently on different distributions, just as linear regression works well on normal distribution, but poorly on skewed data. The same group of authors used the Poisson distribution in the SPARK¹ paper (Sun S, Zhu J & Zhou X), but the NB distribution in the SPARK-X² paper (Zhu J, Sun S & Zhou X). SPARK had a high TPR (>0.75) on simulated data generated by the Poisson distribution, but a low TPR (<0.25) on the NB distribution. SOMDE³ (Hao M, Hua K & Zhang X), scGCO⁴ (Zhang K, Feng W & Wang P), and SPARK **all used Poisson distribution, but they had different TPRs in different papers:** scGCO was much higher in its own paper than in SOMDE; SPARK performed better in the scGCO paper than in its own; and SpatialDE's TPR

was higher in the SOMDE and scGCO papers than in the SPARK paper. Interestingly, we noticed that scGCO's paper did not compare with SOMDE in their own simulation, but they added a comparison with SOMDE's results in applications, which made us suspicious.

Secondly, previous studies usually only used one distribution for gene expression simulation. To ensure more fair and comprehensive comparisons, we generated gene expression data following multiple distributions (ZINB, NB, ZIP, and Pois), and performed SVG identification performance evaluation. We think this is essential, and should not be questioned because it is different from the simulation mechanism used in articles such as scGCO. The gene expression distribution in real spatial transcriptomics data is complex and variable, and there is no single distribution that can perfectly fit all genes. A recently published article on spatial transcriptomics simulator SRTsim⁹ (Zhu, J., Shang, L. & Zhou, X.) also confirmed this, "SRTsim relies on four popular count models that include Poisson, ZIP, NB, and ZINB for generating the synthetic data"⁵. These distributions, ZINB, NB, ZIP, and Pois, have been widely used and validated in studies of spatial transcriptomics^{1-3,5-10}. For example, SPARK¹ uses the Poisson distribution, while SPARK-X² uses the NB distribution, even though they are proposed by the same group of authors. SOMDE³ uses Poisson distribution to simulate gene expression data, scGCO⁴ uses Poisson distribution and Normal distribution, Daniel et al. use the NB distribution to generate simulated data, while SpatialDE⁶ and Squidpy¹¹ do not perform simulation.

Comment 2. Gaussian noise level at 0.6 is quite strong as shown by the pictures provided by the authors, then how can all methods maintain a TPR of 1 at Gaussian noise level 0.6 for the ZIP model (And POIS) as shown in S3? This suggests that ZIP and POIS are not good models to evaluate SVG identification because they are intrinsically biased.

Thank you for your comment. We would like to emphasize the following points:

Firstly, the TPR is not exactly 1, but very close to 1.

Secondly, "high TPRs at Gaussian level 0.6 and Gaussian level 0 in the ZIP and Pois distributions", indicates that the addition of Gaussian noise has minimal impact, suggesting that its inclusion is ineffective rather than indicating any invalidation of the data generation mechanism associated with ZIP and Pois distributions.

The comment challenges our simulation mechanism, arguing that it is unreasonable that the TPR is close to 1 in the ZIP and Pois distributions when the Gaussian noise level is 0.6. However, we add Gaussian noise following the procedures described by scGCO and our simulation results are in agreement with those in the scGCO paper too. The scGCO study used Poisson and Normal distributions to simulate gene expression. When the Gaussian noise level was 0.6, the accuracy (>0.995), TPR (>0.96), and F1 score (>0.975) of scGCO were also very close to 1. The downward trend presented in the scGCO paper was due to the different scaling of the y-axis ($0.97\sim 1$, $0.7\sim 1$, $0.825\sim 1$, instead of a uniform $0\sim 1$). Therefore, our simulation mechanism is valid and reliable.

Thirdly, we need to point out that different assumed distributions (ZINB, NB, ZIP, and Pois) to simulate gene expression data result in distinct gene-wise properties (such as mean, variance, coefficient of variation, and zero proportion). Therefore, adding the same level of Gaussian noise to ZINB, NB, ZIP, and Pois had different effects on the signal of SVG. For a simple example, the changes caused by subtracting 0.5 from 100 and 1 are completely different. We believe that it is normal for the same method to perform differently in different distributions of simulated data, rather than a problem with our simulation mechanism. Moreover, the exchanging noise simulation results also confirmed this conclusion. Exchanging noise simulation is a more stable noise addition method, and the impact of adding the same level of exchanging noise on all distributions is consistent and independent of the expression property. Compared with the exchanging noise level of 0, the TPR always showed lower values when the exchanging noise level was 0.6. Given these findings, we are considering removing the Gaussian noise simulation part from the revised manuscript.

We would like to restate that the distributions and parameter settings we used are based on references from other papers and follow mainstream practices. We understand your concerns about the data distribution in Comments 1 and 2. Therefore, we have attached the detailed settings and justifications for the different distribution parameters we used in our simulation in the appendix (Page 8).

Comment 3.ScGCO's performance is weird for ZINB and NB models. For example, in S2A, it seems scGCO was unable to identify any genes as SVG, even when noise is zero. This is related to point 1. I pointed out issues with squidpy previously, the author should carefully examine their code and evaluate whether additional errors impacted scGCO.

Thank you for your comment. However, after careful consideration, we have a different perspective on this matter.

Firstly, the comment regarding our simulation results, because of the poor performance of scGCO in the ZINB and NB distributions, is unfair. In the ZINB and NB distributions, besides our HEARTSVG method, two other methods, Squidpy and SPARK, also showed an effective ability to identify SVGs. Moreover, the NB performance of SPARK-X, SPARK, and SpatialDE was consistent with the performance reported in the SPARK-X paper². In addition, we would like to point out that the simulated data of scGCO used Poisson distribution and Normal distribution in its own paper. It is very unreasonable to use Normal distribution when simulating spatial gene expression data, as spatial transcriptomics data are known to be sparse, over-dispersed, and non-negative^{12,13}.

Secondly, we conducted a fair comparison of scGCO with other methods and discovered that scGCO is not a very powerful method for identifying SVGs. Since the scGCO paper did not involve the simulation of ZINB and NB distributions, which are commonly used to model the gene expression data, we cannot compare with the results of scGCO's paper. Therefore, we collected other papers that make benchmarking methods (including scGCO) or compare their methods with the scGCO. For example:

- ◆ SOMDE³ used Poisson distribution to simulate gene expression data, and the parameter lambda was estimated from the "Slide-seq (nHipp) data¹⁴ near the hippocampus". SOMDE was compared with SpatialDE and scGCO. In all simulations, SOMDE's performance (TPR and FPR) was close to SpatialDE and much better than scGCO.
- ◆ Liang Y, Shi G, Cai R, et al. ¹⁰ compared PROST, SPARK-X, SINFONIA, scGCO, SpatialDE, and Seurat in identifying SVGs in the DLPFC dataset¹⁵. scGCO was clearly weaker than PROST and SPARK-X. SVGs identified by scGCO had lower average Moran'I values.

- ◆ Charitakis N, Salim A, Piers A T, et al.⁸ benchmarked six methods for identifying SVGs, and the results showed that scGCO identified significantly fewer SVGs than other methods, but it controlled FP well.
- ◆ Li Z, Patel Z M, Pinello et al.⁹ compared 14 methods in massive datasets and conducted a systematic evaluation, including 60 simulated datasets generated by four different simulation strategies, 12 real-world transcriptomics, and three spatial ATAC-seq datasets. They found that scGCO performed worse than Moran's I, and "scGCO produced a small number of SVGs (<1000) in most datasets"⁹.
- ◆ Maxspin (Jones D C, Danaher P, Kim Y, et al.)⁷ used NB distribution to generate simulated data and compared the performance of various SV identification methods. The results showed that scGCO had low PR-AUC (around 0.3). They also pointed out in the paper that "scGCO appears to assign a p value of roughly 0.5 to most genes", which was consistent with our findings.

We also pointed out in our previous reply that scGCO's identification ability depends on its initial smooth factor. As shown in **Figure A1**, RPS20 and GAS5 have very similar patterns, but using the default initial smooth factor = 50, only RPS20 can be identified, and GAS5 cannot be identified (adjusted p_value = 0.5). To identify GAS5, a smaller initial smooth factor needs to be set. This is the limitation of scGCO's performance. In **Figure A2**, scGCO missed Calm1 and Calm2, which have clear spatial expression patterns, and assigned a p value of 0.5. In **Figure A2**, scGCO missed Calm1 and Calm2, which have clear spatial expression patterns, and assigned p value=0.5 to Calm1 and Calm2.

Comment 4. In S2D, TPR becomes zero for all method except scGCO at Gaussian noise 0.6, but corresponding precisions are not zero. The nominators for TPR and precision are the same, both are TP. Then how could precisions be not zero while TPRs are zero? The same error was observed in multiple other graphs as well.

Thank you for your comment. We would like to highlight a point regarding Figure S2d. When high-intensity noise was introduced, the results of HEARTSVG, SPARK-X, and Squidpy **were not precisely 0, but rather close to 0**. It's important to note that it is not uncommon for the TPR to be close to zero while the precision approaches 1. This discrepancy arises from the different

benchmarks considered by TPR and Precision. While both TPR and Precision utilize true positives (TP) in their numerator, they assess the model's ability to identify positives based on different criteria. Precision is determined by the proportion of correctly identified positives among all samples predicted as positives, whereas TPR considers the proportion of correctly identified positives among all actual positives. In Figure S2d, the total number of actual positives (the number of SVGs) is 1000. With the addition of high-intensity noise, the signal of SVGs weakened, resulting in HEARTSVG, SPARK-X, and Squidpy identifying only a few SVGs (resulting in low true positives) but with minimal false positives (FP). Consequently, their TPR approached zero, yet they exhibited high precision.

Figure A1 scGCO missed SVGs (RPS29, ARPC3, GAS5) with clear spatial expression patterns comparing with other methods. (a) Visualizations of spatial expressions of gene RPS20, RPS29, ARPC3, and GAS5. **(b)** Venn diagrams of SVGs in the colorectal cancer data identified by HEARTSVG, SpatialDE, SPARK, SPARK-X, scGCO, and Squidpy. **(c)** Marginal expression plots

of gene RPS20, and GAS5 by HEARTSVG. **(d)** Visualizations of graph cuts by scGCO with different initial smooth factor of gene RPS20, and GAS5 by HEARTSVG.

FigureA2 scGCO missed SVGs (Calm1, Calm2) with clear spatial expression patterns comparing with other methods. (a) Visualizations of spatial expressions of Calm1 and Calm2 in the in the Slide-seqV2 cerebellum data. **(b)** Visualizations of graph cuts by scGCO with default initial smooth factor of Calm1 and Calm2.

Parameter Setting:

We used four distributions, Poisson, ZIP, NB, and ZINB, in our simulation. ZINB distribution is suitable for simulating highly sparse data. Poisson, ZIP, NB are suitable for simulating moderately sparse data, where the mean and variance of Poisson distribution are equal, while ZIP, NB can simulate higher dispersion than Poisson distribution. Over-dispersion is a characteristic of spatial transcriptomics data and single-cell data.

We used Poisson distribution to generate simulated data and followed the parameter settings of SPARK¹. Firstly, we introduce the parameters used in the SPARK manuscript. For the gene expression in the i -th spot/cell of non-SVG and non-marked area of SVG, the parameter of the Poisson distribution is:

$$\lambda_{non} = N_i * \exp(-10.2 + \tau_i)$$

For the gene expression in the marked area of SVG, the parameter of the Poisson distribution is:

$$\lambda_{SVG} = N_i * \exp(-9.1 + \tau_i)$$

Where N_i is the total read counts obtained from the real data seqFISH data¹⁶, τ_i is drawn from a normal distribution with mean zero and variance being 0.35. According to the above method, the range of λ_{non} is about (0.01,1), and the range of λ_{SVG} is about (0.03,3). To simplify our simulation, we set $\lambda_{non} = 0.5$, $\lambda_{SVG} = 1.5$, which is three times of the former.

For the simulated data based on the NB distribution, we followed the simulation parameter settings of SPARK-X². For the gene expression in the i -th spot/cell of non-SVG and non-marked area of SVG, the parameters of the NB distribution are: $mu_{non} = 0.5$, $size_{non} = 1.5$. For the gene expression in the marked area of SVG, the parameter size $size_{SVG} = 1.5$ remains unchanged, and $mu_{SVG} = 1.5$ is three times the value of $mu_{non} = 0.5$.

For both ZIP and ZINB distributions, which are zero-inflated models, we need to set a zero-proportion parameter to control the proportion of zeros. To determine this parameter, we refer to the criteria for SVG in SPA-GCN¹⁷: "(1) the percentage of spots expressing the gene in the target domain, that is, in-fraction, is >80%; (2) for each neighboring domain, the ratio of the percentages of spots expressing the gene in the target domain and the neighboring domain(s), that is, in/out fraction ratio, is >1; and (3) the expression fold change between the target and neighboring domain(s)

is >1.5 . If a user is interested in finding SVGs for a particular combination of spatial domains, SpaGCN offers the option to do so."¹⁷

Therefore, in the ZIP distribution, for the gene expression of non-SVG and non-marked area of SVG, the probability of the zero part is 0.6, and the non-zero part follows a Poisson distribution with parameter $\lambda_{non} = 2$. For the gene expression of marked area of SVG, the probability of the zero part is $0.6/3=0.2$, and the non-zero part follows a Poisson distribution with parameter $\lambda_{SVG} = 6$. In this case, in the ZIP distribution, the zero proportion of the gene of non-SVG and non-marked area of SVG is 0.654, and the mean is 0.8. The zero proportion of the gene of the marked area of SVG is 0.202, and the mean is 4.8. Compared with the Poisson distribution, the zero proportions of both are close, but the dispersion and expression level of the gene of the marked area of SVG are higher.

For the ZINB distribution, we followed the parameter settings for highly sparse data in the SPARK-X² and the criteria for SVG in the SPA-GCN¹⁷. We assumed that the gene expression of non-SVG and non-marked area of SVG had more than 94% zeros, while the gene of marked area of SVG had a significantly lower zero proportion. Specifically, for non-SVG and non-marked area of SVG, the probability of the zero part was 0.8, and the non-zero part followed an NB distribution with parameters $mu_{non} = 0.5, size_{non} = 0.5$. For marked area of SVG, the probability of the zero part was $0.8/3=0.267$, and the non-zero part followed an NB distribution with parameters $mu_{non} = 1, size_{non} = 1$. In this case, in the ZINB distribution, the zero proportion of the gene of non-SVG and non-marked area of SVG was 0.941, and the mean was 0.05. The zero proportion of the gene of marked area of SVG was 0.633, and the mean was 0.733.

Here, we note that the SPARK-X² used the NB distribution to generate highly sparse simulated data. For the gene expression of non-SVG and non-marked area of SVG, the parameters of the NB distribution were: $mu_{non} = 0.005, size_{non} = 2.5$, resulting in 99.5% zeros. For the gene expression of marked area of SVG, the parameters of the NB distribution were: $mu_{non} = 0.015, size_{non} = 2.5$, resulting in 98.5% zeros. This means that the genes of non-marked area of SVG and marked area of SVG had very high zero proportions ($>98.5\%$), and very small non-zero expression values (see Figure A3). We argue that, when the expression of the marked area of SVG is almost zero, it is hard to determine whether a gene is a biologically meaningful SVG, so we also referred to the criteria in the SPA-GCN paper¹⁷.

Figure A3. The relationship between gene expression and density in the highly sparse simulated data generated by SPARK-X. The red line represents the genes of non-SVG and non-marked area of SVG, and the blue line represents the genes of marked area of SVG.

References

1. Sun, S., Zhu, J. & Zhou, X. Statistical analysis of spatial expression patterns for spatially resolved transcriptomic studies. *Nat. Methods* **17**, 193–200 (2020).
2. Zhu, J., Sun, S. & Zhou, X. SPARK-X: non-parametric modeling enables scalable and robust detection of spatial expression patterns for large spatial transcriptomic studies. *Genome Biol.* **22**, 184 (2021).
3. Hao, M., Hua, K. & Zhang, X. SOMDE: a scalable method for identifying spatially variable genes with self-organizing map. *Bioinformatics* **37**, 4392–4398 (2021).
4. Zhang, K., Feng, W. & Wang, P. Identification of spatially variable genes with graph cuts. *Nat. Commun.* **13**, 5488 (2022).
5. Zhu, J., Shang, L. & Zhou, X. SRTsim: spatial pattern preserving simulations for spatially resolved transcriptomics. *Genome Biol.* **24**, 39 (2023).
6. Svensson, V., Teichmann, S. A. & Stegle, O. SpatialDE: identification of spatially variable genes. *Nat. Methods* **15**, 343–346 (2018).
7. Jones, D. C. *et al.* An information theoretic approach to detecting spatially varying genes. *Cell Rep. Methods* **3**, 100507 (2023).
8. Charitakis, N. *et al.* Disparities in spatially variable gene calling highlight the need for benchmarking spatial transcriptomics methods. *Genome Biol.* **24**, 209 (2023).
9. Li, Z. *et al.* *Benchmarking Computational Methods to Identify Spatially Variable Genes and Peaks*. <http://biorxiv.org/lookup/doi/10.1101/2023.12.02.569717> (2023)
doi:10.1101/2023.12.02.569717.
10. Liang, Y. *et al.* PROST: quantitative identification of spatially variable genes and domain detection in spatial transcriptomics. *Nat. Commun.* **15**, 600 (2024).
11. Palla, G. *et al.* Squidpy: a scalable framework for spatial omics analysis. *Nat. Methods* **19**, 171–178 (2022).
12. Zhao, P., Zhu, J., Ma, Y. & Zhou, X. Modeling zero inflation is not necessary for spatial transcriptomics. *Genome Biol.* **23**, 118 (2022).
13. Svensson, V. Droplet scRNA-seq is not zero-inflated. *Nat. Biotechnol.* **38**, 147–150 (2020).
14. Rodriques, S. G. *et al.* Slide-seq: A scalable technology for measuring genome-wide expression at high spatial resolution. *Science* **363**, 1463–1467 (2019).
15. Pardo, B. *et al.* spatialLIBD: an R/Bioconductor package to visualize spatially-resolved transcriptomics data. *BMC Genomics* **23**, 434 (2022).
16. Shah, S., Lubeck, E., Zhou, W. & Cai, L. In Situ Transcription Profiling of Single Cells Reveals Spatial Organization of Cells in the Mouse Hippocampus. *Neuron* **92**, 342–357 (2016).
17. Hu, J. *et al.* SpaGCN: Integrating gene expression, spatial location and histology to identify spatial domains and spatially variable genes by graph convolutional network. *Nat. Methods* **18**, 1342–1351 (2021).

Reviewer #2 (Remarks to the Author):

The current version of HEART-SVG manuscript presented by Yuan et al. meets my prior requirements.

Related to the rebuttal letter associated with this manuscript and shared by another reviewer, the authors utilized four distinct data distribution in their simulation of spatial variable genes (SVGs) and justify the rationale behind their selection. I would propose that a shorten version of the comments provided below is included in the final version of the manuscript, to inform the broad scientific community on why these simulations were performed using a variety of data distributions and how that reflects on the observed results.

"The gene expression distribution in real spatial transcriptomics data is complex and variable, and there is no single distribution that can perfectly fit all genes. A recently published article on spatial transcriptomics simulator SRTsim9 (Zhu, J., Shang, L. & Zhou, X.) also confirmed this, "SRTsim relies on four popular count models that include Poisson, ZIP, NB, and ZINB for generating the synthetic data" 5. These distributions, ZINB, NB, ZIP, and Pois, have been widely used and validated in studies of spatial transcriptomics^{1-3,5-10}."

Additionally, it would be of interest if the authors can comment on using the average false discovery proportion (FDP) compared with nominal false discovery rate (FDR) as a benchmark of performance for a given statistical method. Benidt et al. proposed that this methodology can be used to access how a well an analysis method controls FDR for a given simulation method: (<https://www.ncbi.nlm.nih.gov/pmc/articles/PMC4481850/>)

Reviewer #3 (Remarks to the Author):

The previous reviewer has raised several valid issues regarding the rigor of the evaluation process, which I agree should be addressed carefully. The large discrepancy between different simulations is concerning and needs careful investigation. If the source of discrepancy only comes from the underlying probability distribution of gene expression data used in data simulation. then it would be helpful to know which part of the analysis procedure is highly sensitive to the choice of such distributions. I also agree with the previous reviewer that the accuracy for Gaussian noise seems too good to be true. All in all, the comments raised by the previous reviewer have not been satisfactorily addressed by the rebuttal.

Dear Reviewers:

Thank you for your comments concerning our manuscript entitled "HEARTSVG: a fast and accurate method for spatially variable gene identification in large-scale spatial transcriptomic data" (ID: NCOMMS-23-29155D-Z). We have carefully considered each of your comments and have made the necessary revisions to address them. We provide a detailed point-to-point response for each comment in the following document and make corresponding changes in the manuscript.

We hope these revisions will fully address your concerns and substantially improve our submission.

Resonse to Reviewer #2:

The current version of HEART-SVG manuscript presented by Yuan et al. meets my prior requirements.

Comment1 Related to the rebuttal letter associated with this manuscript and shared by another reviewer, the authors utilized four distinct data distribution in their simulation of spatial variable genes (SVGs) and justify the rationale behind their selection. I would propose that a shorten version of the comments provided below is included in the final version of the manuscript, to inform the broad scientific community on why these simulations were performed using a variety of data distributions and how that reflects on the observed results. “The gene expression distribution in real spatial transcriptomics data is complex and variable, and there is no single distribution that can perfectly fit all genes. A recently published article on spatial transcriptomics simulator SRTsim9 (Zhu, J., Shang, L. & Zhou, X.) also confirmed this, "SRTsim relies on four popular count models that include Poisson, ZIP, NB, and ZINB for generating the synthetic data" 5. These distributions, ZINB, NB, ZIP, and Pois, have been widely used and validated in studies of spatial transcriptomics1–3,5–10.”

Thank you for your comment. We have included a shorten version of this section in the manuscript (Page 7, line 141-145), and the full version has been incorporated into the Supplementary.

Comment2 Additionally, it would be of interest if the authors can comment on using the average false discovery proportion (FDP) compared with nominal false discovery rate (FDR) as a benchmark of performance for a given statistical method. Benidt et al. proposed that this methodology can be used to assess how well an analysis method controls FDR for a given simulation method:

(<https://www.ncbi.nlm.nih.gov/pmc/articles/PMC4481850/>)

We greatly appreciate your insightful suggestions, which have broadened our perspective in assessing various methods. We evaluated the average False Discovery Proportion (FDP) against the nominal False Discovery Rate (FDR) using noise-free simulated data at mean levels of 0.5 and 0.25 (Page 9 of this response for detailed simulation settings). We examined the average FDP of various methods at nominal FDR settings of 0.01, 0.05, and 0.1. Our findings (Fig. S1) indicate that Squidpy's average FDP consistently exceeded the nominal FDR. In contrast, HEARTSVG and SPARK-X maintained an average FDP below the nominal FDR consistently. For scGCO, the average FDP surpassed the nominal FDR at the mean level of 0.5 simulated data with moderate data sparsity. However, at the lower expression level with higher sparsity (mean level of 0.25), scGCO's average FDP fell below the nominal FDR, albeit with a concurrently low TPR. These results highlight the superior performance of HEARTSVG and SPARK-X in controlling false positives.

We noted that the literature¹ suggests "Simulation experiments relying on parametric models may offer an overly optimistic assessment of a method's efficacy". Given the inability to generate spatial transcriptomics simulated data for SVGs using the SimSeq algorithm, we embarked on an alternative intriguing endeavor. We analyzed the changes of average FDP as the capacity to identify SVGs deteriorated (reflected by a decrease in F_1 score). Specifically, we modified the approach of adding Gaussian noise as Reviewer 3's comment. With this modified approach to Gaussian noise incorporation, we observed that as the level of noise increased, all patterns became increasingly difficult to detect (Fig. S2a), causing a decline in the F_1 scores of all methods towards zero (Fig. S2b). At a nominal FDR of 0.05, we analyzed how the average FDP of various methods altered with increasing noise. Our results (Fig. S2b) indicated that Squidpy's average FDP consistently exceeded the nominal FDR. In contrast, HEARTSVG and SPARK-X's average FDP remained below the nominal FDR. For scGCO, the average FDP began to rise with increasing noise levels and ultimately surpassed the nominal FDR.

In summary, HEARTSVG stands out in controlling false positives, indicating that its identified SVGs are highly credible. However, this may also suggest that HEARTSVG's selection is relatively conservative.

Fig. S1 a, Plot of average FDP at different nominal FDR. The solid gray line represents an average FDP that is exactly equal to nominal FDR. b, Plot of average F_1 score at different nominal FDR. c, Plot of average TPR at different nominal FDR.

Fig. S2 a, Visualizations of spatial patterns, incorporating noise using the modified Gaussian noise addition approach. b, Plot of average FDP, F_1 score and TPR at nominal FDR=0.05. The solid gray line represents nominal FDR=0.05.

Reference

1. Benidt, S. & Nettleton, D. SimSeq: a nonparametric approach to simulation of RNA-sequence datasets. *Bioinformatics* 31, 2131–2140 (2015).

Response to Reviewer #3:

Comments The previous reviewer has raised several valid issues regarding the rigor of the evaluation process, which I agree should be addressed carefully. The large discrepancy between different simulations is concerning and needs careful investigation. If the source of discrepancy only comes from the underlying probability distribution of gene expression data used in data simulation. then it would be helpful to know which part of the analysis procedure is highly sensitive to the choice of such distributions. I also agree with the previous reviewer that the accuracy for Gaussian noise seems too good to be true. All in all, the comments raised by the previous reviewer have not been satisfactorily addressed by the rebuttal.

Thank you for your insightful comments. We appreciate the opportunity to enhance the rigor of our study. We will address them in the following two points: **the source of discrepancy and Gaussian noise.**

Comment1 If the source of discrepancy only comes from the underlying probability distribution of gene expression data used in data simulation. then it would be helpful to know which part of the analysis procedure is highly sensitive to the choice of such distributions.

Thank you for the comment. In fact, it is the **data characteristics** of different distributions (**expression levels, degree of dispersion, and sparsity**) that significantly affect the performance of various methods in identifying SVGs. We have added two new simulations in which we adjusted the parameters of all distributions (*ZINB*, *ZIP*, *NB*, *Pois*) to ensure the data characteristics (mean, dispersion, and sparsity levels) produced are similar. It is important to note that, due to the inherent properties of the distributions, *ZIP* and *ZINB* will inherently exhibit greater dispersion than *Pois* and *NB* under similar mean and variability levels.

Our analysis shows that **when the data characteristics are aligned, each method's performance is relatively stable across different simulated datasets, regardless of the underlying distribution (Fig. S1-S3)**. We have also conducted sensitivity analyses to pinpoint which steps in the analysis are particularly sensitive to variations in data characteristics. The main conclusions are as follows. The detailed results of our sensitivity analyses, which are provided in Figs. S4-S11, elucidate the specific parts of the analysis that are sensitive to these data characteristics.

Main Conclusions:

1. scGCO is significantly impacted by increased data dispersion, whereas HEARTSVG, SPARK-X, and Squidpy are robust under these conditions.
2. An increase in data sparsity and a decrease in overall expression levels generally diminishes all methods' capacities to identify SVGs.
3. The normalization procedures by spatialDE and SPARK distorts the data characteristics, negatively affecting SVG detection.
4. A low count of cells or spots uniformly hinders all methods' abilities to identify SVGs effectively.

There are details.

We generated two new simulations using different distributions with similar data characteristics (mean, dispersion, and sparsity level). Each method demonstrated a similar ability to identify SVGs across these simulated datasets from different distributions (Fig. S1-S3). In simulated datasets with higher dispersion (*ZIP*, *ZINB*), scGCO showed lower F_1 scores. Furthermore, in simulated datasets with higher sparsity and lower expression levels (*mean level 0.25*), the SVG identification capabilities of all methods diminished.

Fig. S1 a, Visualization of Ring Pattern for different distributions that share similar data characteristics (mean, dispersion, and sparsity level). b, Each method has similar F_1 score across different simulated datasets. scGCO has lower F_1 score on datasets characterized by higher dispersion, such as those from *ZIP* and *ZINB* distributions.

Fig. S2 a, Visualization of Ring Pattern for different distributions that share similar data characteristics (mean, dispersion, and sparsity level). b. Each method has similar F_1 score across different simulated dataset.

Fig. S3 Each method has similar F_1 score across different simulated dataset of Big circles Pattern. These datasets generated by different distributions sharing similar data characteristics (mean, dispersion, and sparsity level).

The parameter settings for the two new simulations are as follows.

1) Simulations with medium sparsity and high expression levels.

For non-SVGs and the non-marked area of SVGs, the data characteristics (mean, dispersion, and sparsity level) generated by different distributions were adjusted to be close to a $mean = var = 0.5$, $P(X = 0) = 0.6$ (approximating $Pois(\lambda = 0.5)$). For the marked area of SVGs, the data characteristics were made to approximate a $mean = var = 1.5$, $P(X = 0) = 0.3$ (approximating $Pois(\lambda = 1.5)$). The specific parameters are as

follows.

Tab. S1 Parameters of different distributions close to a $mean = var = 0.5, P(X = 0) = 0.6$ and $mean = var = 1.5, P(X = 0) = 0.3$.

	non-SVG and non-marked area of SVGs			marked area of SVGs		
	probability of extra zeros	non-zero part		probability of extra zeros	non-zero part	
		mu/lambda	size		mu/lambda	size
Pois	-	0.5	-	-	1.5	-
NB	-	0.5	30	-	1.5	30
ZIP	0.4	0.833	-	0.2	1.8	-
ZINB	0.5	1	30	0.25	2	30

2) Simulations with high sparsity and low expression levels

For non-SVGs and the non-marked area of SVGs, the data characteristics (mean, dispersion, and sparsity level) generated by different distributions were adjusted to be close to a $mean=var=0.25, P(X=0)=0.8$ (approximating $Pois(\lambda = 0.25)$). For the marked area of SVGs, the data characteristics were made to approximate a $mean=var=0.75, P(X=0)=0.5$ (approximating $Pois(\lambda = 0.75)$). The specific parameters are as follows.

Tab. S2 Parameters of different distributions close to a $mean=var=0.5, P(X=0)=0.6$ and $mean=var=1.5, P(X=0)=0.3$.

	non-SVG and non-marked area of SVGs			marked area of SVGs		
	probability of extra zeros	non-zero part		probability of extra zeros	non-zero part	
		mu/lambda	size		mu/lambda	size
Pois	-	0.25	-	-	0.75	-
NB	-	0.25	30	-	0.75	30
ZIP	0.5	0.5	-	0.2	0.95	-
ZINB	0.5	0.5	30	0.2	0.95	30

Sensitivity Analysis

1. We found that scGCO is significantly impacted by increased data dispersion, whereas HEARTSVG, SPARK-X, and Squidpy are robust under these conditions.

In simulations of our previous manuscript, using *NB* and *Pois* distributions, HEARTSVG, SPARK-X, and Squidpy showed similar performances across both sets of simulated data. However, scGCO showed notable differences, performing significantly worse on the *NB* distribution than on the *Pois* distribution.

The previous simulations with *NB*(1.5,0.5) and *Pois*(0.5) distributions had similar spatial expression patterns (Fig. S4a), equal means (Non-SVG: both *NB* and *Pois* with $\mu = 0.5$; SVG: both *NB* and *Pois* with $\mu = 1.5$) and similar sparsity levels (Fig. S4b). Yet, the *NB* (*size* = 1.5) distribution is more right-skewed, indicating stronger overdispersion (Fig. S4b).

As we know, with the 'size' parameter in the *NB* distribution increases, the data dispersion decreases. When 'size' approaches infinity, the *NB*(*size*, μ) converges to a Poisson distribution *Pois*(λ) with $\lambda = \mu$. Fig-S1c demonstrated that, the *NB*(*size* = 30) and *Pois* distributions' shapes are almost identical ($\lambda = \mu$) (Fig. S4c). Therefore, we generated two sets of *NB* simulation data with *size* = 30 and *size* = 5 (the μ parameter same as before) to compare with the previous *NB*(1.5,0.5) and *Pois*(0.5) distribution results. The simulation results (Fig. S5) showed that with *NB*(*size* = 30), as dispersion decreases, scGCO's F_1 score significantly improves (Fig. S5b), aligning with the F_1 score seen with the Poisson distribution (Fig. S5c). We conducted similar simulations on the 'Big squares' pattern and obtained consistent results (Fig. S6). Compared to *NB*(*size* = 30), *NB*(*size* = 5), *NB*(*size* = 1.5), with a reduced 'size' leads to increased dispersion, significantly diminishing scGCO's ability to identify SVGs, while HEARTSVG, SPARK-X, and Squidpy showed no significant change, demonstrating greater robustness.

Fig. S4 a, Visualizations of the 'Ring pattern' SVGs. Gene expression distributions correspond to $NB(\text{size} = 1.5)$, $NB(\text{size} = 30)$, and Poisson distribution, with same 'mean' parameter. Their visual appearances are fundamentally similar. b, Density comparison of Poisson and $NB(\text{size} = 1.5)$. $NB(\text{size} = 1.5)$ distribution is more right-skewed, indicating stronger overdispersion. c, Density comparison of Poisson and $NB(\text{size} = 30)$. $NB(\text{size} = 30)$ and Poisson distributions' shapes are almost identical.

Fig. S5 a, Visualizations of the 'Ring pattern' SVGs. Gene expression distributions correspond to $NB(size = 1.5)$, $NB(size = 5)$, $NB(size = 30)$, and Poisson distribution, with same 'mean' parameter. Their visual appearances are similar. b, F_1 score comparison of all methods on simulations using $NB(size = 5)$ and $NB(size = 30)$. c, F_1 score comparison of all methods on simulations of previous manuscript. The SVG identification capability of scGCO diminished with high dispersion (small 'size' parameter).

Fig. S6 Big circles Pattern. a, F_1 score comparison of all methods on simulations using $NB(size = 5)$ and $NB(size = 30)$. b, F_1 score comparison of all methods on simulations of previous manuscript. The SVG identification capability of scGCO diminished with high dispersion (small 'size' parameter).

2. An increase in data sparsity and a decrease in overall expression levels generally diminishes all methods' capacities to identify SVGs.

High sparsity often coexists with low expression levels in single-cell and spatial transcriptomics data. In simulations of our previous manuscript, we generated simulated scenarios with high sparsity and low expression levels using the *ZINB* distribution. In these simulations, the non-SVGs had over 94% zeros, while the SVGs had over 60% zeros. High sparsity is common in data generated by techniques like Slide-seqV2, HDST, Visium HD and Stereo-seq at single-cell or subcellular resolution.

To further investigate this, we introduced a new *NB* distribution with parameters (SVG: $NB(size = 0.5, \mu = 0.73)$, non-SVG: $NB(size = 0.065, \mu = 0.1)$), aiming to approximate the original *ZINB* distribution (Fig. S7a). We generated simulated data following this new *NB* distribution and compared it with the previous *ZINB* simulation results. The results (Fig. S7b) show that all methods performed on the new *NB* simulated data that were generally consistent with the original *ZINB* results. The comparison of the new *NB* simulation with the original *NB* results demonstrated that all methods' SVG identification capabilities decreased on the more sparsely distributed new *NB* data.

Additionally, we conducted another simulation (Tab.S3). Using Poisson distributions with decreasing λ values: ($Pois(\lambda = 0.5)$, $Pois(\lambda = 0.25)$, $Pois(\lambda = 0.1)$). As we know, for Poisson distribution, as the parameter λ decreased, the data sparsity increased, and overall expression levels decreased. Similar to the previous simulation, all methods exhibited decreased SVG identification capabilities as λ decreased. HEARTSVG and Squidpy showed greater robustness to changes in sparsity than others.

Fig. S7 a, Density comparison of new *NB* and original *ZINB*. Their distributions' shapes are almost identical. b, F_1 score comparison of all methods on simulated data from new *NB* distributions. c, F_1 score comparison of all methods on simulations of previous manuscript.

Tab. S3 new Poisson distributions parameters.

		non-SVG and non-marked area of SVGs	marked area of SVGs
Pois (lambda)	$Pois(\lambda = 0.1)$	0.1	0.3
	$Pois(\lambda = 0.25)$	0.25	0.75
	$Pois(\lambda = 0.5)$	0.5	1.5

Fig. S8 a, Visualizations of the 'Ring pattern' SVGs. Gene expression distributions correspond to $(Pois(\lambda = 0.5), Pois(\lambda = 0.25), Pois(\lambda = 0.1))$. As the parameter λ decreased, the data sparsity increased, overall expression levels decreased, and visual clarity diminished. The color pattern distribution across the plots gets progressively sparser from left to right, illustrating the effect of decreasing the lambda parameter on the sparsity of the generated data. b, Ring Pattern, comparing the F_1 scores of four different methods across three simulated scenarios with varying λ for Poisson distributions: 0.5, 0.25, and 0.1. c, Big circles Pattern, comparing the F_1 scores of four different methods across three simulated scenarios with varying λ for Poisson distributions: 0.5, 0.25, and 0.1. The bar chart clearly visualizes the decreasing F_1 scores of four methods as data sparsity increases with decreasing λ values.

3. The normalization procedures by spatialDE and SPARK distorts the data characteristics, negatively affecting SVG detection.

In our simulations, SPARK and SpatialDE perform noticeably weaker compared to other methods. Both SPARK and SpatialDE use the **Gaussian process regression as the underlying data model**. It is well-known that spatial transcriptomics data do not follow a normal distribution. SPARK and SpatialDE employ additional normalization mechanisms to approximate the spatial transcriptomics data to a normal distribution before modeling and identifying SVGs¹. However, the normalization mechanism of SPARK and SpatialDE removes excessive heterogeneity, including signals from SVGs, which limits their ability to identify SVGs. Fig. S9 displays SVGs' visualizations before and after SPARK normalization. These visualizations showed the effect of normalization mechanism on spatial gene expression data. The normalization procedures of SPARK and SpatialDE overcorrected the signals of SVGs. Nevertheless, to facilitate a comprehensive comparison of various methods, we still created a **new simulation with higher heterogeneity**. This simulated data possesses increased heterogeneity in order to mitigate the impact of normalization procedures. We maintained the expression distribution and parameters constant, while incorporating variations such as higher expression in the central circle for some SVG genes, and similar expression across three circles for others, as shown in Figure S10. Upon increasing the heterogeneity in the simulated data, SPARK and SpatialDE's performance improved, albeit still not on par with other methods. This was particularly evident in datasets with higher sparsity and dispersion (*NB* and *ZINB*), aligning with the findings reported for SPARK-X¹ in the literature. Notably, enhancing the heterogeneity did not significantly alter the performance of the other methods compared to results from the previous revision.

Fig. S9 a, SVGs' visualization of Big squares in the simulated data. b, SVG in the MERFISH data. The left plot shows the original spatial expression for SVGs. The right plot shows the spatial expression for SVGs after SPARK normalization.

Fig. S10 Simulation results for identifying SVGs using simulated data with higher heterogeneity a, Visualization of SVGs and non-SVG. b-e, Simulation results of six different methods (HEARTSVG, scGCO, SPARK, SPARK-X, SpatialDE and Squidpy) on simulated data generated by four distinct distributions (*ZINB*, *ZIP*, *NB*, *Poisson*).

4. A low count of cells or spots uniformly hinders all methods' abilities to identify SVGs effectively.

Although unrelated to the data distribution and data characteristics, our research indeed found that the capability of all methods to identify SVGs diminishes when the dataset contains a smaller number of cells/spots. We generated a new simulations with the same Poisson distribution parameters as the previous simulation in the manuscript, but with the number of cells set to 500. Compared to the previous simulation results, the new simulation showed a marked decrease in the F_1 scores for all methods (Fig. S11).

Fig. S11 a, New simulation results with the number of cells set to 500. b, Previous simulation results with the number of cells set to 3000.

Comment 2 I also agree with the previous reviewer that the accuracy for Gaussian noise seems too good to be true.

In the previous manuscript, we applied the same level of Gaussian noise to all simulated data from different distributions. For ZINB and NB data, all methods' F_1 scores decreased as the noise level increased. However, after adding Gaussian noise to ZIP and Poisson data, all methods' F_1 scores remained close to 1. We observed that this phenomenon arises due to the varying expression levels, sparsity, and dispersion characteristics across different simulated data distributions, resulting in differing levels of difficulty for spatial pattern identification. Thus, the impact of the same noise level also varied. Figure S12 illustrates the visualization of the same level of Gaussian noise added to simulated data from different distributions, where ZIP and Poisson data still exhibit clear patterns even after the addition of 0.6 noise. In contrast, the visual patterns in ZINB and NB data became blurred after the addition of 0.6 noise.

Fig. S12Ring Pattern. Visualizations of simulated data from different distributions with increasing levels of Gaussian noise, as presented in the previous manuscript.

Gaussian noise is commonly used in the field of computer vision. Consequently, we referred to articles in the field of computer vision and modified the approach for adding Gaussian noise. To ensure a consistent impact across all images, Min-Max normalization is performed before noise addition.

New adding Gaussian noise:

1. Min-Max normalization:

We first normalized the expression data, scaling gene expression values to a uniform range between 0 and 1. This helps maintain consistency in expression levels across different distributions.

2. Generating and adding noise:

We generated noise from a Gaussian distribution $N(0, \sigma)$ and added it to the normalized data, ensuring that the post-noise addition pixel values remained within the valid range.

3. Reverse normalization:

After noise addition, we reverted the expression values from their normalized state back to their original scale.

4. We applied methods to identify SVGs.

Following this modified Gaussian noise addition approach, we found that adding the same level of Gaussian noise consistently impacted different data features. The variation in F_1 scores for all methods was also consistent across the different distributions of simulated data with Gaussian noise (Fig. S13-S15).

ZIP: Patterns became blurred when noise exceeded 0.3, and three methods' F_1 scores decreased from 0.3 noise.

Pois: Patterns became blurred when noise exceeded 0.2, leading to a noticeable decline in performance across methods.

NB: Similar to Poisson, patterns blurred when noise exceeded 0.1, three methods' F_1 score declined significantly.

ZINB: Given its inherent sparsity and dispersion, initial patterns were already somewhat blurred. Adding 0.05 noise resulted in blurred spatial pattern, and three methods' F_1 score declined significantly from 0.05 noise onward.

Similar to our observations in the sensitivity analysis, we found that the performance of scGCO declines with increased Gaussian noise, leading to unreliable outcomes and fluctuations. We hypothesize that this is due to the fixed hyperparameter, the initial factor, but the influence of the initial factor on the direction of result variations remains unclear. The pertinent results have been included in the appendix (Fig. A1).

Fig. S13 Ring Pattern. a, Visualizations of simulated data from various distributions, incorporating noise using the modified Gaussian noise addition approach. b, Comparison of F_1 scores on the new Gaussian noise simulated data.

Fig. S14 Streaks Pattern. a, Visualizations of simulated data from various distributions, incorporating noise using the modified Gaussian noise addition approach. b, Comparison of F_1 scores on the new Gaussian noise simulated data.

Fig. S15 Hotspot Pattern. a, Visualizations of simulated data from various distributions, incorporating noise using the modified Gaussian noise addition approach. b, Comparison of F_1 scores on the new Gaussian noise simulated data.

Appendix

Fig. A1 Comparison of F_1 scores of four methods on the new Gaussian noise simulated data.

Reference

1. Zhu, J., Sun, S. & Zhou, X. SPARK-X: non-parametric modeling enables scalable and robust detection of spatial expression patterns for large spatial transcriptomic studies. *Genome Biol* 22, 184 (2021).

Reviewer #2 (Remarks to the Author):

The revised version of HEART-SVG manuscript presented by Yuan et al. meets my prior requirements. The authors have successfully addressed my last set of comments. Although the supplementary materials already include 103 pages of additional results, I would suggest including the analysis shared by the authors in their last rebuttal letter.

Reviewer #3 (Remarks to the Author):

The sensitivity analysis added in this revision is informative, but it would only be useful if it is incorporated in the manuscript, which does not seem to be the case now.

Dear Reviewers:

Thank you for your valuable feedback on our manuscript titled "HEARTSVG: a fast and accurate method for spatially variable gene identification in large-scale spatial transcriptomic data" (ID: NCOMMS-23-29155E). We have thoroughly reviewed your comments and have incorporated the necessary revisions to enhance the quality of our submission. Below, we present a comprehensive response to each of your suggestions, accompanied by corresponding modifications made to the manuscript.

We hope these revisions adequately address your concerns and significantly strengthen our manuscript.

Resonse to Reviewer #2:

The revised version of HEART-SVG manuscript presented by Yuan et al. meets my prior requirements. The authors have successfully addressed my last set of comments. Although the supplementary materials already include 103 pages of additional results, I would suggest including the analysis shared by the authors in their last rebuttal letter.

Thank you for acknowledging the revisions to our manuscript. We appreciate your suggestions. We have incorporated the analysis mentioned in our last rebuttal letter into the manuscript (Page 8, lines 185-187) and included all relevant content in the Supplementary (Pages 114-116) to ensure the completeness and comprehensibility of the paper.

Resonse to Reviewer #3:

The sensitivity analysis added in this revision is informative, but it would only be useful if it is incorporated in the manuscript, which does not seem to be the case now.

Thank you for your comment. We have revised the paper to include the sensitivity analysis within the manuscript (Page 8, lines 180-185) and included the full sensitivity analysis in the Supplementary (Pages 110-113) to ensure its relevance and usefulness.